# A Symmetry-Aware Exploration of Bayesian Neural Network Posteriors

**Olivier Laurent,**[1,2] **Emanuel Aldea**[1] **& Gianni Franchi**[2,†]

SATIE, Paris-Saclay University,[1] U2IS, ENSTA Paris, Polytechnic Institute of Paris[2]

## Abstract

The distribution of modern deep neural networks (DNNs) weights – crucial for uncertainty quantification and robustness – is an eminently complex object due to its extremely high dimensionality. This paper presents one of the first large-scale explorations of the posterior distribution of deep Bayesian Neural Networks (BNNs), expanding its study to real-world vision tasks and architectures. Specifically, we investigate the optimal approach for approximating the posterior, analyze the connection between posterior quality and uncertainty quantification, delve into the impact of modes on the posterior, and explore methods for visualizing the posterior. Moreover, we uncover weight-space symmetries as a critical aspect for understanding the posterior. To this extent, we develop an in-depth assessment of the impact of both permutation and scaling symmetries that tend to obfuscate the Bayesian posterior. While the first type of transformation is known for duplicating modes, we explore the relationship between the latter and L2 regularization, challenging previous misconceptions. Finally, to help the community improve our understanding of the Bayesian posterior, we release the first large-scale checkpoint dataset, including thousands of real-world models, along with our code.

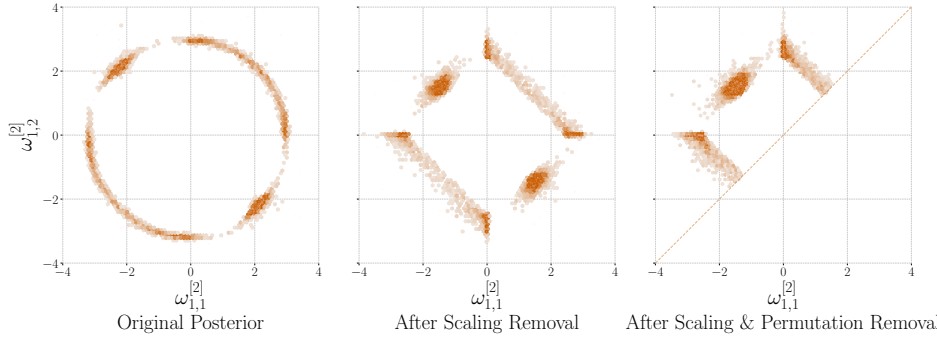

Figure 1: **Weight-space symmetries greatly impact the estimated Bayesian posterior**. Permutation symmetries clearly increase the number of modes of the posterior distribution in the case of the last layer of a 2-hidden neuron perceptron, as detailed in Section 3.1.

## 1 Introduction

Despite substantial advancements in deep learning, Deep Neural Networks (DNNs) remain black box models. Various studies have sought to explore DNN loss landscapes (Li et al., 2018; Fort & Jastrzebski, 2019; Fort & Scherlis, 2019; Liu et al., 2022) to achieve a deeper understanding of these models. Recent works have, for instance, unveiled the interconnection of the modes obtained with Stochastic Gradient Descent (SGD) via narrow pathways that link pairs of modes, or through tunnels that connect multiple modes simultaneously (Garipov et al., 2018; Draxler et al.,

[†]corresponding author – `gianni.franchi@ensta-paris.fr`

2018). This mode connectivity primarily arises from scaling and permutation invariances, which imply that numerous weights can represent the same exact function (*e.g.*, Entezari et al. (2022)). Several studies have delved into the relationship between these symmetries and the characteristics of the loss landscape (Neyshabur et al., 2015; Brea et al., 2019; Entezari et al., 2022). Our work investigates the connections between these symmetries and the distribution of DNN weights, a crucial aspect for uncertainty quantification. These connections are highlighted in Figure 1.

Uncertainty quantification plays a pivotal role in high-stakes industrial applications – such as autonomous driving (Levinson et al., 2011; McAllister et al., 2017; Sun et al., 2019) – where reliable predictions and informed decision-making are paramount. In such critical domains, understanding and effectively managing uncertainties, particularly the model-related epistemic uncertainties (Hora, 1996) arising from incomplete knowledge, is essential. Amongst the various methods introduced to address these challenges, Bayesian Neural Networks (BNNs) (Tishby et al., 1989) offer a principled and theoretically sound approach. BNNs quantify uncertainty by probabilistically modeling beliefs about parameters and outcomes (Tishby et al., 1989; Hinton & Van Camp, 1993). However, this perspective faces significant hurdles when applied to deep learning, primarily related to scalability (Izmailov et al., 2021) and the precision of approximations (MacKay, 1995). Due to their very high dimension, BNNs struggle to estimate the posterior distribution, *i.e.*, the probability density that any set of model parameters/hypothesis $\boldsymbol{\omega}$ generated the observed data $\mathcal{D}$ with a given prior.

Diverging from methods such as the Maximum Likelihood Estimate or Maximum A Posteriori (also Tishby et al. (1989)), which we typically derive through gradient descent optimization of cross-entropy (with L2 regularization for the latter), BNNs assign a probability to each possible model (or hypothesis) and offer predictions considering the full extent of possible models. In mathematical terms, denoting the target as $y$, the input vector as $\boldsymbol{x}$, and the weight space as $\Omega$, we can express this approach through the following intractable formula, often referred to as the marginalization on the parameters of the model (Tishby et al., 1989; Rasmussen et al., 2006):

$$p(y \mid \boldsymbol{x}, \mathcal{D}) = \int_{\boldsymbol{\omega} \in \Omega} p(y \mid \boldsymbol{x}, \boldsymbol{\omega}) p(\boldsymbol{\omega} \mid \mathcal{D}) d\boldsymbol{\omega}. \tag{1}$$

The posterior distribution $p(\boldsymbol{\omega}|\mathcal{D})$ assumes a central and arguably the most critical role in BNNs – and many successful methods for quantifying uncertainty can be viewed as attempts to approximate this posterior, each with its own trade-offs in terms of accuracy and computational efficiency, as illustrated in previous research (Blundell et al., 2015; Gal & Ghahramani, 2016; Lakshminarayanan et al., 2017). While prior work (Kuncheva & Whitaker, 2003; Fort et al., 2019; Ortega et al., 2022) has established the importance of achieving *diversity* in the sampled DNNs drawn from the posterior – particularly when dealing with uncertain input data – permutation and scaling symmetries amongst hidden units in neural networks may lead to an increased number of local minima (Zhao et al., 2023) with no diversity. In the context of BNNs, this phenomenon could result in a proliferation of functionally equivalent modes within the posterior distribution reducing the diversity within the inevitably limited number of samples, and degrading the quality of the uncertainty estimates.

This paper delves into the impact of weight symmetries on the posterior distribution. While there have been numerous efforts to visualize the loss landscape, we explore the possibility of conducting similar investigations for the posterior distribution. Additionally, we introduce a protocol for assessing the quality of posterior estimation and examine the relationship between posterior estimation and the accuracy of uncertainty quantification. Specifically, our contributions are as follows:

**(1)** We build a new mathematical formalism to highlight the different impacts of the permutation and scaling symmetries on the posterior and uncertainty estimation in DNNs. Notably, we explain the seeming equivalence of the marginals in Figure 1. We also perform the first in-depth exploration of the existence of scaling symmetries and their overlooked effect. **(2)** We evaluate the quality of various methods for estimating the posterior distribution on real-world applications using the Maximum Mean Discrepancy, offering a practical benchmark to assess their performance in capturing uncertainty. **(3)** We release Checkpoints, a new dataset including the weights of thousands of models across various computer vision tasks and architectures, ranging from MNIST to TinyImageNet. This dataset is intended to facilitate further exploration and collaboration in the field of uncertainty in deep learning. **(4)** Our investigation delves into the proliferation of modes in the context of posterior symmetries and exhibits the capacity of ensembles to converge toward non-functionally equivalent modes. Furthermore, we discuss the influence of symmetries in the training process.

## 2    RELATED WORK

**Epistemic uncertainty, Bayesian inference, and posterior.** Epistemic uncertainty (Hora, 1996; Hüllermeier & Waegeman, 2021) plays a crucial role in accurately assessing predictive model reliability. However – and despite ongoing discussions – estimating this uncertainty remains a challenge. BNNs (Goan & Fookes, 2020) predominantly shape the landscape of methodologies that tackle epistemic uncertainties (Gawlikowski et al., 2023). Given the complexity of dealing with posterior distributions, these approaches have mostly been tailored for enhanced scalability.

For instance, Hernández-Lobato & Adams (2015) introduced an efficient probabilistic backpropagation, and Blundell et al. (2015) developed BNNs by backpropagation to learn diagonal Gaussian distributions with the reparametrization trick. Similarly, Laplace methods (MacKay, 1992) estimate the posterior distribution, thanks to an approximation of the local curvature of the loss. They often focus on the final layer (Ober & Rasmussen, 2019; Watson et al., 2021), again for scalability.

On a different approach, Monte Carlo Dropout, introduced by Gal & Ghahramani (2016) and Kingma et al. (2015), is a framework that, applied to fully-connected layers, models the posterior as a mixture of Dirac distributions. Broadening the spectrum, deep ensembles (Lakshminarayanan et al., 2017), arguably along with their more computationally efficient alternatives (Wen et al., 2019; Maddox et al., 2019; Franchi et al., 2020; 2023; Havasi et al., 2021; Laurent et al., 2023), have been interpreted by Wilson & Izmailov (2020) as Monte Carlo estimates of Equation 1.

**Markov-chain-based Bayesian posterior estimation.** Neal et al. (2011) introduced Hamiltonian Monte Carlo (HMC) – based on Monte Carlo Markov Chains (MCMC) – as an accurate method for estimating distributions, but its application to large-scale problems, such as the posterior of modern DNNs, remains challenging due to its exceptionally high computational demands.

In response to these challenges, stochastic approximations of MCMC have gained attention for their ability to provide computationally feasible solutions. A prominent example is the stochastic version of Langevin dynamics (Roberts & Tweedie, 1996) by Welling & Teh (2011). By adding noise into the dynamics, stochastic Langevin allows for more practical implementation on large datasets. In addition, other stochastic gradient-based methods have been introduced to improve the efficiency of MCMC sampling. Chen et al. (2014) presented Stochastic Gradient Hamiltonian Monte Carlo (SGHMC), and Zhang et al. (2020) designed C-SGLD and C-SGHMC (Cyclic Stochastic Gradient Langevin Dynamics), introducing controlled noise via cyclic preconditioning.

While stochastic approximation methods offer computational convenience, they come with the trade-off of slowing down the convergence and potentially introducing bias into the resulting inference (Bardenet et al., 2017; Zou & Gu, 2021). As such, the suitability of these approaches depends on the specific application and the level of acceptable bias in the analysis.

Izmailov et al. (2021) estimated the Bayesian posterior scaling full-batch HMC to CIFAR-10 thanks to 512 TPUv3. While we also estimate posteriors, we select another more scalable method supported and compared to HMC in Appendix B. Furthermore, we bring a novel focus that remained mostly uncharted: theoretically and empirically quantifying the impact of symmetries on the posterior.

**Symmetries in neural networks.** The seminal work from Hecht-Nielsen (1990) established a foundational understanding by investigating permutation symmetries and setting a lower bound on symmetries in multi-layer perceptrons. Albertini et al. (1993) extended this work and studied flip-sign symmetries in neural networks with odd activation functions. These works were further generalized to a broader range of activation functions by Kůrková & Kainen (1994), who suggested symmetry removal to streamline evolutionary algorithms.

Recent advancements have generalized symmetries to modern neural architectures. Neyshabur et al. (2015) explored the scaling symmetries that arise in architectures containing non-negative homogeneous activation functions, including Nair & Hinton (2010)'s ubiquitous Rectified Linear Unit (ReLU). This perspective extends our understanding of symmetries to ReLU-powered architectures, *e.g.*, AlexNet (Krizhevsky et al., 2012), and ResNet architectures (He et al., 2016). This paper focuses on scaling and permutation symmetries, but other works, such as Rolnick & Kording (2020); Grigsby et al. (2023), unveil less apparent symmetries. Closer to our work, Wiese et al. (2023) demonstrated that taking weight-space symmetries into account could reduce the support of the Bayesian posterior and improve MCMC posterior estimation.

# 3   SYMMETRIES INCREASE THE COMPLEXITY OF BAYESIAN POSTERIORS

We study *scales* and *permutations*, the most influential weight-space symmetries and their properties related to the Bayesian posterior. Since there is no posterior without prior, we advise the reader that we will work on *maxima a posteriori* and take the most common weight-prior amongst practitioners, the Gaussian prior on the weights, which is equivalent to L2 regularization. We detail the role of the priors in Appendix D.4. Now, let us start with a definition: weight-space symmetries transform the parameters of the neural networks while keeping the networks functionally invariant,

**Definition 3.1.** Let $f_{\boldsymbol{\omega}}$ be a neural network of parameters $\boldsymbol{\omega}$ taking $n$-dimensional vectors as inputs. We say that the transformation $\mathcal{T}$ modifying $\boldsymbol{\omega}$ is a weight-space symmetry operator, iff

$$f_{\mathcal{T}(\boldsymbol{\omega})} = f_{\boldsymbol{\omega}}, \text{ or } \forall \boldsymbol{x} \in \mathbb{R}^n, \ f_{\mathcal{T}(\boldsymbol{\omega})}(\boldsymbol{x}) = f_{\boldsymbol{\omega}}(\boldsymbol{x}). \tag{2}$$

With the notation $f_{\mathcal{T}(\boldsymbol{\omega})}(\boldsymbol{x})$, we apply the symmetry operator $\mathcal{T}$ on the weights $\boldsymbol{\omega}$, resulting in a set of modified weights. In the following, we show that scaling and permutation symmetries have different impacts on the posterior of neural networks. They can, for instance, complicate Bayesian posteriors, creating artificial functionally equivalent modes.

## 3.1   AN INTRODUCTORY EXAMPLE OF ARTIFICIAL SYMMETRY-DRIVEN POSTERIOR MODES

To illustrate the considerable impact of symmetries on the Bayesian posterior, we showcase a small-scale classification example in Figure 1. We generate this example by training two-hidden-neuron perceptrons on linearly separable data. The figure presents the estimation of a bivariate marginal of the posterior of the output weights with 10,000 independently-trained samples (left) when successively removing scaling (center) and then permutation symmetries (right). This figure shows that the scaling symmetries seem to disperse the points from the modes and that the most important mode is duplicated due to the (here, unique) permutation symmetry, which symmetrizes the graph. We detail this toy experiment in Appendix A. In the following, we develop a new mathematical framework tailored to help understand the impact of these symmetries on the posterior, devise mathematical insights explaining these intuitions, and explore more empirical dimensions.

## 3.2   BACKGROUND AND DEFINITIONS

The full extent of this formalism (including sketches of proofs, other definitions, properties, and propositions) is developed in Appendix E. Here, we summarize the minimal information to understand the impact on the Bayesian posterior of the two main symmetries – scales and permutations. This part summarizes the most important results for multi-layer perceptrons, but we provide leads for generalizing our results to modern DNNs such as convolutional residual networks in Appendix D.11.

## 3.3   SCALING SYMMETRIES

For clarity, the following definitions and properties are provided for two-layer fully connected perceptrons, without loss of generality. We first denote the line-wise and column-wise as $\bigtriangledown$ and $\triangleright$, respectively (see Definition E.2). Given that the rectified linear unit $r$ is non-negative homogeneous – *i.e.*, for all non-negative $\lambda$, $r(\lambda x) = \lambda r(x)$ – we have the following core property for scaling symmetries (Neyshabur et al., 2015), trivially extendable to additive biases:

**Property 3.1.** *For all* $\boldsymbol{\theta} \in \mathbb{R}^{\cdot \times m}$, $\boldsymbol{\omega} \in \mathbb{R}^{m \times n}$, $\boldsymbol{\lambda} \in (\mathbb{R}_{>0})^m$,

$$\forall \boldsymbol{x} \in \mathbb{R}^n, \ (\boldsymbol{\lambda}^{-1} \bigtriangledown \boldsymbol{\theta}) \times r(\boldsymbol{\lambda} \triangleright \boldsymbol{\omega}\boldsymbol{x}) = \boldsymbol{\theta} \times r(\boldsymbol{\omega}\boldsymbol{x}). \tag{3}$$

Denoting the transformation of Equation 3 by $\mathcal{T}_s$ – in the case of a two-layer perceptron – the core property directly follows, with the set of parameters $\Lambda = \{\boldsymbol{\lambda}\}$:

**Property 3.2.** *For any* usual *neural network with non-negative homogenous activations* $f_{\boldsymbol{\omega}}$, *the scaling operation* $\mathcal{T}_s$ *with a set of non-negative parameters* $\Lambda$ *is a symmetry,* i.e., $\forall \boldsymbol{x} \in \mathbb{R}^n, \ f_{\mathcal{T}_s(\boldsymbol{\omega},\Lambda)}(\boldsymbol{x}) = f_{\boldsymbol{\omega}}(\boldsymbol{x})$.

### 3.4 PERMUTATION SYMMETRIES

We also present an intuitive formalism for permutation symmetries, multiplying the weights by permutation matrices. For two-layer perceptrons, with $P_m$ the set of permutation matrices, we have that:

**Property 3.3.** *For all $\boldsymbol{\theta} \in \mathbb{R}^{\cdot \times m}$, $\boldsymbol{\omega} \in \mathbb{R}^{m \times n}$, and permutation matrices $\boldsymbol{\pi} \in P_m$,*

$$\forall \boldsymbol{x} \in \mathbb{R}^n, \ \boldsymbol{\theta}\boldsymbol{\pi}^\intercal \times r\left(\boldsymbol{\pi} \times \boldsymbol{\omega}\boldsymbol{x}\right) = \boldsymbol{\theta} \times r(\boldsymbol{\omega}\boldsymbol{x}). \tag{4}$$

The left term of Equation 4 is the definition of the permutation symmetry operator of parameter $\Pi = \{\boldsymbol{\pi}\}$ for a network including two layers. In general, we have the following property:

**Property 3.4.** *For any* usual *neural network $f_{\boldsymbol{\omega}}$, the permutation operation $\mathcal{T}_p$ with a set of parameters $\Pi$ is a symmetry, i.e., $\forall \boldsymbol{x} \in \mathbb{R}^n, \ f_{\mathcal{T}_p(\boldsymbol{\omega}, \Pi)}(\boldsymbol{x}) = f_{\boldsymbol{\omega}}(\boldsymbol{x})$.*

### 3.5 THE BAYESIAN POSTERIOR AS A MIXTURE OF DISTRIBUTIONS

With this formalism, we can establish the following proposition, a formalization and extension of Kurle et al. (2021), clarifying the impact of weight-space symmetries on the Bayesian posterior.

**Proposition 1.** *Define $f_{\boldsymbol{\omega}}$ a neural network and $f_{\tilde{\boldsymbol{\omega}}}$ its corresponding identifiable model – a network transformed for having sorted unit-normed neurons. Let us also denote $\mathbb{\Pi}$ and $\mathbb{\Lambda}$, the sets of permutation sets and scaling sets, respectively, and $\tilde{\Omega}$ the random variable of the sorted weights with unit norm. The Bayesian posterior of a neural network $f_{\boldsymbol{\omega}}$ trained with stochastic gradient descent can be expressed as a continuous mixture of a discrete mixture:*

$$p(\Omega = \boldsymbol{\omega} \mid \mathcal{D}) = \int_{\Lambda \in \mathbb{\Lambda}} |\mathbb{\Pi}|^{-1} \sum_{\Pi \in \mathbb{\Pi}} p(\tilde{\Omega} = \mathcal{T}_p(\mathcal{T}_s(\boldsymbol{\omega}, \Lambda), \Pi), \Lambda \mid \mathcal{D}) d\Lambda. \tag{5}$$

Proposition 1 provides an expression of the Bayesian posterior that highlights the redundancy of the resulting distribution, explaining the symmetry in Figure 1 (left). Interestingly, a direct corollary of this formula is that layer-wise, *all marginal inbound posteriors are identical*. This has practical consequences: in Appendix B, we show that HMC-based posterior estimation breaks this corollary. In Equation 5, the permutations play a transparent role, being independent of $\boldsymbol{\omega}$ (except for strongly quantized spaces for $\boldsymbol{\omega}$ that we leave for future works). On the other hand, the part played by scaling symmetries is more complex, and we discuss their impact in the following section.

### 3.6 ON THE EFFECTIVE IMPACT OF SCALING SYMMETRIES

While the equiprobability of permutations in Equation 5 leads to a simple balanced mixture of $|\mathbb{\Pi}|$ permuted terms, we have no such result on scaling symmetries since the standard L2-regularized loss is not invariant to scaling symmetries (and the initialization is not "uniform"). This absence of invariance obscures the impacts of scaling symmetries, which mostly remains to be addressed, although the "reduction" of their effect due to regularization was mentioned in, *e.g.*, Godfrey et al. (2022). To the best of our knowledge, we provide the first analysis of the tangibility of scaling symmetries and their impact on the Bayesian posterior. With this objective in mind, we define the following problem.

**Definition 3.2.** Let $f_{\boldsymbol{\omega}}$ be a neural network and $\bar{\boldsymbol{\omega}}$ its weights without the biases. We define the scaled network representation cost problem (or *the scaled-representation problem*) as the minimization of the L2-regularization term of $f_{\boldsymbol{\omega}}$ (the "mass") under scaling transformations. In other words,

$$m^* = \min_{\Lambda \in \mathbb{\Lambda}} \left|\mathcal{T}_s(\bar{\boldsymbol{\omega}}, \Lambda)\right|^2. \tag{6}$$

This problem – a restriction of the representation cost minimization, *e.g.* (Jacot, 2022) – has interesting properties. Notably, we show in Appendix E.5 that the *scaled-representation* problem is log-log strictly convex (Boyd & Vandenberghe, 2004; Agrawal et al., 2019):

**Proposition 2.** *The* scaled-representation *problem is log-log strictly convex: it is equivalent to a strictly convex problem on $\mathbb{R}^{|\mathbb{\Lambda}|}$ and admits a single global minimum attained at $\Lambda^*$.*

It follows from Proposition 2 that, if not already optimal at convergence, there is an infinite number of equivalent networks with training loss lower than the original network. We put the proposition into practice in Figure 2 using trained OptuNets (see Appendix C.2.1): we measure their mass distribution and compare them to the masses of the optimal networks found with convex optimization. The

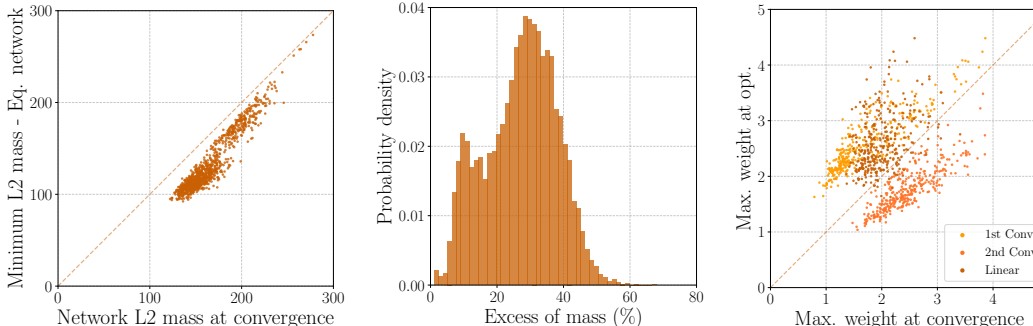

Figure 2: **OptuNets trained with weight decay never converge to the *minimum scaled representation*.** We also note that the maxima of the weights of scaled OptuNets at the minimum scaled representation – referred to as "at opt." in (right) – tend to be greater for layers with fewer parameters than in the original networks.

effect of scaling symmetries remains even with weight decay: neural networks seem to persist in being subject to scaling symmetries as shown in Figure 2 (left). Figure 2 (center) depicts that the ratios of the mass at convergence on the minimum representation are non-negligible, the converged networks being consistently heavier. Figure 2 (right) displays the values of the largest elements of each layer: the minimization of the mass tends to increase the heaviest weights of the layers with fewer parameters (here, the convolutional layers) but does not seem to promote unfeasible values. Finally, we provide a loss landscape interpretation of this property in Appendix D.5 and explain the inability of DNNs to converge to the *minimum mass* by its corresponding gradient being lower than SGD noise. We generalize this result to ResNet-18 on CIFAR-100 in Appendix D.3 and Figure 11. In this case, the networks at minimal representation costs have extremely low mass weights due to the sequences of convolution and batch normalization layers (see Section D.11.3). Until now, we provided theoretical insights on the contrasted impacts of both scaling and permutation symmetries. In the following, we develop more empirical studies and explore the link between posterior quality and performance.

## 4 COMPARING POSTERIOR ESTIMATIONS BY APPROX. BAYESIAN METHODS

In this section, we leverage symmetries to compare popular single-mode methods, namely, Monte Carlo Dropout (Gal & Ghahramani, 2016), Stochastic Weight Averaging Gaussian (SWAG) by Maddox et al. (2019), variational inference BNNs (viBNNs) (Blundell et al., 2015), and Laplace methods (Ritter et al., 2018). We also include their multi-modal variations, corresponding to the application of these methods on ten different independently trained models, as well as SGHMC (Chen et al., 2014), preconditioned SGLD (Li et al., 2016) and deep ensembles (DE) highlighted by Hansen & Salamon (1990) and Lakshminarayanan et al. (2017). We compare these methods on three image classification tasks with different levels of difficulty, ranging from MNIST (LeCun et al., 1998) with OptuNet (392 parameters) to CIFAR-100 (Krizhevsky, 2009) and Tiny-ImageNet (Deng et al., 2009) with ResNet-18 (He et al., 2016). To this extent, we leverage maximum mean discrepancies (MMD) to estimate the dissimilarities between the high-dimensional posterior distributions. We estimate the *target* posterior using 1000 independently trained neural networks and compare it to 100 samples from all previously mentioned techniques. This choice – compared to the more theoretically-grounded HMC (Neal et al., 2011; Izmailov et al., 2021) – is supported by theoretical aspects on the estimation of high-dimensional distributions (Wild et al., 2023), by computational constraints that would shatter *mini-batched* HMC's guarantees in practice, and by experiments showing that full-batch HMC's theoretical performance may not be fully achieved in our real-world settings. However, we stress that sampling from independently trained checkpoints to estimate the posterior remains imperfect and can be debated. We discuss these limitations in-depth in Appendix B.

### 4.1 EVALUATING THE QUALITY OF THE ESTIMATION OF THE BAYESIAN POSTERIOR

One approach to assess the similarities between distributions involves estimating the distributions and subsequently quantifying the distance between these estimated distributions (Smola et al.,

Table 1: **Comparison of popular methods approximating the Bayesian posterior.** All scores are expressed in %, except the ACEs and the MMDs for ResNet-18 networks, expressed in ‰. Acc stands for accuracy, and **ID**MI and **OOD**MI are in-distribution and out-of-distribution mutual information. NS is the MMD computed after removing the symmetries, and DE stands for Deep Ensembles. Multi-mode methods are based on ten independently trained models.

| | | Method | MMD↓ | NS↓ | Acc↑ | ECE↓ | ACE↓ | Brier↓ | AUPR↑ | FPR95↓ | **ID**MI↓ | **OOD**MI↑ |
|---|---|---|---|---|---|---|---|---|---|---|---|---|
| MNIST - OptuNet | One Mode | Dropout | 15.0 | 14.3 | 83.3 | 26.1 | 60.0 | 33.4 | 96.4 | 98.6 | 26.1 | 22.2 |
| | | viBNN | 18.8 | 17.1 | 78.1 | 7.4 | 17.6 | 30.9 | 67.9 | 93.7 | **0.1** | 0.1 |
| | | SWAG | 16.0 | 14.6 | 88.3 | 4.9 | 11.9 | 17.7 | 73.4 | 68.6 | 4.0 | 8.7 |
| | | Laplace | 10.6 | 9.5 | 87.9 | 4.8 | 15.1 | 18.1 | 48.2 | 74.6 | 6.2 | 5.9 |
| | | SGHMC | 16.7 | 17.7 | 95.1 | **2.8** | **3.2** | **7.6** | 73.7 | 98.4 | 4.3 | 14.5 |
| | | pSGLD | 15.1 | 17.3 | 88.1 | 3.8 | 9.1 | 17.7 | 49.2 | 75.5 | 1.0 | 0.9 |
| | Multi Mode | Dropout | 2.1 | 2.1 | 92.1 | 36.8 | 67.5 | 29.2 | **97.2** | 78.2 | 36.6 | 52.5 |
| | | viBNN | 2.8 | 2.5 | 86.5 | 17.5 | 31.3 | 24.4 | 96.9 | 27.2 | 21.1 | 52.3 |
| | | SWAG | 1.8 | 1.3 | 95.0 | 17.5 | 27.6 | 13.1 | 88.7 | 24.6 | 27.6 | 62.2 |
| | | Laplace | 1.8 | 0.8 | 94.8 | 15.8 | 24.5 | **12.8** | 95.4 | 32.1 | 21.1 | 52.2 |
| | | DE | **0.0** | **0.0** | **95.3** | 10.9 | 21.0 | 13.5 | 95.7 | **12.8** | 19.3 | **62.6** |
| CIFAR100 - ResNet18 | One Mode | Dropout | 4.5 | 7.5 | 74.2 | 14.7 | 3.2 | 38.8 | 76.4 | 47.7 | 5.7 | 9.1 |
| | | viBNN | 9.0 | 10.2 | 57.9 | 24.6 | 3.0 | 63.7 | 60.9 | 79.1 | 2.7 | 4.2 |
| | | SWAG | 6.7 | 7.2 | 70.9 | 2.3 | 1.2 | 38.9 | 86.2 | 48.0 | 2.4 | 6.3 |
| | | Laplace | 5.7 | 7.0 | 75.1 | **0.9** | 0.9 | 34.6 | 81.3 | 42.4 | 27.6 | 63.3 |
| | | SGHMC | 7.5 | 7.9 | 73.7 | 4.9 | 1.0 | 36.2 | 79.4 | 62.3 | **0.2** | 0.5 |
| | Multi Mode | Dropout | 0.7 | 4.5 | **79.5** | 4.3 | 1.0 | 29.2 | 78.2 | 48.1 | 20.5 | 46.3 |
| | | viBNN | 6.1 | 5.6 | 66.5 | 2.8 | 2.0 | 45.3 | 71.9 | 71.7 | 45.5 | 81.1 |
| | | SWAG | 5.0 | 5.4 | 72.8 | **1.5** | 1.1 | 36.9 | **89.1** | 50.6 | 6.5 | 19.7 |
| | | Laplace | 0.6 | 4.3 | 78.9 | 6.9 | 0.8 | 30.3 | 82.9 | **41.3** | 44.1 | **98.5** |
| | | DE | **0.0** | **0.0** | 79.5 | 1.6 | **0.6** | **28.7** | 81.1 | 45.6 | 22.5 | 58.0 |
| TinyImageNet - ResNet18 | One Mode | Dropout | 9.5 | 4.9 | 63.2 | 16.4 | 2.4 | 53.9 | 48.8 | 81.1 | 8.3 | 8.4 |
| | | viBNN | / | / | / | / | / | / | / | / | / | / |
| | | SWAG | 9.1 | 3.9 | 66.4 | 10.5 | 0.7 | 46.2 | 61.9 | 57.7 | 3.0 | 4.5 |
| | | Laplace | 5.5 | 6.1 | 33.1 | 6.0 | 3.6 | 77.1 | 48.8 | 77.7 | 200.7 | 228.0 |
| | | SGHMC | 9.8 | 5.3 | 58.3 | **2.6** | 1.0 | 54.1 | 56.3 | 72.7 | **0.24** | 0.30 |
| | Multi Mode | Dropout | 4.3 | 1.8 | 70.2 | 9.9 | 1.2 | 42.1 | 74.8 | 58.2 | 34.1 | 60.0 |
| | | viBNN | / | / | / | / | / | / | / | / | / | / |
| | | SWAG | 6.7 | 5.4 | 69.3 | 3.6 | **0.6** | 41.3 | **96.5** | 55.9 | 17.6 | 32.1 |
| | | Laplace | 0.5 | 3.1 | 37.0 | 10.9 | 3.3 | 75.1 | 48.4 | 72.5 | 219.5 | **254.7** |
| | | DE | **0.0** | **0.0** | **70.3** | 6.5 | 0.7 | **40.9** | 86.3 | **50.2** | 38.4 | 83.4 |

2007; Sriperumbudur et al., 2010). However, these methods can become impractical when dealing with distributions in extremely high-dimensional spaces, such as the posterior of modern DNNs. An alternative solution is to embed the probability measures into reproducing kernel Hilbert spaces (RKHS) (Bergmann, 1922; Schwartz, 1964). Within this framework, a distance metric, the maximum mean discrepancy (MMD) (Song, 2008) - defined as the distance between the respective mean elements within the RKHS - is used to quantify the dissimilarity between the distributions.

Appendix C.4 formalizes and explains our implementation of the MMD reported in Table 1: we follow Schrab et al. (2023) and use multiple Gaussian and Laplace kernels. For tractability, we report the mean – weighted by the number of parameters of each layer – of the median over twenty MMD kernels between the layer-wise DE-based posterior estimation and the approximation provided by each method. The NS metric corresponds to the MMD computed after *a posteriori*-symmetry removal using the algorithms detailed in Appendix D.2. Appendix C gathers all details concerning these experiments (including the means and maxima over the kernels of the MMDs).

## 4.2 PERFORMANCE METRICS AND OOD DATASETS

On top of the MMD quantifying the difference between the posterior estimations, we measure several empirical performance metrics. We evaluate the overall performance of the models using the accuracy and the Brier score (Brier, 1950; Gneiting et al., 2007). Furthermore, we choose the binned expected calibration error (ECE) (Naeini et al., 2015) and adaptive calibration error (ACE) (Nixon et al., 2019) for top-label calibration and measure the quality of the out-of-distribution (OOD) detection using the area under the precision-recall curve (AUPR) and the false positive rate at $95\%$ recall (FPR95), as recommended by Hendrycks & Gimpel (2017). We expect the OOD detection abilities

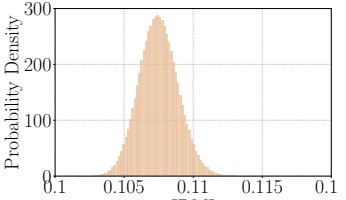 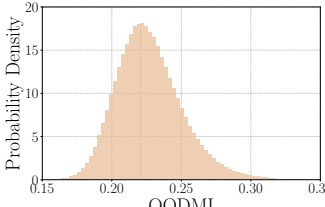 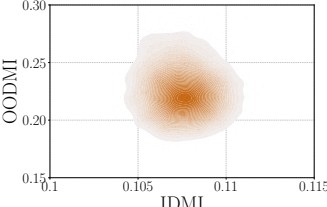

Figure 3: **Experiments show no hint of a functional collapse between couples of independently trained ResNet-18 on CIFAR-100.** Moreover, the in-distribution and out-of-distribution mutual information (IDMI, resp. OODMI) exhibit different variances but do not seem correlated.

of the models to correlate with the quality of the estimated posterior. Finally, we report the mean diversity of the predictions in each ensemble through the mutual information (MI) (*e.g.*, Ash (1965)), often used to measure epistemic uncertainty (Kendall & Gal, 2017; Michelmore et al., 2018). We use FashionMNIST (Xiao et al., 2017), SVHN (Netzer et al., 2011), and Textures (Cimpoi et al., 2014) as OOD datasets for MNIST, CIFAR-100, and TinyImageNet, respectively.

### 4.3 RESULTS

Table 1 demonstrates that multi-mode techniques consistently exhibit superior performance in terms of MMD when compared to their single-mode counterparts. This trend holds true in accuracy, negative log-likelihood, and calibration (ECE and ACE) for ResNet architectures. We provide further details on the calibration performance for OptuNets and Laplace methods in Appendix C.3.

Turning our attention to the assessment of epistemic uncertainty, as quantified by AUPR and FPR95, multi-mode techniques, notably multi-SWAG and DE, consistently outperform other methods. This underscores the strong connection between posterior estimation and the accuracy of epistemic uncertainty quantification. However, we note that the quality of aleatoric uncertainty quantification does not steadily correlate with that of the posterior distribution estimation.

The final two columns of the table shed light on the diversity of the models sampled from the posterior. The objective is to minimize in-distribution mutual information (IDMI) while maximizing out-of-distribution mutual information (OODMI). An analysis shows that mono-mode methods yield lower values than multi-mode methods, suggesting inferior *diversity* for the former.

## 5 DISCUSSIONS

We develop further insights on the posterior of Bayesian neural networks in relationship with symmetries. Notably, we evaluate the risk of *functional collapse*, *i.e.*, of training very similar networks in Section 5.1, and discuss the frequency of weight permutations in 5.2. We showcase "Checkpoints" our dataset in Section D.1. In Appendix B, we discuss using independent checkpoints for the posterior estimation. Appendix D expands these discussions, discusses the impact of the chosen prior in D.4, links the posterior to recent works on the loss landscapes (Section D.5), adds observations on the number of modes of the posterior in D.8, and provides visualizations (Section D.10).

### 5.1 FUNCTIONAL COLLAPSE IN ENSEMBLES: A STUDY OF ID AND OOD DISAGREEMENTS

Given the high number of equivalent modes due to permutation symmetries (see Section 3.5), we support broadening the concept of collapse in the parameter-space – *e.g.*, in D'Angelo & Fortuin (2021) – to *functional collapse* to account for the impact of symmetries on the posterior. Parameter-space collapse is more restrictive and may not be formally involved when ensemble members lack diversity. It is also harder to characterize as it would require an analysis of the loss landscape, at least.

We quantify functional collapse as a potential ground for the need for more complex repulsive ensembling methods (Masegosa, 2020; Rame & Cord, 2021). We take 1000 ResNet-18 trained to estimate the Bayesian posterior in Section 4 and compute the mean over the test set of their pairwise mutual information (see Section D.12), quantifying the divergence between the single models and their

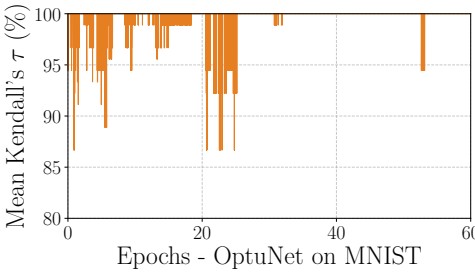 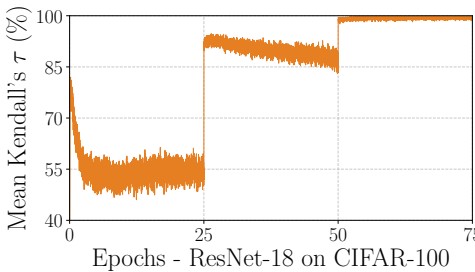

Figure 4: **Evolution during training of the mean Kendall's $\tau$ correlation between the permutations towards the identifiable model for all successive steps**: The correlation between successive permutations increases when the learning rate decreases.

average. We measure these values on in-distribution and OOD data with CIFAR-100 and SVHN, respectively. In Figure 3 (left), we see that the in-distribution MI between any two networks has a very low variance. Given that the usual diversity between models is satisfactory (Lakshminarayanan et al., 2017), there is an extremely low probability of training two similar networks, despite the huge number of potential symmetric models. These results hint that the complexity of the posterior is orders of magnitude greater than what we grasp with symmetries: large DNNs seem to empirically never fall into these numerous symmetric local minima. This may be explained by the high complexity of the network (here, a ResNet-18), and we refer to Appendix D.9 for results on a smaller architecture. Interestingly, we note in Figure 3 (right) that in contrast to intuition, we have no significant correlation between the in-distribution and the OOD MI. This highlights that measuring the in-distribution *diversity* may be a very poor indicator of the OOD detection performance of a model.

### 5.2 FREQUENCY OF WEIGHT PERMUTATIONS DURING TRAINING

We devise a new protocol to evaluate if a network tends to permute during training. Given a DNN $f_{\boldsymbol{\omega}}$, we compute, for each step $s$ of the training, the permutation set $\Pi_s$ sorting its weights (and removing the corresponding symmetries). If the DNN tends to permute during the training, this implies a variation in the $\Pi_s$. We measure the extent of the variations using Kendall's $\tau$ correlation between successive permutations $\Pi_s$ and $\Pi_{s+1}$. We plot the variation of the mean over several training instances, and the Kendall's $\tau$ of each element of the permutation sets in Figure 4. We see that on MNIST (left), the variations of the permutation sets are scarce and gathered around points of instability. These instabilities are due to the sorting mechanism, which is based on the maximum values of the neurons' weights; we have tried other statistics on the values of the weights, but taking the maximum remains the most stable. Moreover, the weights nearly never permute in the last training phase. The analysis differs for ResNet-18 (right) since the number of degrees of freedom is much greater. We see a lot of variation during the phases with a high learning rate (reduced after 25 and 50 epochs). However, as for the first case, we do not see any particular sign of permutations in the last part of the training. This evidence is in favor of weight-averaging methods such as SWAG (Maddox et al., 2019), which, therefore, have only very limited risks of averaging symmetric networks.

## 6 CONCLUSION

In this study, we have examined Bayesian neural network posteriors, which are pivotal for understanding uncertainty. Our findings suggest that part of the complexity in these posteriors can be attributed to the non-identifiability of modern neural networks viewing the posterior as a mixture of permuted distributions. To explore this further, we introduce the *scaled representation* problem and investigate the real impact of scaling symmetries. Using real-world applications, we design a method to assess the quality of the posterior distribution and study its correlation with model performance, particularly regarding uncertainty quantification. While considering symmetries has provided valuable insights, our discussions hint at a more profound complexity going beyond these weight-space symmetries. In future work, we plan to continue our exploration of this intriguing area.

# 7 ACKNOWLEDGEMENTS

This work was performed using HPC resources from GENCI-IDRIS (Grant 2023-[AD011011970R3]).

# 8 REPRODUCIBILITY STATEMENT

To ensure transparency and accessibility, we use publicly available datasets, including MNIST, FashionMNIST, CIFAR100, SVHN, ImageNet-200, and Textures. Please refer to Appendix C.2.2 for details on these datasets. Our detailed experimental methods are outlined in Appendices A and C, and the proofs for our theoretical results are provided in Appendix E.

To help replicate our work, we share the source code of our experiments on GitHub, notably including code to remove symmetries from neural networks *a posteriori*. For our experiments in Section 4, we rely exclusively on open-source libraries. Most of our experiments are performed with TorchUncertainty, including the training of the standard and dropout models, but also the evaluation of their Deep Ensembles versions. For the rest, we use the GitHub repository Bayesian-Neural-Networks for SGHMC, BLiTZ for variational Bayesian neural networks, and Laplace Redux (Daxberger et al., 2021) for Laplace evaluations. We also use the publicly available code from the original paper (Maddox et al., 2019) for the SWAG method.

Finally, we estimate the maximum mean discrepancies with a homemade torch version of the code from Schrab et al. (2023) and solve our convex optimization problems (see Definition E.6) with cvxpy (Diamond & Boyd, 2016). The statistical experiments, such as Pearson's $\rho$ and Kendall's $\tau$, are performed with SciPy.

# 9 ETHICS

Our primary goal in this paper is to improve our comprehension of the Bayesian posterior, which we argue is a fundamental element to understand to contribute to the reliability of machine-learning methods.

We note that training a substantial number of checkpoints for estimating the posterior, especially in the case of the thousand models trained on TinyImageNet, was energy intensive (around 3 Nvidia V100 hours per training). We opted for the Jean-Zay supercomputer, a carbon-efficient cluster to mitigate the environmental impact of our research.

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

### TABLE OF CONTENTS – SUPPLEMENTARY MATERIAL

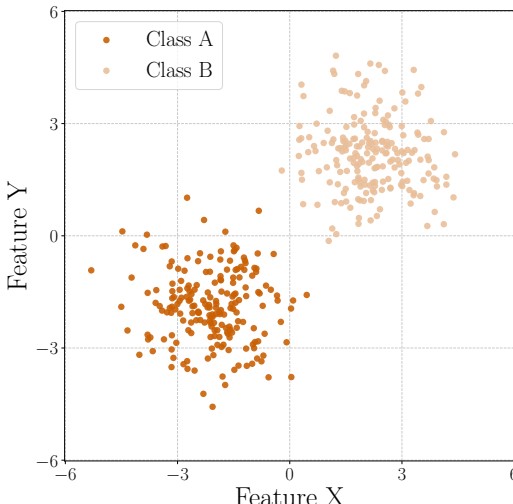

Figure 5: **The training data of the models whose posterior is represented in the introduction, on Figure 1.** The data is separable to ensure efficient training of simple 9-parameter perceptrons.

## A   DETAILS ON THE STARTING EXAMPLE

### A.1   TRAINING THE TWO-LAYER PERCEPTRONS

To create the introductory example – in Figure 1 – we generate the data corresponding to the two classes from two different normal distributions of means $\boldsymbol{\mu}_A = (-2, -2)$ and $\boldsymbol{\mu}_B = (2, 2)$ and of identity covariances. We ensure that the samples (200 points each) are fully separable to make training as simple as possible. We plot the training data in Figure 5. We then train 10,000 two-layer perceptrons with two input, two hidden, and one output neurons with early stopping for ten epochs with SGD and a binary cross-entropy loss. We separate the first and second layers with a ReLU activation function for the posterior to exhibit scaling symmetries. We use a batch size of ten data points and a learning rate of two. Finally, we select networks with a sufficiently low loss on the training set, removing the few outliers (less than 1%) that had not been trained successfully.

### A.2   REMOVING SYMMETRIES

Figure 1 shows the density of weights of the last layer (excluding the last bias). The first Figure shows the unaltered projection of the posterior on these weights, which is then transformed to guarantee a three-norm per neuron. We chose three as the norm for visual purposes since the last layer (whose posterior is plotted in Figure 1) is not directly normalized but rather subject to the normalization of the previous layer. Finally, we remove the permutation symmetries by ordering the weights. Contrary to the formalism developed in the following sections, we decide here to order the weights starting from the last layer (not the first) to convey the message more efficiently. We check with random inputs for each network that the symmetry-removal operations do not alter the networks. Notebooks are available in the supplementary material of the ICLR submission and detail the process for generating the Figure. We provide more detail on the symmetry removal algorithms in Section D.2.

## B  ON THE ESTIMATION OF THE POSTERIOR WITH INDEPENDENTLY TRAINED CHECKPOINTS

Here, we detail the different methods for estimating Bayesian posteriors of DNNs. We provide a small literature review of the strengths and weaknesses of each method, briefly describe the computational complexities, and develop an in-depth study on the comparison of the posterior estimation thanks to the HMC checkpoints gracefully released by Izmailov et al. (2021). Notably, we highlight elements of comparison between HMC and deep ensembles-like generated checkpoints for extremely high dimensional posterior estimation.

### B.1  BIBLIOGRAPHIC AND THEORETICAL ASPECTS

Various methods have been suggested for estimating the Bayesian posterior of DNNs, each with its own strengths and limitations. Noteworthy approaches include variational inference BNNs (Blundell et al., 2015), MCMC (Neal et al., 2011), Laplace methods (MacKay, 1992), and ensembles (Lakshminarayanan et al., 2017; Wilson & Izmailov, 2020). However, all these methods face challenges and criticisms, especially in the context of posterior estimation for high-dimensional modern computer vision networks – the focal point of this paper. Here are key concerns for each method:

**Variational inference methods.** Variational inference BNNs are notoriously unstable and difficult to scale to large networks (Dusenberry et al., 2020). Furthermore, they often provide a simplistic approximation of the posteriors as diagonal Gaussian distributions (Blundell et al., 2015). Moreover, most of these are mono-mode approaches, leading to even coarser posterior estimations.

**Monte-Carlo Markov Chains.** MCMC methods, including HMC (Neal et al., 2011) – sometimes referred to as gold standard –, are reported as some of the best algorithms to estimate distributions, although extremely computationally expensive when applied on deep neural networks (Izmailov et al., 2021). MCMC's efficiency is reported to be highly dependent on their hyperparameters (*e.g.*, this course) which may need tuning, incurring supplemental costs. Moreover, MCMCs are also known for their difficulties in modeling multi-modal distributions (Mangoubi et al., 2018; Izmailov et al., 2021; Park, 2021). This could be problematic, given that the target distributions are very multi-modal (see Wilson & Izmailov (2020) for the multi-modality). However, Izmailov et al. (2021) have shown that these algorithms also demonstrated good properties in practice when applied to estimating the posterior of DNNs. Finally, their faster stochastic approximations like Stochastic Gradient Hamiltonian Monte Carlo have been reported to be sometimes less reliable (Bardenet et al., 2017).

**Laplace methods.** Laplace (MacKay, 1992) methods model the posterior around a single maximum a posteriori. Moreover, they often approximate the posterior around the weights of the last layer. Similarly to variational inference methods, these approaches are constrained in their ability to estimate posterior distributions with single modes.

**Ensembles.** While efficiently handling the posterior's multimodality, theoretical considerations suggest they require repulsive terms for full Bayesian consistency (D'Angelo & Fortuin, 2021). Indeed, ensembles tend to favor sampling local minima and avoid their surroundings. Ensembles also face criticism when used with a limited number of inner estimators.

In Section 4, we suggest estimating the Bayesian posterior using an unrivaled number of independently trained checkpoints. This approach is chosen to address the multimodality inherent in DNN posteriors. A recent work (Wild et al., 2023) demonstrates that – in a problem with a small number of local minima – the robustness of ensembles is theoretically less convincing than that of regularized methods like Langevin ensembles, although still better than variational inference. However, the other methods struggle to compete with ensembles in practice, even with small UCI-regression datasets. We emphasize that the datasets and architectures in this study are complex and extremely high-dimensional, exhibiting a theoretically infinite number of "local minima". We argue – supported by Wild et al. (2023)'s conclusion – that prioritizing local minima over their surroundings when the space cannot be sufficiently sampled may not be clearly detrimental to the estimation. The mechanisms to enhance algorithm convergence (D'Angelo & Fortuin, 2021; Wild

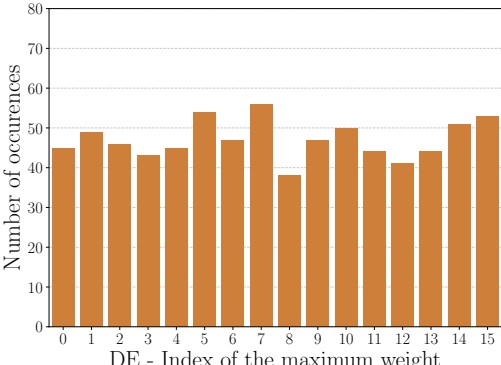 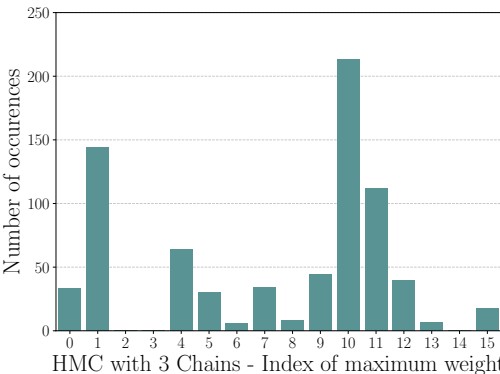

Figure 6: **The number of occurrences of the indices of the maximum weights of each first hidden convolutional layer amongst 750 samples of ResNet-20s estimated via both DE and HMC;** left: the estimation based on independently trained networks leads to a smooth uniform distribution of the indices of the maxima; right: the distribution of the maxima of the samples from HMC is sharp, despite using three independent chains and discarding the first 40 checkpoints of each chain for best fairness. Please note that the two figures have different scales.

et al., 2023) in the sense of Wild et al. (2023) seem more relevant when the number of local minima is small, a condition not met in the real-world applications that we consider in this work.

## B.2 COMPUTATIONAL ASPECTS

Our choice of deep ensemble-style checkpoints for estimating the posterior is also guided by computational constraints. While HMCs are appealing from a theoretical point of view – and even arguably from a practical point of view in some contexts given the inspirational work of (Izmailov et al., 2021) – its computational cost was out of reach for this study. Indeed, Izmailov et al. (2021) benefit from 512 TPUv3 and claim that their study was extremely compute-expensive. Training HMC indeed requires putting the full batch in memory, which may not be feasible for real-world datasets. Additionally, conducting a hyperparameter search for HMC can be time-consuming and resource-intensive. We confidently estimate the cost of their training to be two orders of magnitude greater than ours, and the VRAM requirements are simply out of reach. All of this is without counting the ablation studies, and for a moderate architecture, ResNet-20, which has around 275,000 parameters, 40 times less than the ResNet-18 that we trained on TinyImageNet, whose more numerous images are also four times bigger than CIFAR10 (in number of pixels).

To summarize, we can say that independently-trained checkpoints and HMC scale equivalently in terms of hard disk storage – assuming that we get the same number of checkpoints – yet exhibit a much more scalable behavior compared to HMC concerning the VRAM used. Consequently, DE may be more computationally suitable for deeper and wider architectures and larger datasets.

## B.3 COMPARING ESTIMATIONS FROM HMCS AND INDEPENDENTLY TRAINED CHECKPOINTS

We show that despite its numerous qualities, the estimation of the distribution by HMC raises questions and that even in its best case with independent chains and discarding multiple burn-in samples.

### B.3.1 IS HMC-BASED POSTERIOR ESTIMATION PERMUTATION INVARIANT?

The permutation aspect of Proposition 1 is intuitive and boils down to the non-identifiability of the inner neurons of any neural network. As such, this property is basic and holds in the general setting studied in this work (and beyond since it extends to other activation functions). However, we show that HMC (Neal et al., 2011), breaks this proposition, most likely due to remains of dependence between successive samples. To show this phenomenon, we use the checkpoints provided by Izmailov et al. (2021), namely three full-batch HMC chains of ResNet-20s trained on CIFAR-10, and compare them to the same number of networks that we train and release in our dataset (see Section D.1).

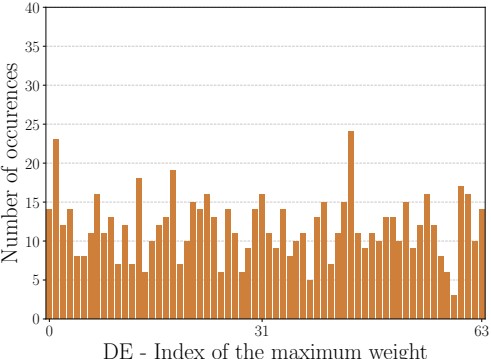 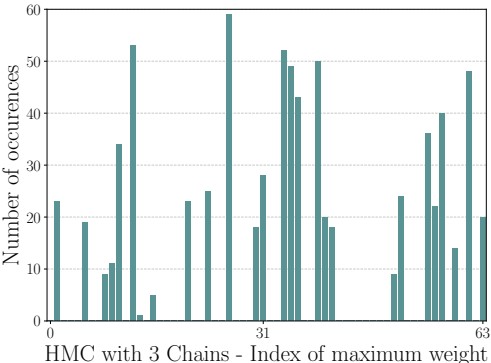

Figure 7: **The number of occurrences of the indices of the maximum weights of each final linear layer amongst 750 samples of ResNet-20s estimated via both deep ensembles and HMC;** The interpretation remains the same as for Figure 6 although DE's p-value is lower than 5%.

More specifically, we show that the HMC-based estimation does not respect Proposition 1's corollary. To recall, this corollary states that given a layer, the marginals of the posterior are exactly similar. Let us consider a simple statistic, such as the index of the channel of the maximum weight. More precisely, we start by averaging the $3 \times 3$ kernels of the convolutional layer to recover two-dimensional tensors to recover a setting similar to the linear layer. Corollary 1 implies that the number of occurrences of the indices should follow a uniform categorical distribution. Intuitively, there is no reason for any non-identifiable neuron to contain the maximum weight more often than the others. Figure 6 (left) shows that an ensemble of independently trained samples seems qualitatively close to this result. However, it seems evident in Figure 6 (right) that it is not the case for the HMC-based samples, even in the most favorable setting: using an ensemble of independent chains and discarding numerous burn-in samples.

We strengthen this empirical evidence with another experiment in Figure 7, which shows the equivalent distribution for the final layer of the residual network. This layer has more input parameters and, therefore, more possible indices, but the sentence remains the same. The observation data contradicts the expectations set forth by Proposition 1 that should theoretically hold in the considered setting. As such, the HMC-estimated posterior is inconsistent with the theory. We limit ourselves to these two layers as they have the smallest number of indices, but release the code that generates these figures.

To be more quantitative, we perform a $\chi^2$ test to compare the distributions of the indices of the maximum weights to the uniform categorical distribution. For deep ensembles, we obtain 94.1% and 2.3% chance of mistake when rejecting that the two distributions are equivalent, for the convolutional and linear layers, respectively. On the other hand, we obtain negligible p-values for HMC (of the order of $10^{-300}$) in both cases. Please note that we use here a simplistic statistic, and a sample compatible with a uniform categorical distribution would be the bare minimum to respect Proposition 1.

We can argue that this could be a favorable point for HMC. Indeed, the previous point suggests that HMC is likely not to waste samples on symmetric modes. However, we recall that independently trained checkpoints do not suffer from functional collapse either (see Section 5.1).

### B.3.2 COMPARING THE ESTIMATIONS OF THE POSTERIOR BY APPROXIMATE BAYESIAN METHODS

**Experimental setup.** We test the performance of our ResNet-20 (He et al., 2016) trained on CIFAR-10 (Krizhevsky, 2009) with the SiLU activation (Ramachandran et al., 2017) and filter response normalization (Singh & Krishnan, 2020), and compare them to the checkpoints provided by Izmailov et al. (2021). To this end, we report the MMD and the NS, the MMD without permutation symmetries (there are no exact-scaling symmetries with SiLU activations) using both independently-trained checkpoints as target distribution (MMD / D, NS / D) and HMC (MMD / H, NS / H).

Table 2: **Comparison of popular methods approximating the Bayesian posterior.** All scores are expressed in %, except for the ACEs, expressed in ‰. Acc stands for accuracy, and **ID**MI and **OOD**MI are in-distribution and out-of-distribution mutual information. NS is the MMD computed after removing the symmetries, and DE stands for Deep Ensembles. MMD / D and NS / D are the MMD and NS computed with a target distribution based on DE and MMD / H and NS / H are based on HMC. Multi-mode methods are based on ten independently trained models except for HMC which is based on three independent chains.

| | Method | MMD / D | MMD / H | NS / D | NS / H | Acc ↑ | ECE ↓ | ACE ↓ | Brier ↓ | AUPR ↑ | FPR95 ↓ | **ID**MI ↓ | **OOD**MI ↑ |
|---|---|---|---|---|---|---|---|---|---|---|---|---|---|
| **One Mode** | Dropout | 5.7 | 5.5 | 5.7 | 5.4 | 89.9 | 2.0 | 7.3 | 14.7 | 64.9 | 81.6 | 7.8 | 5.0 |
| | BNN | 8.4 | 6.7 | 8.5 | 6.6 | 88.0 | 20.0 | 10.3 | 19.9 | 78.1 | 38.6 | **0.1** | 0.3 |
| | SWAG | 6.0 | 5.7 | 3.9 | 4.5 | 89.7 | **0.8** | 5.5 | 14.7 | 82.7 | 34.2 | 2.0 | 7.5 |
| | Laplace | 5.5 | 4.9 | 6.6 | 5.4 | 90.4 | 5.5 | 7.1 | 15.1 | 79.8 | 42.8 | 1.1 | 4.5 |
| **Multi Mode** | Dropout | 0.7 | 2.7 | 0.7 | 2.6 | 93.7 | 2.3 | 2.7 | 9.7 | 91.0 | 26.4 | 11.5 | **70.7** |
| | BNN | 2.8 | 2.8 | 2.9 | 2.7 | 92.7 | 2.4 | 3.3 | 11.3 | 83.2 | 31.7 | 15.6 | 58.2 |
| | SWAG | 1.1 | 2.7 | 0.9 | 2.5 | 92.0 | 3.3 | 3.3 | 11.9 | 83.3 | 31.4 | 7.1 | 24.2 |
| | Laplace | 0.5 | 3.6 | 2.2 | 2.6 | 92.7 | 2.0 | **2.5** | 9.4 | 90.5 | 24.1 | 12.0 | 61.2 |
| | HMC | 2.3 | **0.0** | 2.2 | **0.0** | 90.3 | 5.7 | 8.1 | 15.0 | 90.0 | 23.7 | 29.7 | 90.5 |
| | DE | **0.0** | 2.3 | **0.0** | 2.2 | **94.3** | 2.3 | 3.1 | **8.7** | **92.0** | **20.1** | 13.3 | 67.1 |

**Results.** Table 2 shows that deep ensembles consistently outperform HMC in all metrics except the out-of-distribution diversity. These results confirm Izmailov et al. (2021)'s claim that HMC can find very diverse samples in the function space. This may be related to HMC's hyperparameter $\Delta$ (the step size) that controls the trade-off between the likelihood and the acceptance rate of the samples. Indeed, the accuracy and the Brier score of HMC are below what we could expect from an ensemble of checkpoints and lie between the monomodal estimations made by SWAG and Laplace.

**On posterior discrepancies.** First, Table 2 shows that posterior discrepancy metrics are improved for multi-mode estimations when compared to mono-mode estimations. Interestingly, the MMDs estimated using HMC as reference seem slightly lower and more uniform than with deep ensembles'. Moreover, the MMDs based on DE may correlate slightly more with the OOD detection quality since it ranks the multi-modal MC-Dropout, the second best in AUPR, with a lower NS value than its counterparts. On the other hand, the distance between HMC and the other multi-mode methods is uniform despite differences in OOD detection quality. However, we acknowledge that these correlations are not very clear, and we study another metric in Section C.5.

### B.3.3 VISUALIZATION OF THE MARGINALS

We plot the histograms of the marginal posteriors as estimated by the ensemble of HMC and the ensemble of independent checkpoints in Figure 8. This provides a small qualitative analysis of the estimations of the posterior.

These plots tend to confirm that the posterior estimated by the ensemble is more complex and more multi-modal than the estimation provided by the ensemble of HMC chains. We keep the same number of samples for the ensembles and HMC for the fairness of the comparison despite the greater number of checkpoints available with the ensemble. We also provide another interpretation of these plots:Izmailov et al. (2021) indicate that HMC successfully moves on iso-loss paths that link different modes in a given basin. We could hypothesize that HMC over-samples the low-loss regions along this path, leading to more mono-modal distributions with some amount of parameter correlation. We leave checking this hypothesis for future works.

## C DETAILS ON THE EXPERIMENTS OF SECTION 4

### C.1 EXPERIMENTAL DETAILS

In this section, we develop the training recipes of the different models and the parameters used for all the posterior estimation methods. We train our models with PyTorch (Paszke et al., 2019) on V100 clusters. We refrain from performing any model selection to avoid biasing our posterior with the validation set: *we always keep the final checkpoint*.

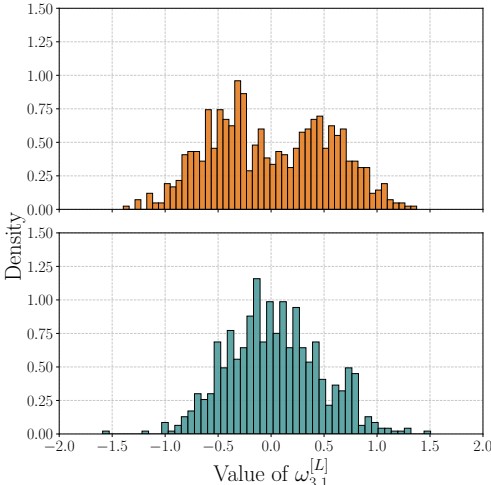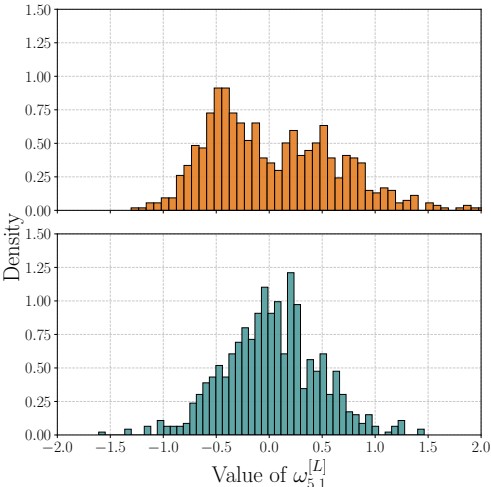

Figure 8: **Estimations by independently trained samples (orange) and 3-chain HMC (blue) of the posterior for two *non-cherry-picked* weights of the linear layer of ResNet-20.** These qualitative results tend to confirm that the posterior estimated by the independently trained samples is "more multi-modal". As specified in the indices and explained in Corollary 1, these two figures correspond to two non-identifiable neurons, explaining their similarity.

For all the *variational inference* Bayesian Neural Networks (viBNN) of Blundell et al. (2015), we use the default priors from Bayesian layers in torch zoo (Blitz).

**OptuNet – MNIST.** We train OptuNet for 60 epochs with batches of size 64 using stochastic gradient descent (SGD) with a start learning rate of 0.04 and a weight decay of $2 \times 10^{-4}$. We decay the learning rate twice during training, at epochs 15 and 30, dividing the learning rate by 2.

We train the viBNNs with the same number of epochs, albeit using 3 Monte Carlo estimates of the Evidence lower bound (ELBO) at each step and using a Kullback-Leibler divergence (KLD) of $10^{-5}$. As usually done, we disable weight decay for the training of viBNNs.

For OptuNet, we use a full Hessian Laplace optimization. We use a dropout rate of 0.2 on the last layer (both for training and testing) for the dropout version and perform SWAG using 20 different models (also set as the maximum number to keep a low-rank matrix), continuing the training with the classic high-learning rate schedule (starting with a linear increase) for twice as long. We collect the models 20 times with 10-epoch intervals between each of them. We sample the models with a scale of 0.1.

Concerning the pSGLD Bayesian method, we train the neural network for 220 epochs with a starting learning rate of $4 \times 10^{-3}$ decayed linearly during 20 epochs of burn-in towards $4 \times 10^{-10}$. We use a diagonal bias of $10^{-5}$ with a precondition decay rate of 0.99. The size of the batches remains 64, and the optimizer is a pSGLD version of RMSProp. We select a sample for the posterior and the predictions every two epochs.

We normalize the dataset as usual and perform a random crop of size 28 and padding four during training. We use the normalized test images for testing.

For more details on the architecture of OptuNet, the reader may refer to Section C.2.1.

**ResNet-20 – CIFAR-10.** We train our modified Resnet-20 with Filter Response Normalization (Singh & Krishnan, 2020) and SiLU activation (Ramachandran et al., 2017) for 200 epochs with batches of size 128, SGD with a starting learning rate of 0.1 decayed twice by 10 at epochs 100 and 150, a momentum of 0.9 and a weight decay of $1 \times 10^{-4}$. We perform the classic normalization and random crop with a four-pixel padding and a random horizontal split.

ViBNNs are also trained for 200 epochs with the same learning rate schedule albeit with a base learning rate of 0.05. We weight the KLD with a coefficient $2 \times 10^{-9}$.

We use last-layer Kronecker factorization (LLKFAC) to fit the Laplace methods (Laplace, 1774; Ritter et al., 2018). We manually average the predictions of the Monte Carlo estimation of the Laplace posterior with 100 samples. We apply dropout on the last layer only with a Bernoulli coefficient of $p = 0.5$ for the dropout models. As for MNIST, we train SWAG models twice the time of the original training recipe with the usual settings (start learning rate with a linear ramp) and collect 20 models with 10-epoch intervals. We sample the models with a scale of 0.1.

When training ResNet-20 and ResNet-18 models, we perform last-layer approximations of Laplace and Dropout. For more information on last-layer approximation, the reader may refer to Brosse et al. (2020).

**ResNet-18 – CIFAR-100.** We train the ResNet-18 for 75 epochs with batches of size 128 using SGD with Nesterov (Nesterov, 1983; Sutskever et al., 2013), with a base learning rate of 0.1, a momentum of 0.9, and a weight decay of $5 \times 10^{-4}$. Similarly to MNIST, we decay the learning rate twice during training, here at epochs 25 and 50, by a factor of 10.

We train the variational BNNs with SGD for 150 epochs, with a starting learning rate of 0.01, which we decay once after 80 epochs. We weight the KLD with a coefficient $2 \times 10^{-9}$ and perform three ELBO samples at each step.

As for ResNet-20, we use last-layer Kronecker factorization (LLKFAC) to fit the Laplace methods: the last layer of ResNet-18 is too large for its full Hessian to fit in memory. Moreover, Kronecker is not available for the whole network as it is not implemented for batch normalization layers, and the low-rank version is too long to train. We manually average the predictions of the Monte Carlo estimation of the Laplace posterior with 100 samples. For SWAG and dropout, we use the same parameters as for MNIST and CIFAR-10.

As for CIFAR-10, we perform the classic normalization and random crop with a four-pixel padding and a random horizontal split.

**ResNet-18 – TinyImageNet.** We train the ResNet-18 for 200 epochs with a batch size of 128 using SGD with a start learning rate of 0.2, a weight decay of $10^{-4}$, and a momentum of 0.9. We use a cosine annealing scheduler until the end of the training.

We did not manage to scale viBNNs to TinyImageNet. The other methods use precisely the same hyperparameters and methods as for ResNet-18 on CIFAR-100.

For TinyImageNet, we perform the same preprocessing as for CIFAR-100, except that we keep the original resolution of $64 \times 64$ pixels and use the proper dataset statistics.

## C.2   DETAILED ON THE ARCHITECTURES AND DATASETS

### C.2.1   DETAILS ON OPTUNET

For OptuNet, we aimed to create the smallest number of parameters possible while keeping decent performance on MNIST. To this extent, we performed an architecture search with tree-structured Parzen estimators in Optuna (Akiba et al., 2019). To reduce the number of parameters, OptuNet uses grouped convolutions, introduced by Krizhevsky et al. (2012) – see *e.g.* Laurent et al. (2023) for their formal definition – in its second convolutional layer.

### C.2.2   DETAILS ON THE DATASETS

This part provides some details on the different datasets used throughout the paper.

### C.2.3   IN-DISTRIBUTION DATASETS

In this work, we used the following datasets for training and testing.

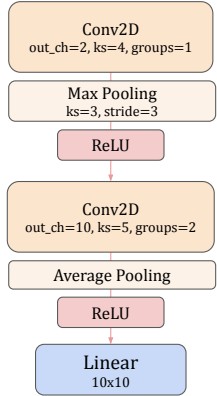

Figure 9: **Detailed architecture of OptuNet**. OptuNet includes only 392 parameters yet achieves between 85% and 90% accuracy on MNIST. Only the last fully connected layer contains biases; out_ch is the number of output channels, and ks is the kernel size.

**MNIST.** The MNIST dataset LeCun et al. (1998) comprises 70,000 binary images of handwritten digits, each of size 28×28 pixels. These images are divided into a training set with 60,000 samples and a testing set with 10,000 samples.

**CIFAR-100.** CIFAR-100 Krizhevsky (2009) consists of 100 object classes and contains a training set with 50,000 images and a testing set including 10,000 images. Each of the images is RGB and of size 32×32 pixels.

**Tiny-ImageNet.** TinyImageNet is a subset of ImageNet-1k (Deng et al., 2009). It consists of 200 object classes, each with 500 training and 50 validation images. Additionally, there are 50 test images per class for evaluating models. The images are RGB and have a size of 64×64 pixels.

### C.2.4    OUT-OF-DISTRIBUTION DATASETS

The objective of the following out-of-distribution datasets is to evaluate the quality of the posterior as a means to quantify the epistemic uncertainty. This work does not consider these datasets fully representative of complex real-world out-of-distribution detection tasks.

**Fashion-MNIST.** The Fashion-MNIST dataset (Xiao et al., 2017) is a set of 28×28 pixel-grayscale images consisting of 60,000 training and 10,000 test images. We kept the test set as is for the out-of-distribution tasks with MNIST.

**SVHN.** The Street View House Numbers (SVHN) dataset (Netzer et al., 2011) is a large-scale dataset of 600,000 images of digits obtained from house numbers in Google Street View. In our work, we kept a fixed set of 10,000 images that we cropped to squares of 32×32 pixels.

**Textures.** The Describable Textures Dataset (Cimpoi et al., 2014) is a dataset containing 5640 images of textures divided into three subsets. Considering that the number of images is limited compared to the other testing sets, we use the concatenation of all the subsets for OOD detection. We resize all the images to 64×64 pixels to stick to the size of TinyImageNet.

### C.3    ON THE TOP-LABEL CALIBRATION RESULTS

Considering the criticism received by the ECE (Naeini et al., 2015) – for instance in Nixon et al. (2019) – we also report the ACE in Table 1. The ACE (Nixon et al., 2019) is based on adaptive binning and avoids putting too much weight on bins including very small numbers of samples.

It may be surprising to the reader that Table 1 depicts single-mode methods based on OptuNet as having better top-label calibration than ensembles for OptuNet. We recall that OptuNets are

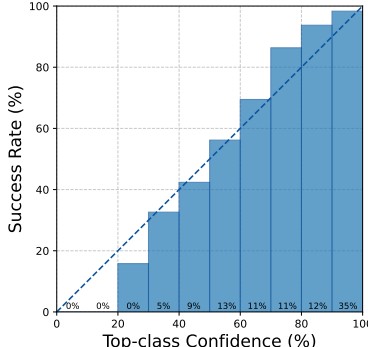 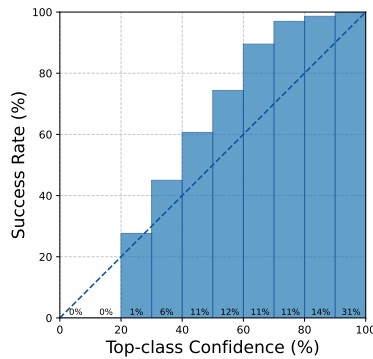

Figure 10: **Calibration plots of OptuNet-based variational inference BNN (left) and Multi-viBNN (right) trained on MNIST.** Both models output underconfident predictions that worsen in the multi-mode setting: averaging underconfident predictions is often more underconfident.

very shallow and do not exhibit overconfidence (Guo et al., 2017). Quite the contrary, we provide two calibration plots on Figure 10 that show the underconfidence of the variational inference BNNs (Blundell et al., 2015), which worsens for Multi-BNN, its corresponding multi-mode method. Note that we expect the confidence to decrease when resorting to ensembles since we provide an average of multiple predictions that may involve an amount of disagreement.

Regarding ResNet-18, only Laplace sees its calibration worsen. A hypothesis explaining this phenomenon is that Laplace optimizes its calibration post-hoc and that averaging optimized solutions may not lead to a calibrated ensemble but rather to an *under-confident* meta-estimator.

Finally, we recall that comparing calibrations of mono and multi-mode methods is not completely fair, given that there is a non-negligible difference in accuracy between the different methods.

## C.4 COMPUTING MAXIMUM MEAN DISCREPANCIES

We formalize the computation of MMDs as used in our paper following (Schrab et al., 2023). For simplicity, we keep their formalism unchanged. In our work, we use kernel-based Maximum Mean Discrepancy (Gretton et al., 2012), that given a Reproducing Kernel Hilbert Space (RKHS) $\mathcal{H}_k$ of kernel $k$, is computed as:

$$\text{MMD}\left(p, q; \mathcal{H}_k\right) := \sup_{f \in \mathcal{H}_k : \|f\|_{\mathcal{H}_k} \leq 1} \left|\mathbb{E}_{X \sim p}[f(X)] - \mathbb{E}_{Y \sim q}[f(Y)]\right|. \tag{7}$$

We use the estimator $\widehat{\text{MMD}}$ of the kernel-based MMD developed by Gretton et al. (2012) and formerly implemented in Matlab, now ported to JAX by Schrab et al. (2023) and PyTorch by the authors of this document. We replace $p$ by the most precise posterior approximated by independently trained checkpoints (see Appendix B) or HMC (in Appendix B) and $q$ by the estimation provided by the other methods. Specifically, we leverage 20 different kernels: 10 Gaussian kernels supplemented by 10 Laplace kernels, each with different parameters, and report statistics on the obtained values. Denoting $\mathfrak{H}$ the set of the kernel-based RKHS, $p$ the *supposedly* most precise posterior estimation by DE or HMC, and $q$ the coarser posterior estimation, we report in Table 4,

the median: $\quad \text{median}(\text{MMD}_{\mathfrak{H}}(p, q)) = \text{median}\left\{\widehat{\text{MMD}}\left(p, q; \mathcal{H}_k\right)\right\}_{\mathcal{H}_k \in \mathfrak{H}}$,

the maximum: $\quad \max(\text{MMD}_{\mathfrak{H}}(p, q)) = \max\left\{\widehat{\text{MMD}}\left(p, q; \mathcal{H}_k\right)\right\}_{\mathcal{H}_k \in \mathfrak{H}}$, and $\tag{8}$

the mean: $\quad \overline{\text{MMD}}_{\mathfrak{H}}(p, q) = \sum_{\mathcal{H}_k \in \mathfrak{H}} \widehat{\text{MMD}}\left(p, q; \mathcal{H}_k\right).$

For computational scalability, $p$ and $q$ are not the full posterior estimations but rather $p_l$ and $q_l$, the layer-wise marginals. We report their weighted mean, with $n_l$ the number of parameters of layer $l$:

$$\text{median}(\text{MMD}_{\mathfrak{H}}^L(p, q)) = \sum_{l=1}^{L} n_l \,\text{median}(\text{MMD}_{\mathfrak{H}}(p_l, q_l)) \tag{9}$$

Table 3: **Comparison of the Wasserstein distances on the marginals.** All scores are expressed in %, except the ACEs expressed in ‰and the WDs are kept unchanged. WD is the Wasserstein distance and WNS is the Wasserstein distance after the removal of symmetries and M-Mode stands for Multi-Mode.

| | | Method | MMD ↓ | NS ↓ | WD ↓ | WNS ↓ | Acc ↑ | ACE ↓ | Brier ↓ | AUPR ↑ | FPR95 ↓ | **OOD**MI ↑ |
|---|---|---|---|---|---|---|---|---|---|---|---|---|
| MNIST - OptuNet | One Mode | BNN | 18.8 | 17.1 | 2.3 | 2.0 | 78.1 | 17.6 | 30.9 | 67.9 | 93.7 | 0.1 |
| | | SWAG | 16.0 | 14.6 | 1.3 | 1.2 | 88.3 | 11.9 | 17.7 | 73.4 | 68.6 | 8.7 |
| | | Laplace | 10.6 | 9.5 | 0.3 | 0.6 | 87.9 | 15.1 | 18.1 | 48.2 | 74.6 | 5.9 |
| | | SGHMC | 16.7 | 17.7 | 0.3 | 2.0 | 95.1 | **3.2** | **7.6** | 73.7 | 98.4 | 14.5 |
| | M-Mode | BNN | 2.8 | 2.5 | 0.3 | 0.4 | 86.5 | 17.5 | 24.4 | 96.9 | 27.2 | 52.3 |
| | | SWAG | 1.8 | 1.3 | 0.1 | 0.1 | 95.0 | 27.6 | 13.1 | 88.7 | 24.6 | 62.2 |
| | | Laplace | 1.8 | 0.8 | **0.0** | 0.1 | 94.8 | 24.5 | 12.8 | 95.4 | 32.1 | 52.2 |
| | | DE | **0.0** | **0.0** | **0.0** | **0.0** | **95.3** | 21.0 | 13.5 | 95.7 | **12.8** | **62.6** |

Contrary to Schrab et al. (2023), we do not perform two-sample tests, which power we would try to maximize. Hence, we prefer using a robust statistic, the median, to the maximum and report it as MMD in Table 1. The other metric, NS, corresponds to the MMD applied on $p$ and $q$ after removing symmetries using the algorithms detailed in Appendix D.2.

## C.5 USING A WASSERSTEIN DISTANCE ON THE MARGINALS

Given that the correlations between the MMDs of the posterior estimations and the OOD detection performance are not completely satisfying, we test another distance, the one-dimensional Wasserstein distance (WD), on the binned marginals. Although, by definition, somewhat limited, the Wasserstein distance may also be more interpretable. We provide the results on MNIST with Optunet in Table 3 along with the most important metrics that we recall from Table 1.

Table 3 show that this distance favors, just as the MMD, multi-modal distribution estimations versus mono-modal ones. Furthermore, there is a good correlation between the Wasserstein distances and the MMDs in relatively small dimensions. Regarding the removal of symmetries, the WDs of SGHMC change drastically as opposed to the corresponding MMDs that remain constant. However, just as for the MMDs, WDs are unable to predict the good performance in OOD detection of the BNNs and do not discriminate the multi-modal Laplace from DE and Multi-SWAG despite its lower out-of-distribution functional diversity.

## C.6 FULL MMD TABLE

In Table 4, we report the weighted average over the layers of the medians, means, and maxima of the Maximum Mean Discrepancies of each layer, computed in Section 4. We recall that the maximum is used to obtain the best discriminative of two-sample tests, for instance, in Gretton et al. (2012); Schrab et al. (2023). We report the median in the main table to improve the representativity of the values. However, we note that the order of the values does not seem to change, particularly between the means, medians, and maxima.

We see that, after Deep Ensembles – that have perfect MMD as expected – Laplace methods perform best on all measures, be it with or without symmetries, as well as for all types of architectures. However, as shown in Table 1, they are inferior to SWAG on performance metrics. This hints that the correlation between the estimated quality of the posterior and the real-world metrics may not be fully correlated. Please note that for Deep Ensembles, we used different checkpoints for the second sample than in the sample corresponding to the posterior estimation.

Table 4: **Comparison of popular methods approximating the Bayesian posterior.** Maximum Mean Discrepancies for OptuNet are expressed in %. For ResNet-18 networks, MMDs are expressed in ‰. NS MMDs are the MMDs computed after the removal of the symmetries. Multi-mode methods use ten independently trained models. DE stands for Deep Ensembles (Lakshminarayanan et al., 2017) and TinyIN for TinyImageNet.

| | | Method | MMD | | | NS MMD | | |
|---|---|---|---|---|---|---|---|---|
| | | | Median | Mean | Max. | Median | Mean | Max. |
| **MNIST - OptuNet** | **One Mode** | Dropout | 14.9 | 14.6 | 22.9 | 14.3 | 14.3 | 22.5 |
| | | BNN | 18.8 | 18.3 | 29.5 | 17.1 | 17.0 | 27.4 |
| | | SWAG | 15.9 | 15.8 | 25.9 | 14.6 | 13.8 | 21.6 |
| | | Laplace | 10.6 | 10.5 | 19.1 | 9.5 | 9.4 | 16.4 |
| | | SGHMC | 16.7 | 16.6 | 27.8 | 17.7 | 18.3 | 32.2 |
| | | pSGLD | 14.9 | 14.8 | 25.1 | 17.2 | 17.8 | 30.4 |
| | **Multi Mode** | Dropout | 2.1 | 2.3 | 4.1 | 2.1 | 2.0 | 3.5 |
| | | BNN | 2.8 | 2.9 | 5.8 | 2.5 | 2.4 | 4.3 |
| | | SWAG | 1.8 | 2.0 | 4.1 | 1.3 | 1.3 | 2.5 |
| | | Laplace | 1.8 | 2.0 | 4.1 | 0.8 | 0.8 | 1.5 |
| | | DE | 0.0 | 0.0 | 0.0 | 0.0 | 0.0 | 0.0 |
| **CIFAR-100 - ResNet-18** | **One Mode** | Dropout | 4.5 | 4.6 | 7.2 | 7.4 | 7.5 | 12.0 |
| | | BNN | 9.0 | 9.2 | 14.6 | 10.0 | 10.2 | 16.0 |
| | | SWAG | 6.7 | 7.0 | 11.2 | 7.2 | 7.5 | 12.3 |
| | | Laplace | 5.7 | 6.0 | 9.7 | 7.0 | 7.2 | 11.7 |
| | | SGHMC | 7.5 | 7.3 | 11.0 | 7.9 | 8.0 | 13.0 |
| | **Multi Mode** | Dropout | 0.7 | 0.7 | 1.2 | 4.4 | 4.5 | 7.5 |
| | | BNN | 6.1 | 6.3 | 10.6 | 5.6 | 5.8 | 10.0 |
| | | SWAG | 5.0 | 5.2 | 8.4 | 5.4 | 5.6 | 9.2 |
| | | Laplace | 0.6 | 0.6 | 1.0 | 4.3 | 4.4 | 7.2 |
| | | DE | 0.0 | 0.0 | 0.0 | 0.0 | 0.0 | 0.0 |
| **TinyIN - ResNet-18** | **One Mode** | Dropout | 9.5 | 9.6 | 15.0 | 4.9 | 5.0 | 7.8 |
| | | BNN | / | / | / | / | / | / |
| | | SWAG | 9.1 | 9.5 | 15.6 | 5.4 | 5.6 | 9.5 |
| | | Laplace | 5.5 | 5.8 | 9.6 | 6.1 | 6.3 | 10.1 |
| | | SGHMC | 9.8 | 10.0 | 15.7 | 5.3 | 5.2 | 7.9 |
| | **Multi Mode** | Dropout | 4.3 | 4.4 | 7.4 | 1.8 | 1.8 | 3.0 |
| | | BNN | / | / | / | / | / | / |
| | | SWAG | 6.7 | 6.9 | 11.3 | 3.9 | 4.2 | 7.3 |
| | | Laplace | 0.6 | 0.6 | 9.9 | 3.1 | 3.2 | 5.3 |
| | | DE | 0.0 | 0.0 | 0.0 | 0.0 | 0.0 | 0.0 |

# D    OTHER DISCUSSIONS AND DETAILS

## D.1    THE CHECKPOINTS DATASET

We release "Checkpoints", our extensive dataset of deep neural network checkpoints on multiple tasks and architectures on Hugging Face. Specifically, this dataset contains:

- 100,000 15-neuron two-layer perceptrons trained on a regression task on the energy efficiency dataset (see Section C.2.2),
- 10,000 small time-series forecasting LSTMs (Hochreiter & Schmidhuber, 1997), trained on the Exchange-rate dataset,
- 10,000 OptuNets (see Section C.2.1) trained on MNIST (LeCun et al., 1998),
- 1,024 ResNet-20 (He et al., 2016) with a SiLU activation (Ramachandran et al., 2017) and filter response normalization (Singh & Krishnan, 2020) trained on CIFAR-10 (Krizhevsky, 2009)
- 2,048 ResNet-18 (He et al., 2016) trained on CIFAR-10
- 9,216 ResNet-18 trained on CIFAR-100 (Krizhevsky, 2009), and
- 2,048 ResNet-18 trained on ImageNet-200 (Deng et al., 2009).

## D.2    ALGORITHMS FOR REMOVING SYMMETRIES A POSTERIORI

**Removing permutation symmetries.**    Similar to Pourzanjani et al. (2017), we remove permutation symmetries by ordering neurons according to the value of their first corresponding parameter. In dense layers, this first parameter is the weight of the first input neuron, and in convolutional layers, it corresponds to the top-left weight of the kernel of the first channel. This solution is more general than that of Pourzanjani et al. (2017) since neural network layers do not always have biases. Specifically, practitioners often remove biases from a layer when followed by a batch normalization (Ioffe & Szegedy, 2015) to reduce the number of inference and backpropagation steps computations.

**Removing scaling symmetries.**    In this paper, we use two different algorithms to remove the scaling symmetries of a neural network *a posteriori*. Neyshabur et al. (2015) designed the first algorithm to remove these symmetries by normalizing the norm of the weights of each neuron in all the layers except the last. In most of the paper, including Table 1, we use a simple *a posteriori* normalization of the weights. We also implement this algorithm to scale the standard deviation of the weights to one.

Furthermore, we define in Section 3.6 a restriction of the representation cost problem that enables the deletion of scaling symmetries. We show the existence and uniqueness of the solution to this problem and provide a code to obtain it using a convex optimization solver. It takes around 10 minutes to find the solution for a ResNet-18, in which case the dimension of the scaling symmetries (and therefore the number of variables in the convex optimization problem) is equal to 7680.

While scalable to current architectures, the convex optimization is slower than the normalization of the weights, and its behavior may be more challenging to grasp. Hence, we stick to the normalization of the weights when many computations are needed.

**Removing softmax additive symmetries.**    To remove this type of symmetry, it is sufficient to scale the sum of the biases of the last layer to some constant, for instance, to 1. This dimension is not studied in this paper.

## D.3    MINIMUM SCALED REPRESENTATION COSTS OF RESNET-18

The minimum scaled representation costs of Resnet-18 (He et al., 2016) are very different from those obtained with OptuNet in Section 3.6. A key changing element between the two architectures is the use of batch normalization layers. As stated in Section D.11.3, we do not use the inverse of the outbound scaling coefficients when updating the batch normalization layers. Instead, we just update the mean and the variance. This means that the cost of using extremely small outbound coefficients is minimal: it only reduces the variance of the output. This translates in practice to very

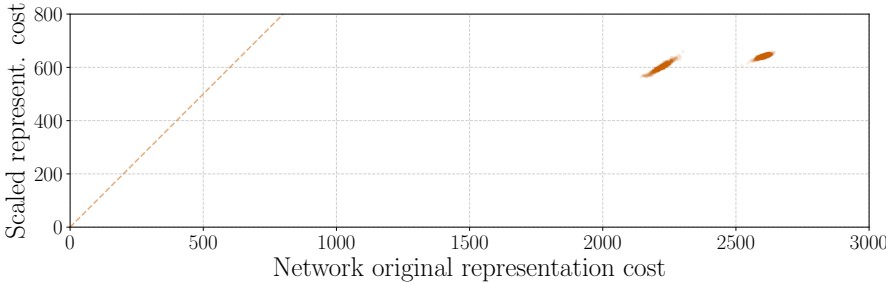

Figure 11: **The distribution of the minimal scaled representation cost ("mass") vs. the representation cost at convergence.** The minimum mass is much lower compared to OptuNets in Section 3.6 due to the batch normalization layers.

low minimum representation costs, as seen in Figure 11. Further inspection shows that the mass is concentrated in the last layers of the networks.

### D.4 ON PRIORS AND THEIR IMPACT ON THE POSTERIOR

In this section, we detail the priors used in our theory and experiments and provide leads on their impact on our results and analyses.

#### D.4.1 THE BAYESIAN INTERPRETATION OF WEIGHT DECAY

Practitioners recognize the crucial role of weight regularization when employed with cross-entropy, as it introduces a bias in optimizing neurons toward networks characterized by lower representation cost or "mass". This bias promotes the emergence of potentially simpler and more generalizable functions within deep learning architectures (Goodfellow et al., 2016).

Additionally, weight regularization has a Bayesian interpretation. It encourages the network to prefer simpler models when faced with multiple plausible explanations for the observed data. This is akin to Ockham's razor: given some data, the simplest explanation is the most likely, and there is no need to add unnecessary hypotheses. Indeed, L2 weight regularization corresponds exactly to setting a centered Gaussian prior on the networks' weights.

More formally, in classification and given a sample $(\boldsymbol{x}, y_i)$ from the joint distribution $\mathcal{P}_{X,Y}$, we try to fit a model of parameters $\boldsymbol{\omega}$ to predict $p(y_i \mid \boldsymbol{x}_i, \boldsymbol{\omega})$. This can be done by minimizing the likelihood of the predicted distribution: this typically corresponds to the cross-entropy loss, which practitioners often minimize with stochastic gradient descent to obtain maximum likelihood estimates:

$$\mathcal{L}_{\text{MLE}}(\boldsymbol{\omega}) = - \sum_{(\boldsymbol{x}_i, y_i) \in \mathcal{D}} \log p(y_i \mid \boldsymbol{x}_i, \boldsymbol{\omega}). \tag{10}$$

Going further, the Bayesian framework enables the incorporation of prior knowledge regarding $\boldsymbol{\omega}$ denoted as the distribution $p(\boldsymbol{\omega})$ that complements the likelihood. It leads to the research of the maximum a posteriori (MAP) via the minimization of the following loss function:

$$\mathcal{L}_{\text{MAP}}(\boldsymbol{\omega}) = - \sum_{(\boldsymbol{x}_i, y_i) \in \mathcal{D}} \log p(y_i \mid \boldsymbol{x}_i, \boldsymbol{\omega}) - \log(\boldsymbol{\omega}). \tag{11}$$

The *i.i.d.* normal prior is the standard choice for $p(\boldsymbol{\omega})$, leading to the omnipresent L2 weight regularization (or weight decay). This paper aims to tackle real-world applications and, therefore, follows the practitioners in this regularization, *de facto* assuming Gaussian priors on our weights.

#### D.4.2 THE IMPACT OF THE PRIORS ON SYMMETRIES & THE POSTERIOR

Most priors do not impact the results linked to permutations. Indeed, priors are mostly constant for the entire network and, therefore, constant within the layers. This means that the neurons remain non-identifiable and that permutations continue to occur in the posterior.

However, priors do have an important impact on scaling symmetries as they are closely related to the *minimum scaled representation cost*. Indeed, increasing the value of the variance of the Gaussian prior will most likely change the outcome of the training and lead to networks that are further from their corresponding network of minimal representation cost. More specifically, in Equation 5, we can expect the posterior associated to $\lambda$ that are further from the minimum to decrease faster. We intend to study this relationship in future works.

Changing the prior distribution could also completely change the analysis of scaling symmetries. Indeed, choosing a Laplace prior, available on most deep-learning frameworks, would make the *scaled representation cost* problem non-convex and likely hinder any hope of easily computable solutions. However, this type of priors is *very* rare in the community.

Finally, priors are key to understanding the complexity of the relationship between the loss landscape and the posterior, but we leave this point to the next section.

### D.5   ON THE LINKS BETWEEN POSTERIORS AND LOSS LANDSCAPES

The impact of symmetries on the loss landscape has been studied in several works (Li et al., 2018; Fort et al., 2019; Fort & Jastrzebski, 2019; Liu et al., 2022). However, their effect on the posterior is much less chartered. In this discussion, we argue that this is due to the inherent difficulty of this problem. Indeed, contrary to the loss landscape, it is not possible to estimate quickly the density of a Bayesian posterior in some discretized bin of the space. Consequently, measuring the impact of the weight-space symmetries on the posterior is difficult. We argue that a large part of this difficulty comes from the fact that the posterior is determined both by the loss landscape and the prior in the weight space, which hinders direct links between the first two.

#### D.5.1   (ALSO) A PROBLEM OF PRIORS

Let us take a classic standard Gaussian distribution as prior. We know, thanks to scaling symmetries (see Section 3.1), that we can find a non-regularized-loss-invariant line (for simplicity) between any local minimum and its corresponding scaled networks with arbitrary representation cost. In doing so, we move on a flat surface and keep the network functional unchanged. However, we get a completely different story when considering the prior. Indeed, the prior will be (linearly) increasing with the representation cost of the scaled networks leading to a linearly decreasing Bayesian posterior density. This small example hints that the interaction between the loss landscape and the prior contributes to the complexity of the posterior.

However, please note that the initialization in not scaling invariant. This implies that even without prior, one wouldn't get "equiprobable" scaling symmetries. This would be impossible anyway as the uniform distribution cannot be defined on continuous unbounded spaces.

#### D.5.2   A LOSS-LANDSCAPE POINT OF VIEW ON THE SCALED-REPRESENTATION PROBLEM

Property 2 establishes the strict convexity of the *minimum scaled representation cost* minimization after change of variables. In this section, we study the impact of this property on the loss landscape.

Given that the *minimum scaled representation cost* is log-log strictly convex, we know that there exists a single global minimum of the problem, *i.e.*, a global minimizer of the regularization term on the scaling symmetries. This means that if we take some neural network $f_\omega$ at the end of its training, there exists a linear path of decreasing regularization loss towards the global minimum. Moreover – by definition of the symmetries (see Definition 3.1), and in particular the scaling symmetries (Definition 3.1) – we know that moving in the continuous space of permutation symmetries keeps the function unchanged. Hence, it also keeps the negative-log-likelihood constant. Combining these two results, we deduce that there exists a linear path of decreasing loss towards the minimum of the scaled-representation problem. We show the loss profiles for 200 trained OptuNets on this path in Figure 12 (left).

This raises a question: if a path of decreasing loss exists towards a region of lower loss, why can't the neural network minimize it? We argue that this may be due to the regularization-induced gradient's negligibility compared to the stochastic gradient descent oscillations. Indeed, Figure 12 (right) shows that the gradient and the direction towards the minimum of the *scaled-representation* mini-

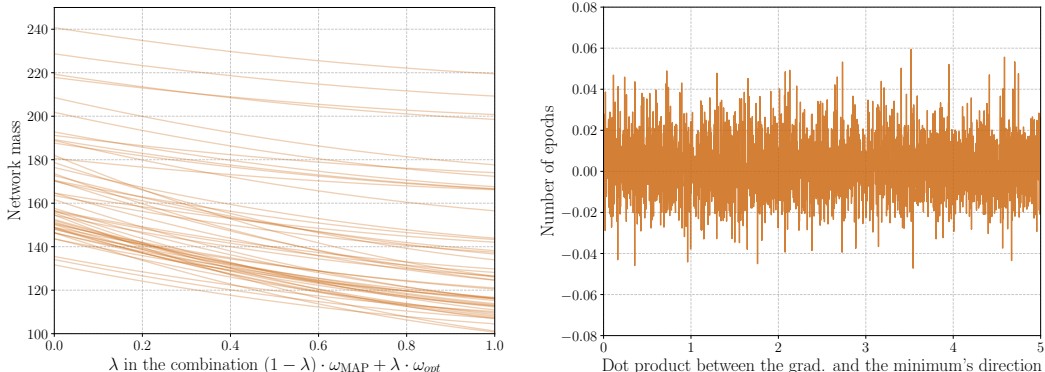

Figure 12: **Left: Evolution of the mass of OptuNets on the path between the maximum *a posteriori* $\omega_{\mathbf{MAP}}$ and the global minimum of the minimum *scaled representation* problem $\omega_{opt}$.** At first glance, the regularization term seems smooth and easy to optimize. **Right: Evolution of the scalar product between the gradient and the unit vector to the minimum.** The mean of the dot-product is 0.002, compared to the mean of the full gradient's norm of around 1.302.

mization do not align much. Their dot-product is only very small compared to the norm of the gradients. We obtain these results post-training an OptuNet for 5 epochs with a weight decay of $10^{-4}$.

### D.6 DOES SOLVING THE SCALED-REPRESENTATION PROBLEM IMPROVE YOUR NETWORK?

As written in Section D.4, it seems clear that Gaussian priors (L2 regularization) are beneficial for generalization. Therefore, it is legitimate to wonder whether the solution of the *scaled-representation* problem improves over the starting network.

We recall that solving the scaled-representation problem is done independently from the likelihood maximization and on the space of the scaling symmetries. As such, by definition of the symmetries (see Definition 3.1), the network will not move in the function space: the function will not be altered. Consequently, there will be no impact on the evaluation performance, whether in-distribution or out-of-distribution.

Nevertheless, this may impact post-training, such as fine-tuning or weight-averaging (Maddox et al., 2019), but we leave this for future works.

### D.7 PUTTING PROPOSITION 1 IN PRACTICE

We develop a small toy example to put Proposition 1 in practice, check our results in small dimensions, and show the impact of symmetries on the estimated posterior.

To this extent, we train 100,000 2-layer MLPs on a small circle classification dataset. We plot the data in Figure 13. This small dataset forces the MLPs to benefit from non-linearities allowed by their activation functions. We suggest using two-layer perceptions with two input neurons, three hidden neurons, and one output neuron. In total, the MLPs have 13 parameters. To enable exact scaling symmetries, we use ReLU activations and train for 300 epochs with a batch size of 8, a constant learning rate of 0.03, and a Gaussian prior corresponding to an L2-regularization of 0.003.

Figure 14 shows the projection of 10,000 trained models after dimensionality reduction by principal component analysis (PCA) (left) and T-distributed stochastic neighbor embedding (TNSE) (right). It confirms the multi-modality of the posterior distribution even for such a simple model, with 13 parameters, on a very basic dataset.

### D.8 ESTIMATING THE NUMBER OF PERMUTATION-GENERATED MODES

For simplicity, suppose that the layer-marginal posterior of the identifiable model is an $n$-dimensional Dirac distribution of coordinates $\boldsymbol{\theta}$ (or a distribution on a small bounded domain around

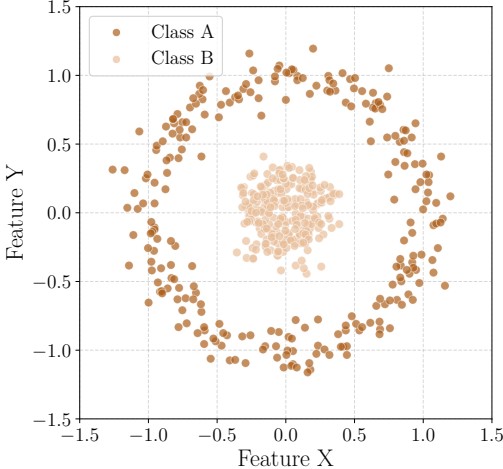

Figure 13: **The two-circles classification training dataset used as the basis to train toy models.**
We train the small 2-layer perceptrons to separate and classify the samples from A and B. The mean
accuracy after 300 epochs is around 98% on the test set sampled from the same distribution.

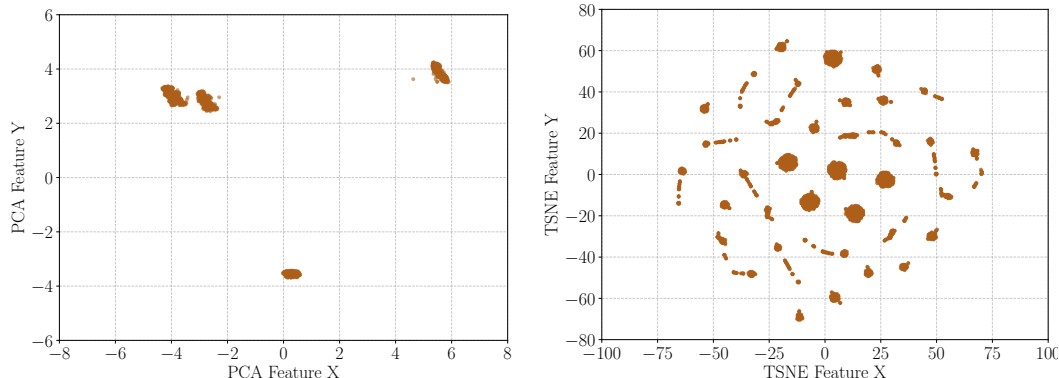

Figure 14: **Projections of the trained models.** PCA (left) and TSNE (right) provide two comple-
mentary views on the estimation of the posterior, which seems clearly multi-modal. The explained
variance of components X and Y of the PCA are 18% and 15%, respectively.

$\boldsymbol{\theta}$). The number of modes of the resulting layer-marginal posterior will be $(n-k)!$, with $k$ the number
of (nearly-)identical terms in $\boldsymbol{\theta}$. To recall, $n$ is of the order of $10^{1150}$ for the last layer of a ResNet-18.

This result seems difficult to extend to other distributions. While there exist some results on the
number of modes of a mixture of Gaussians, to the best of our knowledge, they either treat the case
with only two elements (Eisenberger, 1964) or are conjectures of an upper bound (Carreira-Perpinán
& Williams, 2003).

On an experimental basis, we tried to use the MeanShift algorithm (Comaniciu & Meer, 2002) to es-
timate the number of modes. Despite the number of checkpoints, Mean Shift remained very sensitive
to the bandwidth. It was, as expected, failing to detect modes in high dimensions but also providing
unsatisfactory results in simpler cases, such as for the MLP on energy-efficiency (see Section D.1).

## D.9 DETAILS ON THE FUNCTIONAL COLLAPSE IN ENSEMBLES

**Functional collapse on MNIST/FashionMNIST with OptuNet.** In this paragraph, we study
the potential functional collapse of OptuNet on MNIST and FashionMNIST, depicted in Figure 15.
First, we see that the dispersion of the mutual information both in the training domain (right) and
out of the domain (center) is greater than that of the ResNet-18. Indeed, in the left plot of Figure 15,

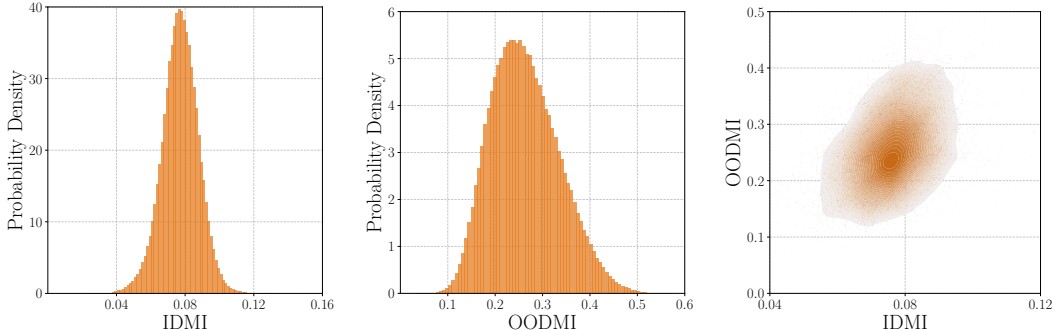

Figure 15: **Histograms of the in-distribution (left) and out-of-distribution distribution (center) functional diversities between couples of checkpoints as computed with mutual information; and joint plot (right) of OptuNets on MNIST.** There is a slightly more important functional collapse between couples of independently trained OptuNet on MNIST. Contrary to ResNet-18 on CIFAR-100, the in-distribution and out-of-distribution mutual information (IDMI, resp. OODMI) seem correlated, at least for small values.

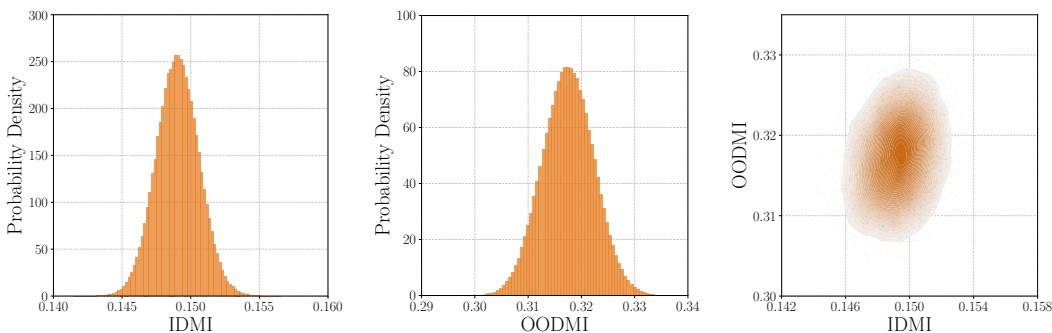

Figure 16: **Histograms of the in-distribution (left) and out-of-distribution distribution (center) functional diversities between couples of checkpoints as computed with Mutual Information; and joint plot (right) of ResNet-18s on Tiny-Imagenet.** As for ResNet-18 on CIFAR-100, the in-distribution and out-of-distribution mutual information (IDMI, resp. OODMI) seem only very slightly correlated.

there is a non-negligible probability of obtaining mutual information twice lower than the expected value. We note that the dispersion of the out-of-distribution values on FashionMNIST is also greater than for ResNets. Finally, in the case of OptuNet and contrasting with the results obtained with ResNets, there seems to be some positive correlation between the in-distribution MI and the OOD MI, at least when considering the lowest values. We can conclude that there is a high chance of *functional collapse* in small networks, which is expected considering that their posterior is simpler.

**Functional collapse on TinyImageNet and Textures.** Figure 16 shows the mutual information on both the in-distribution (left) and out-of-distribution (center) datasets and their correlation (right). The plots are much closer to Figure 3, which was expected as they share the same architecture. However, we see that the OOD dataset (here, Textures) potentially impacts the values of the mutual information. The dispersion (center) is lower than for ResNet-18 on SVHN, which may be more diverse. We measure the Pearson's $\rho$ correlation coefficient and obtain a statistically significant value of $\rho \approx 0.06$. The two values are, therefore, positively correlated with a very low coefficient.

### D.10 VISUALIZATION

In this section, we visualize the marginals of the posterior of neural networks, including OptuNets. Figure 17 provides the marginal distributions of the weights of a multi-layer perceptron trained on energy efficiency. We check with two-sample Kolmogorov-Smirnov tests that we cannot rule out

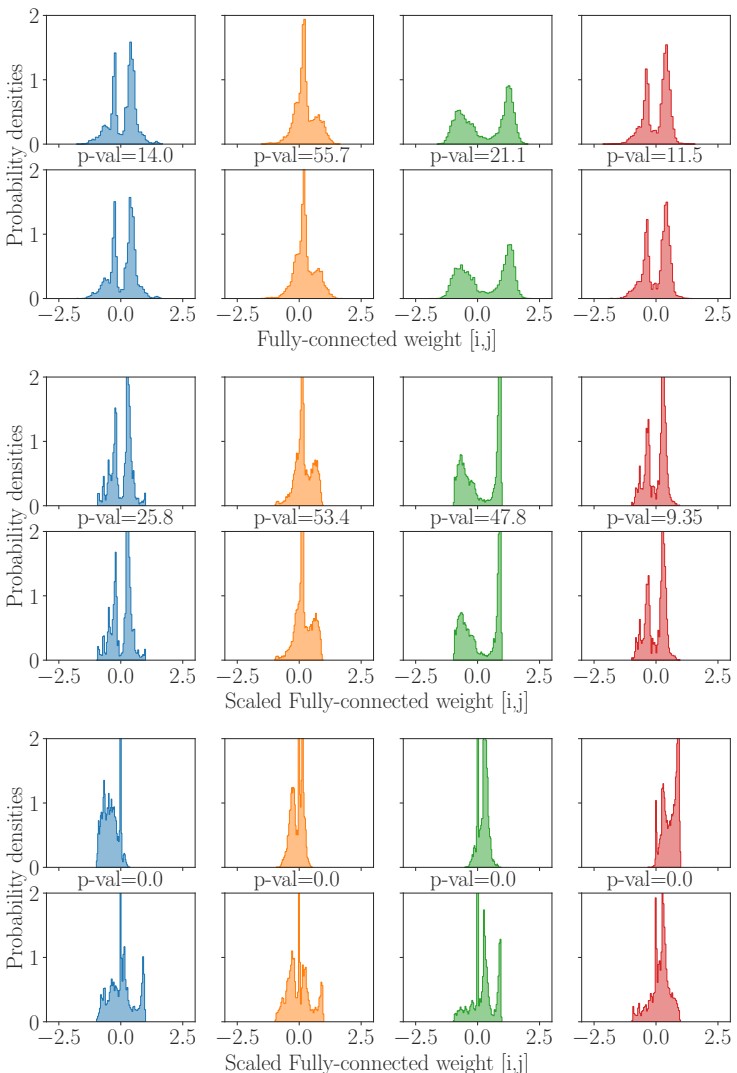

Figure 17: **Marginal posteriors of the last weights from a multi-layer perception:** The top rows are the original posterior, the middle rows have been scaled to a unit norm, and the bottom rows have been scaled and permuted. We indicate the p-values (in %) of a Kolmogorov-Smirnov two-sample test between the marginals of the pair of neurons.

that the distributions of input neurons from any layer are the same, indicating the corresponding p-values. Indeed, as long as the permutations are not removed, we keep the equivalence property of all the marginals.

Another objective is to provide visualizations of the posterior, as can be done on the loss landscape (see for instance (Garipov et al., 2018)). However, this task is extremely complex given that no means exist to estimate the discretized density of the posterior distribution. To circumvent this problem, we reduce the dimension of the posterior with principal component analysis with scikit-learn (Pedregosa et al., 2011). However the explained variance increases very slowly with the number of elements, which prevents meaningful interpretations.

## D.11 SYMMETRIES IN MODERN NETWORKS

This section provides details on how we compute symmetries throughout the paper.

### D.11.1 CONVOLUTIONAL LAYERS

In Appendix E, we generalize the row-wise and column-wise products to enable their use between vectors and four-dimensional tensors. We also detail how to understand the product between permutation matrices and four-dimensional tensors.

On a higher level, we see that the number of symmetries in convolutional layers is smaller than the number of parameters in a linear layer. Indeed, in our formalism, a linear layer's number of permutations and scaling degrees of freedom (sDoF) corresponds to its number of output neurons. In contrast, the number of permutations of a convolutional layer equals its number of output channels. The ratio of the number of permutations/sDoF over the number of parameters between linear and convolutional layers, therefore, equals the surface of the kernel of the convolutional layer.

### D.11.2 GROUPED-CONVOLUTIONAL LAYERS

The number of permutations of a grouped convolutional layer (Krizhevsky et al., 2012) reduces the number of permutations of the previous layer. Indeed, we have to account for the impossibility of fully reversing the permutation applied on the output neurons/channels of the preceding layer. This implies that the permutations of the previous layer need to remain intra-group. Say that the previous layer has $C_{\text{out}}$ output channels and that the current layer has $\gamma$ groups. Instead of $C_{\text{out}}!$ possible permutations, they are reduced to $\left(\frac{C_{\text{out}}}{\gamma}!\right)^{\gamma}$. On the other hand, the number of output permutations does not change (provided that the next layer does not have groups). Likewise, the number of scaling degrees of freedom remains unchanged as scaling symmetries modify the values of the weights channel per channel.

### D.11.3 BATCH NORMALIZATION LAYERS

Counting permutation symmetries on batch normalization layers (Ioffe & Szegedy, 2015) is quite straightforward. Regarding permutation symmetries, batch normalization layers are seamless, as one only needs to permute the weights, bias, running mean, and running variance. As such, they do not increase nor decrease the number of permutations.

On the other hand, batch normalization layers increase the number of degrees of freedom of scaling symmetries and make the minimum scaled representation cost degenerate. Indeed, we can scale the running variances and means to the input scaling coefficients and then choose new output coefficients for the weights and biases. They increase the number of sDoF by their number of features. Moreover, they allow the weights of the next layer to be extremely small as their weights are *not* multiplied by the inverse of the input scaling coefficient. Denoting with a tilde the updated weights, $\boldsymbol{\lambda}_{in}$ and $\boldsymbol{\lambda}_{out}$ the inbound and outbound scaling coefficients, $\boldsymbol{\omega}$ the weights of the layer, $\boldsymbol{b}$ its biases and $\boldsymbol{\mu}$, $\boldsymbol{\sigma}^2$ the mean and variance, we have

$$
\begin{aligned}
\tilde{\boldsymbol{\mu}} &= \boldsymbol{\mu} \cdot \boldsymbol{\lambda}_{in}, \\
\tilde{\boldsymbol{\sigma}^2} &= \boldsymbol{\sigma}^2 \cdot \boldsymbol{\lambda}_{in}^2, \\
\tilde{\boldsymbol{\omega}} &= \boldsymbol{\omega} \cdot \boldsymbol{\lambda}_{out}, \\
\tilde{\boldsymbol{b}} &= \boldsymbol{b} \cdot \boldsymbol{\lambda}_{out}.
\end{aligned}
\tag{12}
$$

This confirms that the inverse of $\boldsymbol{\lambda}_{in}$ is not taken into account in the computation of the representation cost. This allows convolutions to use extremely low $\boldsymbol{\lambda}_{out}$ reducing their weights to values as small as wanted. This involves that the minimum representation cost is degenerate (see Definition E.7) and we consider that the weights of the layers followed by batch normalization are null. Please note that we set the batch normalization epsilon (for training numerical stability) to 0 as it is not scaling invariant.

### D.11.4 RESIDUAL CONNECTIONS

Residual connections tend to reduce the number of permutation and scaling symmetries (Kurle et al., 2021), and this is an important aspect in ResNets. Indeed, residual connections need the output coefficient of the last layers and the permutation matrices from all inner nodes in the computation graph to be equal. This potentially reduces both the number of possible permutations

and the number of sDoF. In the case of ResNets, this effect is counterbalanced by the shortcut convolutional layers that can create permutations and sDoF, albeit less.

### D.12 ON MUTUAL INFORMATION

In our work, we use discrete mutual information as the main criterion for out-of-distribution detection and quantify the functional diversity between the predictions in the ensemble. Denoting the entropy function with $\mathcal{H}$, we compute the discrete mutual information with the following formula:

$$\text{MI} = \mathcal{H}\left(\frac{1}{M}\sum_{m=1}^{M}P_{\boldsymbol{\omega}_m}(\hat{y}|\boldsymbol{x}, \mathcal{D})\right) - \frac{1}{M}\sum_{m=1}^{M}\mathcal{H}(P_{\boldsymbol{\omega}_m}(\hat{y}|\boldsymbol{x}, \mathcal{D})), \tag{13}$$

where M is the number of predictors, each of weights $\boldsymbol{\omega}_m$. Mutual information is often used to quantify the epistemic uncertainty since it can be expressed as the difference between the total uncertainty and the aleatoric uncertainty (Smith & Gal, 2018).

# E  DETAILED FORMALISM, PROPOSITIONS, AND PROOFS

This section details the definitions, properties, and propositions and provides sketches of proofs.

Throughout the paper, we use an intuitive definition of neural networks that we formalize in the following definition:

**Definition E.1** (Neural network). Let $\boldsymbol{x} \in \mathbb{R}^d$ be an input datum, $r$ the rectified linear unit, and $s$ an almost everywhere differentiable activation function. We define the $L$-layer neural network $f_\theta$ with $\theta = \{W^{[l]}, b^{[l]}\}_{l \in [\![1,L]\!]}$ as follows:

$$
\begin{aligned}
&\boldsymbol{a}^{[0]} = \boldsymbol{x} \\
&\forall l \in [\![1, L]\!], \ \boldsymbol{z}^{[l]} = W^{[l]} \boldsymbol{a}^{[l-1]} + b^{[l]} \\
&\forall l \in [\![1, L-1]\!], \ \boldsymbol{a}^{[l]} = r(\boldsymbol{z}^{[l]}) \\
&f_\theta(\boldsymbol{x}) = s(\boldsymbol{z}^{[l]})
\end{aligned}
\tag{14}
$$

This definition of neural networks is limited to standard multilayer perceptions. However, it can be easily extended to a mix of convolutional and fully connected layers to the cost of a heavier formalism. To do this, we should replace the relationship between $\boldsymbol{z}^{[l]}$ and $\boldsymbol{a}^{[l-1]}$ in Equation 14 by a sum on the kernels of the cross-correlation operator. It would also be possible to account for residual connections using the computation graph.

**Property E.1** (Gradients and backpropagation). *Let us denote the training loss $\mathcal{L}(\boldsymbol{\omega})$ of the $L$-layer neural network $f_{\boldsymbol{\omega}}$ and $\boldsymbol{\omega} = \{W^{[l]}, b^{[l]}\}_{l \in [\![1,L]\!]}$. With, for all $l \in [\![1, L]\!]$, $\boldsymbol{\delta}^{[l]}$ as the gradient of $\mathcal{L}(\boldsymbol{\omega})$ with respect to $\boldsymbol{z}^{[l]}$, we have,*

$$
\begin{aligned}
&\boldsymbol{\delta}^{[L]} = \nabla_{\boldsymbol{z}^{[L]}} \mathcal{L}(\boldsymbol{\omega}) \\
&\forall l \in [\![1, L-1]\!], \boldsymbol{\delta}^{[l]} = \nabla_{\boldsymbol{z}^{[l]}} \mathcal{L}(\boldsymbol{\omega}) = \left( W^{[l+1]\intercal} \boldsymbol{\delta}^{[l+1]} \right) \circ \nabla_{\boldsymbol{z}^{[l]}} \boldsymbol{a}^{[l]} \\
&\forall l \in [\![1, L]\!], \nabla_{W^{[l]}} \mathcal{L}(\boldsymbol{\omega}) = \boldsymbol{\delta}^{[l]} \boldsymbol{a}^{[l-1]\intercal} \\
&\forall l \in [\![1, L]\!], \nabla_{b^{[l]}} \mathcal{L}(\boldsymbol{\omega}) = \boldsymbol{\delta}^{[l]}.
\end{aligned}
\tag{15}
$$

## E.1  EXTENDED FORMALISM ON SYMMETRIES

In this section, we present several definitions and properties to extend the formalism quickly described in the main paper.

### E.1.1  SCALING SYMMETRIES

We start by recalling the definitions of the line-wise and column-wise products between a vector and a matrix.

**Definition E.2.** We denote the line-wise product as $\triangledown$ and the column-wise product as $\triangleright$. Let $\boldsymbol{\omega} \in \mathbb{R}^{n \times m}$, $\boldsymbol{\lambda} \in \mathbb{R}^m$, and $\boldsymbol{\mu} \in \mathbb{R}^n$, with $n, m \geq 1$,

$$
\forall i \in [\![1, n]\!], j \in [\![1, m]\!], \ (\boldsymbol{\lambda} \triangledown \boldsymbol{\omega})_{i,j} = \lambda_j \omega_{i,j} \text{ and } (\boldsymbol{\mu} \triangleright \boldsymbol{\omega})_{i,j} = \mu_i \omega_{i,j}.
\tag{16}
$$

To handle biases, we extend these definitions to the product of vectors, considering the right vector as a one-dimensional matrix. It follows that for all vectors $\boldsymbol{\lambda}, \boldsymbol{a} \in \mathbb{R}^m$,

$$
\boldsymbol{\lambda} \triangledown \boldsymbol{a} = \boldsymbol{\lambda} \triangleright \boldsymbol{a} = \boldsymbol{\lambda} \cdot \boldsymbol{a}.
\tag{17}
$$

The line-wise and column-wise products between vectors and matrices *have priority* over the classic matrix product "$\times$" but not over the element-wise product between vectors "$\cdot$". We will include parentheses when deemed necessary for the clarity of the expressions.

**Non-negative homogeneity of the rectified linear unit.**  The rectified linear unit (Nair & Hinton, 2010), used as the standard activation function in most networks – with the notable exception of vision transformers (Dosovitskiy et al., 2021) – is non-negative homogenous and therefore authorizes scaling symmetries (Neyshabur et al., 2015).

**Property E.2.** *Denote $r$ the rectified linear unit such that $r(x) = x_+$. For all vectors $\boldsymbol{\lambda} \in (\mathbb{R}^+)^d$ and $\boldsymbol{x} \in \mathbb{R}^d$, $r$ is positive-homogeneous,* i.e.

$$r(\boldsymbol{\lambda} \cdot \boldsymbol{x}) = \boldsymbol{\lambda} \cdot r(\boldsymbol{x}). \tag{18}$$

**Some properties on line-wise and column-wise products.** We provide a few simple properties to help understand these notations.

**Property E.3.** *For all vectors $\boldsymbol{\lambda}^{(1)}, \boldsymbol{\lambda}^{(2)} \in \mathbb{R}^n$, $\boldsymbol{\theta} \in \mathbb{R}^{\cdot \times n}$, and $\boldsymbol{\omega} \in \mathbb{R}^{n \times \cdot}$*

$$(\boldsymbol{\lambda}^{(1)} \bigtriangledown \boldsymbol{\theta}) \times (\boldsymbol{\lambda}^{(2)} \triangleright \boldsymbol{\omega}) = (\boldsymbol{\lambda}^{(1)} \cdot \boldsymbol{\lambda}^{(2)}) \bigtriangledown \boldsymbol{\theta} \times \boldsymbol{\omega} = \boldsymbol{\theta} \times (\boldsymbol{\lambda}^{(1)} \cdot \boldsymbol{\lambda}^{(2)}) \triangleright \boldsymbol{\omega} \tag{19}$$

*Proof.* Let us denote $A = (\boldsymbol{\lambda}^{(1)} \bigtriangledown \boldsymbol{\theta}) \times (\boldsymbol{\lambda}^{(2)} \triangleright \boldsymbol{\omega})$

$$A_{i,j} = \sum_{k=1}^{n} \lambda_k^{(1)} \theta_{i,k} \lambda_k^{(2)} \omega_{k,j} = \sum_{k=1}^{n} \theta_{i,k} \lambda_k^{(2)} \lambda_k^{(1)} \omega_{k,j} = (\boldsymbol{\theta} \times (\boldsymbol{\lambda}^{(1)} \cdot \boldsymbol{\lambda}^{(2)}) \triangleright \boldsymbol{\omega})_{i,j} \tag{20}$$

$$= \sum_{k=1}^{n} \lambda_k^{(2)} \lambda_k^{(1)} \theta_{i,k} \omega_{k,j} = ((\boldsymbol{\lambda}^{(1)} \cdot \boldsymbol{\lambda}^{(2)}) \bigtriangledown \boldsymbol{\theta} \times \boldsymbol{\omega})_{i,j} \tag{21}$$

$\blacksquare$

However, please note that we do not have $\boldsymbol{\lambda}^{(1)} \bigtriangledown (\boldsymbol{\lambda}^{(2)} \triangleright \boldsymbol{\omega}) = (\boldsymbol{\lambda}^{(1)} \cdot \boldsymbol{\lambda}^{(2)}) \bigtriangledown \boldsymbol{\omega}$ in general.

**Property E.4.** *For all vectors $\boldsymbol{\lambda}^{(1)} \in (\mathbb{R}_{\neq 0})^n$, $\boldsymbol{\lambda}^{(2)} \in \mathbb{R}^n$, $\boldsymbol{\theta} \in \mathbb{R}^{\cdot \times n}$, and $\boldsymbol{\omega} \in \mathbb{R}^{n \times \cdot}$*

$$(\boldsymbol{\lambda}^{(1)} \bigtriangledown \boldsymbol{\theta}) \times r(\boldsymbol{\lambda}^{(2)} \triangleright \boldsymbol{\omega}) = \boldsymbol{\theta} \times r\left((\boldsymbol{\lambda}^{(1)} \cdot \boldsymbol{\lambda}^{(2)}) \triangleright \boldsymbol{\omega}\right) \tag{22}$$

*Proof.* Let us denote $B = (\boldsymbol{\lambda}_1 \bigtriangledown \boldsymbol{\theta}) \times r(\boldsymbol{\lambda}_2 \triangleright \boldsymbol{\omega})$

$$B_{i,j} = \sum_{k=1}^{n} \lambda_k^{(1)} \theta_{i,k} r(\lambda_k^{(2)} \omega_{k,j}) \tag{23}$$

$$= \sum_{k=1}^{n} \theta_{i,k} r(\lambda_k^{(1)} \lambda_k^{(2)} \omega_{k,j}) \tag{24}$$

$$= (\boldsymbol{\theta} \times r((\boldsymbol{\lambda}^{(1)} \cdot \boldsymbol{\lambda}^{(2)}) \triangleright \boldsymbol{\omega}))_{i,j} \tag{25}$$

$\blacksquare$

This property is crucial for scaling symmetries, which are one of its special cases. Indeed, for all vectors $\boldsymbol{\lambda} \in (\mathbb{R}_{>0})^n$,

$$(\boldsymbol{\lambda}^{-1} \bigtriangledown \boldsymbol{\theta}) \times r(\boldsymbol{\lambda} \triangleright \boldsymbol{\omega}) = \boldsymbol{\theta} \times r(\boldsymbol{\omega}). \tag{26}$$

Finally, we can gather the previous results to extend Equation 26 to weights and biases in the following property.

**Property E.5.** *For all vectors $\boldsymbol{\lambda} \in (\mathbb{R}_{>0})^n$, $\boldsymbol{b} \in \mathbb{R}^n$, $\boldsymbol{\theta} \in \mathbb{R}^{\cdot \times n}$, and $\boldsymbol{\omega} \in \mathbb{R}^{n \times \cdot}$*

$$(\boldsymbol{\lambda}^{-1} \bigtriangledown \boldsymbol{\theta}) \times r(\boldsymbol{\lambda} \triangleright (\boldsymbol{b} + \boldsymbol{\omega})) = \boldsymbol{\theta} \times r(\boldsymbol{b} + \boldsymbol{\omega}) \tag{27}$$

*Proof.*

$$(\boldsymbol{\lambda}^{-1} \bigtriangledown \boldsymbol{\theta}) \times r(\boldsymbol{\lambda} \triangleright (\boldsymbol{b} + \boldsymbol{\omega})) = (\boldsymbol{\lambda}^{-1} \bigtriangledown \boldsymbol{\theta}) \times r(\boldsymbol{\lambda} \cdot \boldsymbol{b} + \boldsymbol{\lambda} \triangleright \boldsymbol{\omega}) \tag{28}$$

$$= \boldsymbol{\theta} \times r(\boldsymbol{\lambda}^{-1} \boldsymbol{\lambda} \cdot \boldsymbol{b} + \boldsymbol{\lambda}^{-1} \boldsymbol{\lambda} \triangleright \boldsymbol{\omega}) \tag{29}$$

$$= \boldsymbol{\theta} \times r(\boldsymbol{b} + \boldsymbol{\omega}) \tag{30}$$

$\blacksquare$

This property is provided for the simplest case with only two matrices. However, it can be trivially extended to an arbitrary number of matrices, chaining the different line-wise and column-wise multiplications. It follows that we can extend the definition of scaling symmetries to deeper networks.

Moreover, we can also extend the scaling symmetries to mixes of convolutional and linear networks. To do this, we first extend the definition of the line-wise and column-wise products to convolutional layers through the following definition.

**Definition E.3.** Let the 4-dimensional tensor $\boldsymbol{\omega} \in \mathbb{R}^{C_{\text{out}} \times C_{\text{in}} \times k_v \times k_h}$ be the weight of a convolutional layer of input and output channels $C_{\text{in}}$ and $C_{\text{out}}$, and kernel $k_v \times k_h$. Denote $\boldsymbol{\lambda} \in \mathbb{R}^{C_{\text{in}}}$ and $\boldsymbol{\mu} \in \mathbb{R}^{C_{\text{out}}}$ We extend the operators $\triangleright$ and $\triangledown$ to products between vectors and 4-dimensional tensors and have,

$$\forall c_{out} \in [\![1, C_{\text{out}}]\!], c_{in} \in [\![1, C_{\text{in}}]\!], i \in [\![1, n]\!], j \in [\![1, m]\!],$$
$$(\boldsymbol{\lambda} \triangledown \boldsymbol{\omega})_{c_{out}, c_{in}, i, j} = \lambda_{c_{in}} \omega_{c_{out}, c_{in}, i, j}, \text{ and } (\boldsymbol{\mu} \triangleright \boldsymbol{\omega})_{c_{out}, c_{in}, i, j} = \mu_{c_{out}} \omega_{c_{out}, c_{in}, i, j}. \tag{31}$$

With this definition, we keep the previous properties and enable chaining transformations on both convolutional and fully connected layers.

**Equivalence with standard linear algebra.** We design the line-wise ($\triangledown$) and column-wise ($\triangleright$) notations for their intuitiveness. However, when applied to fully connected layers, they are equivalent to more common notations from linear algebra. Indeed, sticking to $\boldsymbol{\omega} \in \mathbb{R}^{n \times m}, \boldsymbol{\lambda} \in \mathbb{R}^m$, and $\boldsymbol{\mu} \in \mathbb{R}^n$, with $n, m \geq 1$, we have that

$$\boldsymbol{\lambda} \triangledown \boldsymbol{\omega} = \text{diag}(\boldsymbol{\lambda}) \times \boldsymbol{\omega} \text{ and } \boldsymbol{\mu} \triangleright \boldsymbol{\omega} = \boldsymbol{\omega} \times \text{diag}(\boldsymbol{\mu}). \tag{32}$$

Furthermore, we can also write, with $\circ$ the Hadamard (elementwise) product between matrices,

$$\boldsymbol{\lambda} \triangledown (\boldsymbol{\mu} \triangleright \boldsymbol{\omega}) = (\boldsymbol{\lambda}^\mathsf{T} \times \boldsymbol{\mu}) \circ \boldsymbol{\omega}. \tag{33}$$

These equivalences are important for implementing the log-log convex problem to minimize the L2 regularization term on the space of the scaling symmetries (see Section 3.6).

### E.1.2 PERMUTATION SYMMETRIES

**Definition E.4.** We define $P_n$ as the set of permutations of vectors of size $n$. $P_n$ contains $|P_n| = n!$ elements. The elements of $P_n$ are doubly-stochastic binary matrices from $\{0, 1\}^{n \times n}$. Finally, for all $\boldsymbol{\pi}_1 \in P_n, \boldsymbol{\pi}_1 \times \boldsymbol{\pi}_1^\mathsf{T} = \mathbb{1}_n$.

From this definition and property, we can directly derive the following result:

**Property E.6.** *For all $\boldsymbol{\theta} \in \mathbb{R}^{\cdot \times m}, \boldsymbol{\omega} \in \mathbb{R}^{m \times n}$, and permutation matrix $\boldsymbol{\pi} \in P_m$,*

$$\forall \boldsymbol{x} \in \mathbb{R}^n, \; \boldsymbol{\theta} \boldsymbol{\pi}^\mathsf{T} \times r (\boldsymbol{\pi} \times \boldsymbol{\omega} \boldsymbol{x}) = \boldsymbol{\theta} \times r(\boldsymbol{\omega} \boldsymbol{x}). \tag{34}$$

*Proof.* Let us take $m, n \in \mathbb{N}, \boldsymbol{\omega} \in \mathbb{R}^{m \times n}, \boldsymbol{b} \in \mathbb{R}^m, \boldsymbol{x} \in \mathbb{R}$, and $\boldsymbol{\pi} \in P_m$ a permutation matrix corresponding to the bijective mapping $\sigma$ in $[\![1, m]\!]$. We have that,

$$[\boldsymbol{\pi}(\boldsymbol{b} + \boldsymbol{\omega}) \times \boldsymbol{x}]_i = \boldsymbol{b}_{\sigma(i)} + \sum_{k=1}^n \omega_{\sigma(i), k} x_k. \tag{35}$$

Hence,

$$[\boldsymbol{\pi}^\mathsf{T} \times r(\boldsymbol{\pi}(\boldsymbol{\omega}) \times \boldsymbol{x})]_i = \boldsymbol{b}_{\sigma^{-1}(\sigma(i))} + \sum_{k=1}^n r \left( \omega_{\sigma^{-1}(\sigma(i)), k} \cdot x_k \right) \tag{36}$$

$$= [r(\boldsymbol{b} + \boldsymbol{\omega} \times \boldsymbol{x})]_i, \tag{37}$$

and it comes directly that,

$$\boldsymbol{\theta} \boldsymbol{\pi}^\mathsf{T} \times r (\boldsymbol{\pi} \times \boldsymbol{\omega} \boldsymbol{x}) = \boldsymbol{\theta} \times r(\boldsymbol{\omega} \boldsymbol{x}). \tag{38}$$

$\blacksquare$

This property is provided for the simplest case with only two matrices. However, it can be trivially extended to an arbitrary number of matrices, chaining the different permutations. Just like for scaling symmetries, it follows that we can extend the definition of permutation symmetries to deeper networks.

Moreover, as for scaling symmetries in definition E.3, we can extend permutation operations and symmetries to convolutional layers and their 4-dimensional tensors. To do this, we consider that the permutation matrices act on the output and input channels and permute the whole kernels.

### E.1.3 SOFTMAX ADDITIVE SYMMETRY

For the sake of completeness, we also recall the additive softmax symmetries in this section. We left this symmetry apart in our analysis as it involves, at most, one degree of freedom.

**Definition E.5.** We recall that the softmax function $\sigma$ is defined by the following equation, given the logits $\boldsymbol{a} \in \mathbb{R}^n$:

$$\forall i \in [\![1, n]\!], \sigma(\boldsymbol{a})_i = \frac{e^{a_i}}{\sum\limits_{j=1}^{n} e^{a_j}}. \tag{39}$$

**Property E.7.** *For all $\boldsymbol{a}, \boldsymbol{b} \in \mathbb{R}^n$, $\lambda \in \mathbb{R}$, and + the point-wise sum when applied between vectors and scalars,*

$$\sigma(\boldsymbol{a} + \boldsymbol{b} + \lambda) = \sigma(\boldsymbol{a} + \boldsymbol{b}). \tag{40}$$

*Proof.* Denote $S = \sigma(\boldsymbol{a} + \boldsymbol{b} + \lambda)$ and take $i$ in $[\![1, n]\!]$,

$$S_i = \frac{e^{a_i + b_i + \lambda}}{\sum\limits_{j=1}^{n} e^{a_j + b_j + \lambda}} \tag{41}$$

$$S_i = \frac{e^{\lambda} e^{a_i + b_i}}{e^{\lambda} \sum\limits_{j=1}^{n} e^{a_j + b_j}} \tag{42}$$

$$S_i = (\sigma(\boldsymbol{a} + \boldsymbol{b}))_i \tag{43}$$

∎

### E.2 PERMUTATION-EQUIVARIANCE OF THE GRADIENT OF THE LOSS AND SGD

Before coming to the equivariance of the training operator, we start by proving the following lemma.

**Lemma 1.** *Let $f_\theta$ be a neural network trained with a loss $\mathcal{L}(\theta)$, the gradient of the loss $\nabla\mathcal{L}$ is permutation equivariant, i.e.,*

$$\forall \Pi \in \mathbb{\Pi}, \ \nabla\mathcal{L}(\mathcal{T}_p(\theta, \Pi)) = \mathcal{T}_p(\nabla\mathcal{L}(\theta), \Pi). \tag{44}$$

Please be aware that the notation $\mathcal{T}_p(\nabla\mathcal{L}(\theta), \Pi)$ signifies that we apply a permutation operation to the gradient of the loss function $\mathcal{L}$ with respect to the weight $\theta$.

*Proof.* Let $f_\theta$ be a neural network trained with a loss $\mathcal{L}(\theta)$ and $\Pi \in \mathbb{\Pi}$. We prove the case with $\Pi$ containing only Identity matrices except for the $i$-th matrix, that we denote $\pi$.

From property E.1, we have that $\forall l \in [\![1, L]\!], \nabla_{W^{[l]}}\mathcal{L}(\theta) = \boldsymbol{\delta}^{[l]}(\theta)\boldsymbol{a}^{[l-1]^\top}(\theta)$ with $\boldsymbol{\delta}^{[l]}(\theta) = \nabla_{\boldsymbol{z}^{[l]}}\mathcal{L}(\theta)$ and $\boldsymbol{a}^{[l-1]}(\theta)$ the activation after the layer $l-1$. We add the notation of dependence of $\boldsymbol{\delta}^{[l]}$ and $\boldsymbol{a}^{[l]}$ in $\theta$ to express that they correspond to the gradient and the activation

of the network of weights $\theta$. We can adapt this result to the layer $i$ and $\mathcal{T}(\theta, \Pi)$:

$$\nabla_{\pi W^{[i]}} \mathcal{L}(\mathcal{T}_p(\theta, \Pi)) = \boldsymbol{\delta}^{[i]}(\mathcal{T}_p(\theta, \Pi)) \boldsymbol{a}^{[i-1]^\intercal}(\mathcal{T}_p(\theta, \Pi)) \tag{45}$$

$$= \left( \left( W^{[i+1]} \boldsymbol{\pi}^\intercal \right)^\intercal \boldsymbol{\delta}^{[i+1]}(\mathcal{T}_p(\theta, \Pi)) \circ \nabla_{\pi z^{[l]}} \boldsymbol{\pi} \boldsymbol{a}^{[l]}(\theta) \right) \boldsymbol{a}^{[i-1]^\intercal}(\mathcal{T}_p(\theta, \Pi)) \tag{46}$$

$$= \left( \boldsymbol{\pi} W^{[i+1]^\intercal} \boldsymbol{\delta}^{[i+1]}(T(\theta, \Pi)) \circ \boldsymbol{\pi} \nabla_{z^{[l]}} \boldsymbol{a}^{[l]}(\theta) \right) \boldsymbol{a}^{[i-1]^\intercal}(\mathcal{T}_p(\theta, \Pi)) \tag{47}$$

$$= \left( \boldsymbol{\pi} W^{[i+1]^\intercal} \boldsymbol{\delta}^{[i+1]}(\theta) \circ \boldsymbol{\pi} \nabla_{z^{[l]}} \boldsymbol{a}^{[l]}(\theta) \right) \boldsymbol{a}^{[i-1]^\intercal}(\theta) \tag{48}$$

$$= \boldsymbol{\pi} \boldsymbol{\delta}^{[l]}(\theta) \boldsymbol{a}^{[i-1]^\intercal}(\theta) \tag{49}$$

$$= \boldsymbol{\pi} \nabla_{W^{[i]}} \mathcal{L}(\theta) \tag{50}$$

This proof can be extended backward to any $\Pi \in \mathbb{\Pi}$ as the action of $\pi^\intercal$ and of $\pi$ cancel out for all the following weights. $\blacksquare$

We proved Lemma 1 with an MLP, but this result extends to convolutional layers. To do this, we need to define that the permutation matrices apply on the channels of the weight tensors.

To simplify the notations, we use neural networks and weights indifferently as arguments for the optimization and symmetry operators $\star$ and $\mathcal{T}$. For instance, we will simplify $\star(f_\theta)$ by $\star(\theta)$, implicitly assuming the architecture of $f_\theta$.

**Proposition 3.** *Let $f_{\theta_0}$ be a randomly initialized neural network and $\star_s$ the stochastic gradient descent operator applying $s$ times the update rule of learning rate $\gamma_s$. For all steps $s \in \mathbb{N}$, $\star_s$ is permutation equivariant. In other words,*

$$\forall s \in \mathbb{N}, \forall \Pi \in \mathbb{\Pi}, \ \star_s(\mathcal{T}_p(\theta, \Pi)) = \mathcal{T}_p(\star_s(\theta), \Pi). \tag{51}$$

We provide the proof below for multi-layer perceptrons, but it extends to convolutional neural networks.

*Proof.* Let $\Pi \in \mathbb{\Pi}$ be a permutation set applied on a neural network $f_\theta$ trained with an objective $\mathcal{L}$ of gradient $\nabla \mathcal{L}$. Let us define $P(s)$ as follows:

$$P(s) = \text{``} \mathcal{T}_p\left(\star_s(\theta_0), \Pi\right) = \star_s\left(\mathcal{T}_p(\theta_0, \Pi)\right) \text{''} \tag{52}$$

We give a proof that $P(s)$ is verified for all $s \in \mathbb{N}$ by induction on $s$.

*Base case:* $P(0)$ is trivially true as $\star_0(f_{\theta_0}) = f_{\theta_0}$, hence

$$\star_0\left(\mathcal{T}_p(\theta_0, \Pi)\right) = \mathcal{T}_p(\theta_0, \Pi) = \mathcal{T}_p\left(\star_0(\theta_0), \Pi\right). \tag{53}$$

*Induction step:* Assume the induction hypothesis that for a particular $k$, the single case $s = k$ holds, meaning $P(k)$ is true:

$$\mathcal{T}_p\left(\star_k(\theta_0), \Pi\right) = \star_k\left(\mathcal{T}_p(\theta_0, \Pi)\right) \tag{54}$$

Let us start with $\mathcal{T}_p(\star_{k+1}(\theta_0), \Pi)$. We have that

$$\mathcal{T}_p(\star_{k+1}(\theta_0), \Pi) = \mathcal{T}_p(\theta_{k+1}, \Pi) \tag{55}$$

$$= \mathcal{T}_p(\theta_k - \gamma_s \nabla \mathcal{L}(\theta_k), \Pi) \tag{56}$$

$$= \mathcal{T}_p(\star_k(\theta_0), \Pi) - \gamma_s \mathcal{T}_p(\nabla \mathcal{L}(\star_k(\theta_0)), \Pi) \tag{57}$$

$$= \star_k\left(\mathcal{T}_p(\theta_0), \Pi\right) - \gamma_s \nabla \mathcal{L}(\star_k(\mathcal{T}_p(\theta_0, \Pi))) \tag{58}$$

$$= \star_{k+1}\left(\mathcal{T}_p(\theta_0, \Pi)\right), \tag{59}$$

Equation 58 uses the permutation-equivariance of both the gradient (see Lemma 1) and the training operator until step $k$. Furthermore, equation 59 is exactly $P(k+1)$, establishing the induction step.

*Conclusion:* Since both the base case and the induction step have been proven, by induction, the statement $P(s)$ holds for every natural number $s$. $\blacksquare$

We provide the proof of Proposition 3 with SGD for simplicity. However, it intuitively extends to SGD with momentum since the momentum is also permutation-equivariant and updated in a permutation-equivariant way. Furthermore, it also generalizes to (mini-)batch stochastic gradient descent, provided that the batches contain the same images.

While this also extends to the Adam optimizer (Kingma & Ba, 2015), this is most likely not true for new meta-learned optimizers such as VeLO (Metz et al., 2022).

### E.3 EQUIPROBABILITY OF THE PERMUTATIONS

We build upon the results from the previous sections to devise the following proposition.

**Proposition 4.** *Let $f_\theta$ be a neural network with initial weights independently and layer-wise identically distributed. The probability of converging towards any symmetrically permuted network $\mathcal{T}_p(f_\theta, \Pi)$ given the dataset and training hyperparameters does not depend on the permutation set $\Pi$. In other words,*

$$p(\Theta_s = \mathcal{T}_p(\theta_s, \Pi)|\mathcal{D}) = p(\Theta_s = \theta_s|\mathcal{D}). \tag{60}$$

*Proof.* For simplicity, we assume an optimization scheme with early stopping and final weights $\theta_s$ at the end of step $s$. Given $\theta_s$, we can denote $\theta_{0 \rightarrowtail s}$ the space of initializations with non-zero probability to converge to $\theta_s$ after $s$ steps.

Let there be a permutation set $\Pi$ leaving $f_{\theta_s}$ invariant. We want to prove Equation 60, *i.e.*, that the probability to converge to $\mathcal{T}_p(\Theta_s = \theta_s, \Pi)$ given $\mathcal{D}$ is the same as for $\theta_s$ for any permutation set $\Pi \in \mathbb{\Pi}$.

We can inject information about the potential initialization points, defining $\theta_{0 \rightarrowtail s} = \{\theta | \star_s [\theta] = \theta_s\}$ as the pre-image of $\theta_s$ by the optimization procedure with $s$ steps. It comes that

$$p(\Theta_s = \theta_s, \mathcal{D}) = p(\Theta_0 \in \theta_{0 \rightarrowtail s}, \Theta_s = \theta_s, \mathcal{D}). \tag{61}$$

Indeed, as we use the notation $\Theta_0 \in \theta_{0 \rightarrowtail s}$ for

$$p(\Theta_0 \in \theta_{0 \rightarrowtail s}) = \int_{\theta \in \theta_{0 \rightarrowtail s}} p(\Theta_0 = \theta)d\theta, \tag{62}$$

we can get, with $\Omega$ the set of possible weights,

$$\int_{\theta \in \Omega} p(\Theta_0 = \theta|\Theta_s = \theta_s, \mathcal{D})d\theta = \int_{\theta \in \theta_{0 \rightarrowtail s}} p(\Theta_0 = \theta|\Theta_s = \theta_s, \mathcal{D})d\theta + \int_{\theta \in \Omega \setminus \theta_{0 \rightarrowtail s}} p(\Theta_0 = \theta|\Theta_s = \theta_s, \mathcal{D})d\theta \tag{63}$$

$$\tag{64}$$

The left term is equal to 1, and so is the middle term as, by definition of $\theta_{0 \rightarrowtail s}$, it is impossible to have initialized the weights in $\Omega \setminus \theta_{0 \rightarrowtail s}$ and converge to $\theta_s$ given the dataset $\mathcal{D}$. Since $p(\Theta_0 \in \theta_{0 \rightarrowtail s}, \Theta_s = \theta_s, \mathcal{D}) = p(\Theta_0 \in \theta_{0 \rightarrowtail s}|\Theta_s = \theta_s, \mathcal{D})p(\Theta_s = \theta_s, \mathcal{D})$, we have Equation 61.

Furthermore, the definition of conditional probabilities implies that the probability of converging to $\theta_s$ given $\mathcal{D}$ can be rewritten as

$$p(\Theta_s = \theta_s|\mathcal{D}) = \frac{p(\Theta_0 \in \theta_{0 \rightarrowtail s}, \Theta_s = \theta_s, \mathcal{D}) \cdot p(\Theta_0 \in \theta_{0 \rightarrowtail s})}{p(\mathcal{D}) \cdot p(\Theta_0 \in \theta_{0 \rightarrowtail s})} \tag{65}$$

$$= \frac{p(\Theta_0 \in \theta_{0 \rightarrowtail s}, \Theta_s = \theta_s, \mathcal{D}) \cdot p(\Theta_0 \in \theta_{0 \rightarrowtail s})}{p(\Theta_0 \in \theta_{0 \rightarrowtail s}, \mathcal{D})} \quad \text{[independence of } \Theta_0 \text{ and } \mathcal{D}] \tag{66}$$

$$= p(\Theta_s = \theta_s|\Theta_0 \in \theta_{0 \rightarrowtail s}, \mathcal{D})p(\Theta_0 \in \theta_{0 \rightarrowtail s}). \tag{67}$$

We can also apply Equation 67 for a permuted network. Let there be a permutation set $\Pi$; Equation 67 is also valid for the permuted network $\mathcal{T}_p(\theta_s, \Pi)$ with its pre-image the set $\mathcal{T}_p(\theta_{0 \rightarrowtail s}, \Pi)$ (defined as $\{\mathcal{T}_p(\theta, \Pi)\}_{\theta \in \theta_{0 \rightarrowtail s}}$). Indeed, Proposition 3 exactly proves that training a network starting from the permuted weights $\mathcal{T}_p(\theta_0, \Pi)$ converges towards the permutation of the original weights $\mathcal{T}_p(\theta_s, \Pi)$.

$$p(\Theta_s = \mathcal{T}_p(\theta_s, \Pi)|\mathcal{D}) = p(\Theta_s = \mathcal{T}_p(\theta_s, \Pi)|\Theta_0 \in \mathcal{T}_p(\theta_{0 \rightarrowtail s}, \Pi), \mathcal{D})p(\Theta_0 \in \mathcal{T}_p(\theta_{0 \rightarrowtail s}, \Pi)) \tag{68}$$

*Deterministic Training:* Suppose the optimization algorithm is fully deterministic, given the initialization state. In practice, the dataloader seed would be set to some value, and the backpropagation kernels would be chosen as deterministic. In that case, the convergence to $\theta_s$ is fully determined by the initialization $\Theta_0$ and the dataset $\mathcal{D}$, *i.e.*,

$$p(\Theta_s = \theta_s | \Theta_0 \in \theta_{0 \to s}, \mathcal{D}) = 1 \tag{69}$$

*Non-deterministic Training:* In the non-deterministic setting, corresponding in practice to the order of the inputs in the batches as well as the stochasticity of backpropagation algorithms. We can marginalize the (eventually multivariate) sources of stochasticity $\xi \in \Xi$ to regain a deterministic relationship.

$$p(\Theta_s = \theta_s | \Theta_0 \in \theta_{0 \to s}, \mathcal{D}) = \int_{\xi \in \Xi} p(\Theta_s = \theta_s | \Theta_0 \in \theta_{0 \to s}, \xi, \mathcal{D}) d\xi \tag{70}$$

$$= \int_{\xi \in \Xi} p(\Theta_s = \mathcal{T}_p(\theta_s, \Pi) | \Theta_0 \in \mathcal{T}_p(\theta_{0 \to s}, \Pi), \xi, \mathcal{D}) d\xi \tag{71}$$

$$= p(\Theta_s = \mathcal{T}_p(\theta_s, \Pi) | \Theta_0 \in \mathcal{T}_p(\theta_{0 \to s}, \Pi), \mathcal{D}) \tag{72}$$

In the case of deterministic training, the left term of Equation 68 is equal to 1 from Equation 69 with the permuted weights $\mathcal{T}_p(\theta_0, \Pi)$ and $\mathcal{T}_p(\theta_s, \Pi)$.

For the right term, $p(\Theta_0 \in \mathcal{T}_p(\theta_{0 \to s}, \Pi)) = p(\Theta_0 \in \theta_{0 \to s})$, since the initial weights are layer-wise identically distributed and the permutations act separately inside the layers.

Finally, we have

$$p(\Theta_s = \mathcal{T}_p(\theta_s, \Pi) | \mathcal{D}) = p(\Theta_s = \theta_s | \mathcal{D}), \tag{73}$$

which is Proposition 4. ∎

### E.4 THEORETICAL INSIGHTS ON THE BAYESIAN POSTERIOR

We start with the following proposition, independent from Proposition 1, but still deemed interesting for the reader.

**Proposition 5.** *Let $f_{\boldsymbol{\omega}}$ be a neural network, $\boldsymbol{x}$ an input vector, and $y$ a scalar. We also denote $\Omega_>$ the space of sorted weights, and $\Pi$ remains the set of possible permutations for $f_{\boldsymbol{\omega}}$. We have*

$$p(y \mid \boldsymbol{x}, \mathcal{D}) = |\Pi| \int_{\boldsymbol{\omega} \in \Omega_>} p(y \mid \boldsymbol{x}, \boldsymbol{\omega}) p(\boldsymbol{\omega} \mid \mathcal{D}) d\boldsymbol{\omega}. \tag{74}$$

*Proof.* Let $f_{\boldsymbol{\omega}}$ be a neural network. We recall the marginalization on the network's weights:

$$p(y \mid \boldsymbol{x}, \mathcal{D}) = \int_{\boldsymbol{\omega} \in \Omega} p(y \mid \boldsymbol{x}, \boldsymbol{\omega}) p(\boldsymbol{\omega} \mid \mathcal{D}) d\boldsymbol{\omega}. \tag{75}$$

We can split the space $\Omega$ and get $\Omega = \Omega_\geq \cup \bigcup_{i \in [\![1, |\Pi|]\!]} \Omega_{>_i}$, with $\Omega_{>_i}$ the space of the weights ordered and permuted according to $\Pi$ and $\Omega_\geq$ the space including corner-cases: the networks with at least twice the same weights in a layer. In our case, $\Omega_\geq$ is theoretically of measure zero. In practice, we note that the reals are discretized on a computer, and the edge cases may happen with an extremely small probability, decreasing with the chosen precision. Denote $\Omega_>$ the space with descending ordered weights, we have

$$p(y \mid \boldsymbol{x}, \mathcal{D}) = \sum_{i=1}^{|\Pi|} \int_{\boldsymbol{\omega} \in \Omega_>} p(y \mid \boldsymbol{x}, \mathcal{T}_p(\boldsymbol{\omega}, \Pi_i)) p(\mathcal{T}_p(\boldsymbol{\omega}, \Pi_i) \mid \mathcal{D}) d\boldsymbol{\omega}. \tag{76}$$

By definition of the symmetry operator, we know that it does not affect the prediction of the transformed network:

$$\forall i \in [\![1, |\mathbb{\Pi}|]\!], \; p(y \mid \boldsymbol{x}, \mathcal{T}_p(\boldsymbol{\omega}, \Pi_i)) = p(y \mid \boldsymbol{x}, \boldsymbol{\omega}). \tag{77}$$

Moreover, Proposition 4 provides

$$\forall i, j \in [\![1, |\mathbb{\Pi}|]\!], \; p(\mathcal{T}_p(\boldsymbol{\omega}, \Pi_i) \mid \mathcal{D}) = p(\boldsymbol{\omega} \mid \mathcal{D}). \tag{78}$$

It follows that

$$p(y \mid \boldsymbol{x}, \mathcal{D}) = |\mathbb{\Pi}| \int_{\boldsymbol{\omega} \in \Omega_>} p(y \mid \boldsymbol{x}, \boldsymbol{\omega}) p(\boldsymbol{\omega} \mid \mathcal{D}) d\boldsymbol{\omega}. \tag{79}$$

∎

Then, we recall Proposition 1 of the main paper:

**Proposition.** *Define $f_\omega$ a neural network and $\tilde{\omega}$ its corresponding identifiable model - a network transformed for having sorted unit-normed neurons. Let us also denote $\mathbb{\Pi}$ and $\mathbb{\Lambda}$, respectively, the sets of permutation sets and scaling sets, and $\tilde{\Omega}$ the random variable of the sorted weights with unit norm. The Bayesian posterior of a neural network $f_\omega$ trained with stochastic gradient descent can be expressed as a continuous mixture of a discrete mixture:*

$$p(\Omega = \boldsymbol{\omega} \mid \mathcal{D}) = |\mathbb{\Pi}|^{-1} \int_{\Lambda \in \mathbb{\Lambda}} \sum_{\Pi \in \mathbb{\Pi}} p(\tilde{\Omega} = \mathcal{T}_p(\mathcal{T}_s(\boldsymbol{\omega}, \Lambda), \Pi), \Lambda \mid \mathcal{D}) d\Lambda. \tag{80}$$

*Proof.* Let $f_\omega$ be a neural network and start with the right term. We can denote by $\tilde{\Pi}$ and $\tilde{\Lambda}$ the only couple of permutations and scale such that the weights $\mathcal{T}_p(\mathcal{T}_s(\boldsymbol{\omega}, \tilde{\Lambda}), \tilde{\Pi})$ are sorted and unit-normed. Given that $\tilde{\Omega}$ is also sorted and unit-normed, $\tilde{\Pi}$ and $\tilde{\Lambda}$ are the only supports of the respective sum and integral. Therefore, we have that

$$|\mathbb{\Pi}|^{-1} \int_{\Lambda \in \mathbb{\Lambda}} \sum_{\Pi \in \mathbb{\Pi}} p(\tilde{\Omega} = \mathcal{T}_p(\mathcal{T}_s(\boldsymbol{\omega}, \Lambda), \Pi), \Lambda \mid \mathcal{D}) d\Lambda = |\mathbb{\Pi}|^{-1} p(\tilde{\Omega} = \mathcal{T}_p(\mathcal{T}_s(\boldsymbol{\omega}, \tilde{\Lambda}), \tilde{\Pi}), \tilde{\Lambda} \mid \mathcal{D}). \tag{81}$$

In parallel, with $\tilde{\omega}$ the sorted unit-normed weights associated to $\boldsymbol{\omega}$, *i.e.*, $\tilde{\omega} = \mathcal{T}_p(\mathcal{T}_s(\boldsymbol{\omega}, \tilde{\Lambda}), \tilde{\Pi})$,

$$p(\Omega = \boldsymbol{\omega} | \mathcal{D}) = p(\tilde{\Omega} = \tilde{\omega}, \tilde{\Pi}, \tilde{\Lambda} | \mathcal{D}) \tag{82}$$

$$= p(\tilde{\Omega} = \tilde{\omega}, \tilde{\Lambda} | \mathcal{D}) p(\tilde{\Pi}) \tag{83}$$

$$= |\mathbb{\Pi}|^{-1} p(\tilde{\Omega} = \tilde{\omega}, \tilde{\Lambda} | \mathcal{D}). \tag{84}$$

Indeed, $\Pi$ and $\mathcal{D}$ are independent, and $p(\Pi)$ is determined by the initialization of the neural network that we suppose layer-wise identically and independently distributed. We obtain the Proposition directly when replacing $\tilde{\omega}$ by $\tilde{\omega} = \mathcal{T}_p(\mathcal{T}_s(\boldsymbol{\omega}, \tilde{\Lambda}), \tilde{\Pi})$. ∎

We then establish the following corollary in the case of a network of fully connected layers. This extends to convolutional networks, paying attention that the identical distributions are multidimensional (and of dimensions the surface of the convolutional kernels).

**Corollary 1.** *Let $f_\omega$ be aneural network containing $L$ layers. For any layer $l \in [\![1, L]\!]$, for any input neuron $i$ and output neuron $j$,*

$$p(\omega_{i,j}^{[l]} | \mathcal{D}) = p(\omega_{0,j}^{[l]} | \mathcal{D}). \tag{85}$$

*Proof.* Let us take an input neuron $i$ and an output neuron $j$. We have, from Proposition 1, and integrating over the whole space except $\omega_{i,j}^{[l]}$ and switching integrals,

$$p(\omega_{i,j}^{[l]} \mid \mathcal{D}) = \int\limits_{\Lambda \in \mathbb{A}} |\Pi|^{-1} \sum_{\Pi \in \mathbb{I}} \int\limits_{\boldsymbol{\omega} \in \Omega \setminus \omega_{i,j}^{[l]}} p(\tilde{\Omega} = \mathcal{T}_p(\mathcal{T}_s(\boldsymbol{\omega}, \Lambda), \Pi), \Lambda \mid \mathcal{D}) d\Lambda d\boldsymbol{\omega} \tag{86}$$

$$p(\omega_{i,j}^{[l]} \mid \mathcal{D}) = \int\limits_{\Lambda \in \mathbb{A}} |\Pi|^{-1} \sum_{\Pi \in \mathbb{I}} \int\limits_{\boldsymbol{\omega} \in \Omega \setminus \omega_{0,j}^{[l]}} p(\tilde{\Omega} = \mathcal{T}_p(\mathcal{T}_s(\boldsymbol{\omega}, \Lambda), \Pi), \Lambda \mid \mathcal{D}) d\Lambda d\boldsymbol{\omega} \tag{87}$$

$$p(\omega_{i,j}^{[l]} \mid \mathcal{D}) = p(\omega_{0,j}^{[l]} \mid \mathcal{D}). \tag{88}$$

∎

### E.5 THE MINIMUM SCALED REPRESENTATION COST PROBLEM

In this section, we develop the formalism leading to the proof of the log-log strict convexity of the minimum scaled representation cost problem (or *scaled-representation* problem). We start by recalling its definition.

#### E.5.1 DEFINITIONS

**Definition E.6** (Minimum scaled representation cost problem)**.** Let $f_\theta$ a neural network and $\bar{\theta}$ its weights without the biases. We define the minimum scaled representation cost problem or *scaled-representation* problem as the minimization of $f_\theta$'s L2 regularization term (the "mass") under invariant scaling transformations. In other words,

$$m^* = \min_{\Lambda \in \mathbb{A}} \left| \mathcal{T}(\bar{\theta}, \Lambda) \right|^2. \tag{89}$$

We also denote the mass of the neural network $f_\theta$ as $m(f_\theta) = \left| \bar{\theta} \right|^2$.

An extension of this problem, referred to as "representation cost" has been tackled by *e.g.* Jacot (2022). However, our slightly more limiting context allows us to derive more powerful results, holding for real-world neural networks.

In this work, we are especially interested in the versions of this problem that are not degenerate. This condition was not given in the main paper for conciseness, but please note that it is not restrictive in real-world cases and could be appropriately handled in the theory with some caution.

**Definition E.7.** We say that the *scaled-representation* problem of the network neural network $f_\theta$ is non-degenerate if, taking any $\Lambda \in \mathbb{A}$, we have that for all scales $\lambda \in \Lambda$, at least one term multiplied by each $\lambda_i^{-1}$ that is non-zero.

Degenerate problems will tend to setting the weights of a layer to the smallest value possible, meaning that their is no solution to the corresponding scaled representation cost minimization. In this case (notably for the ResNet-18) we remove these layers from the computation of the cost and set the corresponding weights to 0.

#### E.5.2 COMPUTING THE MASS OF THE NETWORK

For a neural network without skip-connections and constituted of linear and convolutional layers only, we first provide a definition and use it to devise an important property:

**Definition E.8.** Let $W$ be the weight of a linear or convolutional layer. We denote $M(W)$, the matrix representation of the mass of the layer. For linear layers, we have, with $\circ$, the Hadamard product,

$$M(W) = W \circ W, \tag{90}$$

and for convolutional layers of kernel size $(h, v)$,

$$M(W) = \sum_{i=1}^{h} \sum_{j=1}^{v} (W \circ W) [:, :, i, j]. \tag{91}$$

**Property E.8.** *Let $f_\theta$ be an $L$-layer neural network of weights $\theta = \{W^{[l]}, b^{[l]}\}_{l \in [\![1,L]\!]}$. The mass of the neural network scaled by $\Lambda \in \mathbb{A}$ with $\Lambda = \{\boldsymbol{\lambda}_l\}_{l \in [\![1,L-1]\!]}$ and $\boldsymbol{\lambda}_0 = \boldsymbol{\lambda}_L = \mathbf{1}$ can be expressed as, with $\mathrm{sum}(W)$ the sum of all elements of W,*

$$m\left(f_{\mathcal{T}_s(\theta,\Lambda)}\right) = \sum_{l=1}^{L} \mathrm{sum}\left(\boldsymbol{\lambda}_l^2 \bigtriangledown \left(\boldsymbol{\lambda}_{l-1}^{-2} \rhd M\left(W^{[l]}\right)\right)\right). \tag{92}$$

This result can be extended to networks with batch normalization layers and residual connections. To build the convex problem out of this formula, we use the connection between the line-wise and column-wise products and linear algebra, as presented in E.1.1.

### E.5.3 LOG-LOG CONVEXITY

Finally, we have the following core property on *scaled-representation* problems:

**Proposition 6.** *Let $f_\theta$ be a neural network. The corresponding minimum scaled network representation cost problem is log-log strictly convex, provided it is non-degenerate. As such, the problem is equivalent to a strictly convex problem on $\mathbb{R}^{|\mathbb{A}|}$ and admits a global minimum attained in a single point denoted $\Lambda^*$.*

*Proof.* Let $f_\theta$ be an $L$-layer neural network of weights $\theta = \{W^{[l]}, b^{[l]}\}_{l \in [\![1,L]\!]}$. We recall the corresponding *scaled-representation* problem:

$$P \equiv \min_{\Lambda \in \mathbb{A}} m\left(f_{\mathcal{T}_s(\theta,\Lambda)}\right). \tag{93}$$

With Property E.8, we have, for $\Lambda \in \mathbb{A}$ with $\Lambda = \{\boldsymbol{\lambda}_l\}_{l \in [\![1,L-1]\!]}$ and $\boldsymbol{\lambda}_0 = \boldsymbol{\lambda}_L = \mathbf{1}$,

$$P \equiv \min_{\Lambda \in \mathbb{A}} \sum_{l=1}^{L} \mathrm{sum}\left(\boldsymbol{\lambda}_1^2 \bigtriangledown \left(\boldsymbol{\lambda}_{l-1}^2 \rhd M\left(W^{[l]}\right)\right)\right). \tag{94}$$

Let us now write $\boldsymbol{u}_l = \log(\boldsymbol{\lambda}_l)$ for all $l \in [\![1, L-1]\!]$. This is possible because due to the non-negative homogeneity of the ReLU activation, the $\boldsymbol{\lambda}_l$ were already chosen with elements in $\mathbb{R}_{>0}$. Please note that the need for this change of variables is why $P$ will be referred to as log-log strictly convex. We have,

$$P \equiv \min_{\Lambda \in \mathbb{A}} \sum_{l=1}^{L} \mathrm{sum}\left(e^{2\boldsymbol{u}_l} \bigtriangledown \left(e^{2\boldsymbol{u}_{l-1}} \rhd M\left(W^{[l]}\right)\right)\right) \tag{95}$$

We can convert our operators to products of diagonal matrices and Hadamard products using the formulae from E.1.1. It comes that,

$$P \equiv \min_{\Lambda \in \mathbb{A}} \Big[\mathrm{sum}\left(\mathrm{diag}\left(e^{2\boldsymbol{u}_1}\right) M\left(W^{[1]}\right)\right) \tag{96}$$

$$+ \sum_{l=2}^{L-1} \mathrm{sum}\left(\left(e^{2\boldsymbol{u}_l^\mathsf{T}} e^{2\boldsymbol{u}_{l-1}^{-1}}\right) \circ M\left(W^{[l]}\right)\right) \tag{97}$$

$$+ \mathrm{sum}\left(M\left(W^{[L]}\right) \mathrm{diag}\left(e^{-2\boldsymbol{u}_{L-1}}\right)\right)\Big]. \tag{98}$$

In Equation 98, we see that P is equivalent to a sum of convex and strictly convex functions. Indeed, the product $e^{2\boldsymbol{u}_l^\mathsf{T}} e^{2\boldsymbol{u}_{l-1}^{-1}}$ can be expressed as a matrix containing elements of the form $e^{2(\boldsymbol{u}_l)_i - 2(\boldsymbol{u}_{l-1})_j}$. Therefore, $P$ is equivalent to a strictly convex problem. It comes that $P$ has, at most, a single solution. Moreover, since the elements of the $\boldsymbol{u}_l$ are defined on $\mathbb{R}$ and the inner function is infinite at infinity (as a sum of positive terms with at least one of them diverging due as the problem is non-degenerate), we have that $P$ has a solution, which is unique. ∎

This proof can be easily extended to batch normalization layers that include one-dimensional weights. The extension to residual networks is a bit more complicated, as one should probably resort to graphs to write the equations properly. We then prove the corollary from the main paper.

**Corollary 2.** *Let $f_\theta$ be a neural network trained with L2-regularization. Suppose the mass of $f_\theta$ is not optimal in the sense of the minimum scaled network representation cost problem. In that case, there is an infinite number of equivalent networks $\mathcal{T}_p(f_\theta, \Lambda)$ with training loss lower than that of the original network.*

*Proof.* Let $f_\theta$ be a neural network and denote the training loss by $\mathcal{L}(\theta) = \mathcal{L}_m(\theta) + \mathcal{L}_{L2}(\theta)$ with $\mathcal{L}_{L2}(\theta) = m(f_\theta)$ the L2-regularization term. We have, by definition of the symmetry operator $\mathcal{T}_s$,

$$\forall \Lambda \in \mathbb{A}, \ \mathcal{L}_m(\theta) = \mathcal{L}_m(\mathcal{T}_s(\theta, \Lambda)) \tag{99}$$

Take $\Lambda^*$ as the optimal scaling set, $\mathcal{L}_{L2}(\theta)$ is continuous everywhere in $\mathbb{A}$ as a sum of products of continuous functions and hence continuous in $\Lambda^* \in \mathbb{A}$. Therefore, we can find an open ball $B$ around $\Lambda^*$, where $\forall \Lambda \in L, \ \mathcal{L}_m(\mathcal{T}_s(\theta, \Lambda)) > \mathcal{L}_m(\mathcal{T}_s(\theta, \Lambda^*))$. Hence the corollary. ∎

# F NOTATIONS

We summarize the main notations used in the paper in Table 5.

Table 5: **Summary of the main notations of the paper.**

| Notations | Meaning |
|---|---|
| $\boldsymbol{\omega} \in \Omega$ | $\boldsymbol{\omega}$ is an element of the set of possible weights/hypotheses $\Omega$ |
| $f_{\boldsymbol{\omega}}$ | a neural network of weights $\boldsymbol{\omega}$ |
| $\mathcal{D} = \{(\boldsymbol{x}_i, y_i)\}_{i=1}^{|\mathcal{D}|}$ | the set of $|\mathcal{D}|$ data samples and the corresponding labels |
| $\mathcal{T}(\boldsymbol{\omega})$ | a transformation of the weights $\boldsymbol{\omega}$ that leaves the corresponding neural network functionally invariant |
| $\triangleright$ | the column-wise product between vectors and matrices (see E.2) |
| $\triangledown$ | the line-wise product between vectors and matrices (see E.2) |
| $r$ | the non-linear ReLU activation (the positive part function) |
| $\sigma$ | the softmax activation function |
| $\boldsymbol{\lambda} \in \Lambda \in \mathbb{A}$ | $\boldsymbol{\lambda}$ is a vector belonging to a set of scaling parameters $\Lambda$ among all possible sets of scaling parameters $\mathbb{A}$ for a given neural network architecture |
| $\mathcal{T}_s(\boldsymbol{\omega})$ | a scaling symmetry of the weights $\boldsymbol{\omega}$ of parameter $\Lambda$ |
| $\boldsymbol{\pi} \in \Pi \in \mathbb{\Pi}$ | $\boldsymbol{\pi}$ is a matrix belonging to a set of permutation matrices $\Pi$ among all possible sets of permutation matrices $\mathbb{A}$ for a given neural network architecture |
| $\mathcal{T}_p(\boldsymbol{\omega})$ | a permutation symmetry of the weights $\boldsymbol{\omega}$ of parameter $\Pi$ |
| $P_n$ | the set of permutation matrices of size $n$ |
| $\mathcal{L}(\boldsymbol{\omega})$ | training loss of a neural network of weights $\boldsymbol{\omega}$ |
| $m(f_{\boldsymbol{\omega}})$ or $m(\boldsymbol{\omega})$ | The "mass" or representation cost of the hypothesis $\boldsymbol{\omega}$ (see E.6) |
| $W, b$ | The weights and biases of a layer. |
| diag | a function that maps a vector to its corresponding square diagonal matrix |
| $\mathcal{H}$ | the entropy |

