# OpenReview forum: "A Symmetry-Aware Exploration of Bayesian Neural Network Posteriors"
_ICLR.cc/2024/Conference — ICLR 2024 poster_

### Official Review · Reviewer_hdLm · 2023-10-30

**Soundness:** 3 good
**Presentation:** 3 good
**Contribution:** 3 good
**Rating:** 6
**Confidence:** 4

**Summary:**

The paper formalizes permutation and scaling symmetries in Bayesian neural networks (BNNs), proposing to express the posterior as a mixture density with components corresponding to symmetric copies of the underlying identifiable model. Investigating scaling symmetries, the authors find the minimization of an L2 term over scaling transformations to be a convex problem. In an extensive benchmark study, they shed light on the benefit of diversity realized by multi-mode methods, the opacity of the link between in-distribution diversity and out-of-distribution (OOD) performance, and the frequency of weight permutation during training. A large codebase is to be released.

**Strengths:**

* [S1] **Originality**. Expressing the BNN posterior as a mixture w.r.t. two types of symmetries seems novel and compelling, as does posing the minimization problem of L2 regularization over scaling transformations. The empirical study of the frequency of weight permutations also adds an interesting new perspective.
* [S2] **Formalization**. Introducing symmetry operators and the mixture posterior is beneficial to the field of BNN symmetries that so far lacks a comprehensive language.
* [S3] **Contextualization**. The authors relate established concepts like functional collapse and OOD generalization to the quality of the posterior approximation (in the light of symmetries).
* [S4] **Empirical evidence**. Substantive experimental results are provided and the authors are to release a broad benchmark dataset.

**Weaknesses:**

* [W1] **Lack of clarity**. There are some statements whose scope/explanation is not quite clear to me:
  * “The scaling operation (...) is a symmetry for all neural networks” (Property 3.2); “neural networks seem to always be subject to scaling symmetries” – I feel this should be restricted to ReLU-type nonlinearities. Why does this hold in general?
  * More generally about the *min-mass* problem: What, exactly, is the implication? Do we impose a second, ex-post minimization problem over the solution of standard training? How comes the problem is convex if existing literature on [representation costs](https://openreview.net/pdf?id=6iDHce-0B-a) (which tackles the same problem and is not cited?) turns out not to be a convex problem?
  * “All the layer-wise marginal distributions are identical” (Proposition 1) – how so?
* [W2] **Conflating Bayesian and risk-minimization perspectives**. The paper is supposed to be about BNN posteriors, yet most claims and analyses focus on standard network training. It would be useful to elaborate more on both perspectives, or reconcile them. E.g., what do Definition 3.3 / Proposition 2 mean for a Gaussian prior equivalent to L2 regularization? As-is, the impact of the paper for actual Bayesian inference remains unclear.
* [W3] **Unclear baseline / omissions in experiments**. Somewhat related to [W2]: I don’t quite understand how the “true” posterior distribution, w.r.t. which the MMD is reported, is computed. How, exactly, is the RKHS constructed? Also, I feel that the Bayesian perspective is quite under-represented with only one MCMC method (the other methods present have no guarantees of retrieving the true posterior).
* [W4] **Minor points**. Not relevant for my score, just to point out:
  * Abbreviations are not always handled consistently (e.g., HMC not introduced in 2; “deep neural networks” in 3.2 despite abbreviation introduced before).
  * Figure 1 should have axis labels.
  * Figure 3 (right) has very low contrast.
  * P. 2, second-to-last line: “functionnally” (should be “functionally”)
  * P. 4, first line: “influencial” (should be “influential”)
  * P. 4, Property 3.1: “all neural network” (should be “all neural networks”)
  * P. 5, second line: “perceptions” (should be “perceptrons”)
  * P. 7, second line: “efficacity” (should be “efficacy”)
  * P. 8, third line under 5: “weights permutations” (should be “weight permutations”)
  * $\omega$ is not always bold.
  * Describing Laplace methods as “characterizing the posterior distribution” is pretty vague.

**Questions:**

* [Q1] Could you elaborate on the role of $\theta$ in Property 3.1, Property 3.3?
* [Q2] What kind of objects are $\Pi$ and $\mathbb{\Pi}$?
* [Q3] Does the posterior expression in Eq. 8 allow for other types of symmetries?
* [Q4] What is your intuition on scaling symmetries being always present even with weight decay? Could this be because an infinite number of scaling transformations exists that leaves the (optimal) L2 norm unchanged?
* [Q5] Why are single-mode methods competitive in terms of ECE? Is there a tendency towards either over- or underconfidence?
* [Q6] Could you provide some intuition about the similarity of two networks having a MI of, say, 0.1?
* [Q7] What is your idea behind connecting functional collapse to OOD performance?

---

> ### Author Response · Authors · 2023-11-14
>
> Dear Reviewer **hdLm**,
>
> We thank you for your detailed comments, relevant questions, and help chasing the remaining typos.
>
> We understand that you recommend a weak rejection based on several purportedly inadequate aspects of our work, namely the lack of clarity, the link with Bayesian inference, and our choice of baselines.
>
> - **Concerning the lack of clarity [W1]**, we thank the reviewer for pointing out that our phrasing in Property 3.2 is imprecise. Since we cannot develop the full formalism in the core paper, we decided to restrict it to two-layer perceptrons. However, symmetries are not restricted to this case, and the sentence highlighted by the reviewer aimed to convey that our work tackles other architectures. In this paper, we focus on ReLU-activated neural networks and confirm that properties on the scaling symmetries only hold in this case. We have rephrased this sentence in the updated document for clarity.
>
> - **Continuing on [W1]**, the main implication of the min-mass problem is, to our understanding, that we cannot rule out the impact on scaling symmetries when focusing on the posterior. Supposing that neural networks empirically converge to the minimum of this problem, it would mean that scaling symmetries do not play any role on the posterior. The difficulty with scaling symmetries is that, contrary to permutation symmetries, we cannot make their role explicit, as they actively interact with the main training loss. We do not think that we need to run another optimization since this would only result in a lower regularization term, which is without impact on the actual performance of the models. However, it is unclear for now, but minimizing this term could have an impact on fine-tuning, for instance.
>
> We thank the reviewer for the exciting reference on representation costs that we missed. We are currently reading it in detail; however, our hypothesis is that - in our paper - we reduce the size of the domain of the variables (to strictly positive vectors for the lambdas) and exhibit the explicit impact of scaling symmetries on the weights. It seems that Arthur Jacot’s work is somewhat more general, tackling all possible weight symmetries, whereas we restrict ourselves to scaling symmetries (you may see the work of Rolnick et al. [**2**] on the other “hidden” symmetries for further details). For completeness, our sketch of proof of the log-log strict convexity of the min-mass problem is available in Appendix C.7.3.
>
> - **Continuing on [W1]**, given that the posterior is a mixture of all permutations, marginalizing any neuron in a given layer will lead to the same distribution. We can detail this point if it is not clear or convincing enough.
>
> - **Regarding the Bayesian perspective [W2]**, we chose to focus our paper on the posterior, a part of the Bayesian framework that has been recently exposed as one of its most important aspects (e.g., in [**1**]) and that remains not fully understood. Concerning Definition 3.3, the fact that modern neural networks undergo scaling symmetries that modify the L2 regularization term without functional changes can lead us to slightly update our interpretation of the regularization term as Ockham’s razor. Notably, this again stresses that the mass of the network is a very simplistic heuristic of the complexity of the resulting model.
>
> - **Concerning the unclear baseline/MMD [W3]**, we are rewriting the paragraph on the Maximum Mean Discrepancy to clarify this part of the paper. We will come back to you when this is done. In the meantime, we recall that we independently train 1000 DNNs that we use as target samples for the MMDs that we compute on 20 different kernels, as in Schrab and Gretton et al. [**4**]. We provide different statistics on these kernels in Table 2.
>
> - **Continuing on [W3], for the baseline omissions**, we agree that we present only one strictly MCMC method and that it is not fully satisfactory. However, we tackle real-world datasets/architectures in this work, and many MCMC methods cannot scale to these problems. We propose to add stochastic Langevin dynamics [**5**] in the next update of our paper on MNIST. We will come back to you when integrated into our work.

---

> ### Author Response · Authors · 2023-11-14
>
> Concerning the questions,
>
> - **[Q1]** $\theta$ is essential in these properties as the second layer allows to compensate for the permutation and scale that have transformed the previous layer. More details on this are provided in Appendix C.1.
>
> - **[Q2]** Let us take a neural network with $k+1$ layers, $\Pi$ is a set (ordered) of $k$ permutation matrices. The size of each permutation matrix is adapted to the corresponding hidden dimension of the network at this layer. In our paper, we chose to formalize permutations with matrices for clarity. Of course, they are implemented by operations on indices in our codes for computation efficiency. $\mathbb{\Pi}$ is the set of possible sets of permutation matrices for a given architecture. To further improve the clarity, we added a Table including the main notations of the paper at its end.
>
> - **[Q3]** Equation (8) is not very precise on the role of the scaling symmetries on the posterior. Yet, we still exhibit the pattern of the impact of scales on the weights (through the operator $\mathcal{T}_s$). To our knowledge, this is impossible with hidden symmetries (Rolnick et al. [**2**]). Right now, it seems that it would be difficult to explicit these symmetries in the formula.
>
> - **[Q4]** There is clearly a linear path with decreasing loss between the optimal solution and the converged network. We have several hypotheses concerning why networks do not seem to converge to the minimum mass/representation cost. The first one is that the solution to the min-mass problem varies a lot between successive steps of SGD, some functionals being easier to represent with a lower norm than others. As such, it would be impossible to converge to the minimum as it would require this minimum to stay more or less the same between successive iterations. The second one is that this path is not convex but log-log convex and, therefore, exhibits a flatness that may hinder the convergence towards the minimum. We intend to test both hypotheses and get back to you. Finally, we are not sure to fully understand your hypothesis since - following our reasoning - the problem admits a single solution, being strictly convex after the change of variables.
>
> - **[Q5]** This remark is shared with reviewer **Mjnv**. Considering the criticisms on the equal-width binned ECE, we have added the adaptive calibration errors (ACE) to Table 1. We show that OptuNets - very shallow networks - exhibit underconfident behavior and provide the calibration plots in the case of the BNN and Multi-BNN in Appendix B.4. We can provide more plots corresponding to other methods and architectures if desired.
>
> - **[Q6]** Concerning the value of the mutual information, we are preparing an experiment and will get back to you.
>
> - **[Q7]** The mutual information - which emerges when the predictions of elements in the ensemble differ - is often used to quantify epistemic uncertainty, although its relevance has also been disputed [**3**]. In this last section, we uncovered that, at least in our setting, couples of independently trained neural networks tend to exhibit similar quantities of diversity on in-distribution and out-of-distribution data. If this were not the case, ensembles could collapse in the function space despite their weight-space differences. In this case, we would not gain much from ensembling. These experiments tend to show that this is not a risk in practice.
>
> - [**Figures**] We acknowledge your comments on the figures. They will be updated in the next version of the document.
>
> We hope that our answer helped make our work more convincing, and we remain available if you feel that your concerns have not been fully addressed or if you have other concerns not mentioned in the first review.
>
> Sincerely,
>
> The authors
>
>
> ### References
>
> [**1**] Andrew G Wilson and Pavel Izmailov. Bayesian deep learning and a probabilistic perspective of generalization. In NeurIPS, 2020.
>
> [**2**] David Rolnick and Konrad Kording. Reverse-engineering deep relu networks. In ICML, 2020.
>
> [**3**] Wimmer, Lisa, et al. "Quantifying Aleatoric and Epistemic Uncertainty in Machine Learning: Are Conditional Entropy and Mutual Information Appropriate Measures? In UAI 2023.
>
> [**4**] Schrab, Antonin, et al. "MMD aggregated two-sample test." JMLR 2023.
>
> [**5**] Li, Chunyuan, et al. "Preconditioned stochastic gradient Langevin dynamics for deep neural networks." In AAAI. 2016.

---

> > ### Comment · Reviewer_hdLm · 2023-11-16
> > **Reponse to Author Comments**
> >
> > Thank you for addressing my concerns and making changes to the manuscript. Below my responses to the points that have not been fully clarified:
> > - **[W1] min-mass problem** Do I understand correctly that Def. 3.3 is more about the existence of this problem and its minimizer, but there is no guarantee in practice that the optimization procedure will actually converge to $m^\ast$? Furthermore, I do not agree that the magnitude of the regularization term “is without impact on the actual performance”, why would anyone bother with regularization if it did not matter for generalization?
> > - **[W1] layer-wise marginals** Thank you for explaining this. However, I’m not still quite sure – your point makes sense w.r.t. equiprobable permutation symmetries, but how does this hold up in the presence of scaling symmetry, where you state yourselves: “The equiprobability of permutations in Equation ... We have no such result on scaling symmetries since the standard L2-regularized loss is not invariant to scaling symmetries (contrary to permutations)”?
> > - **[W2]** Personally, I don’t think that an isolated view on the posterior, without taking the other components of the Bayesian paradigm into account, can provide a comprehensive picture. You set out to obtain fundamental insights by considering symmetries, which are arguably tightly connected to the choice of prior (as you acknowledge by putting so much emphasis on regularization), but end up discussing mainly MAP convergence and approach the experiments from a not very rigorously-Bayesian angle (e.g., inferring a ground-truth posterior from a deep ensemble). I feel the argumentation is not as well-rounded and holistic as it could be, similar to what reviewers QYDw and 24pF have remarked.
> > - **[W3] baselines** I agree that experiments in the Bayesian context become expensive very quickly. However, I would argue that the claims that can be made are then limited by the amount of evidence that can be provided. Real-world experiments certainly have their merit, but having additional toy-sized problems that allow for more accurate inference might help to bolster the empirical support and insights.
> > - **[Q1]** So $\theta$ is simply the weight matrix associated with the previous layer?
> > - **[Q4]** Thank you for elaborating, and apologies for my confusing remark about non-uniqueness of the solution, which cannot happen with a strictly convex problem. I think that your hypotheses would also add value to the discussion in the main paper.
> > - **[Q7]** Do I understand correctly that you assume functional collapse to be detrimental to OOD detection because signaling for a sample to be OOD is typically connected to disagreement / diversity? If so, then I have two follow-up questions:
> >     1. What, exactly, does the mutual information refer to? As a measure of epistemic uncertainty, it is taken to be the MI between the distribution of target labels and the model parameters. However, when you speak of a “pairwise MI” between two DNNs, are you computing the MI between their respective predictions?
> >     2. How are the results in 5.1 connected to symmetries? I fail to see how they “indicate that the complexity of the posterior is orders of magnitude higher than what we understand when considering symmetries”.

---

> ### Author Response · Authors · 2023-11-18
>
> Dear Reviewer **hdLm**,
>
> We thank you very much for your reactivity and your feedback. Here are our answers to your remaining concerns:
>
> - **[W1]** Yes, your understanding is entirely correct. Although the regularization problem on the space of scaling symmetries is strictly convex after a change of variables, it is never the problem that the neural network actually tries to solve. Please note that it is possible to propose an algorithm to solve this problem (convex solvers can solve log-log convex problems efficiently) but only when discarding the cross-entropy part of the loss. This is what we do in Figure 2.
>
> We are sorry if we were not clear enough about the role of regularization. We fully agree that regularization is essential in practice when supplemented with cross-entropy. It biases the optimization of the neurons towards networks with lower representation cost/mass, leading to potentially simpler and more generalizable functions. However, solving the regularization problem - alone and after training with both the cross-entropy and the regularization - on the scaling symmetries will leave the corresponding function unchanged. Therefore, it does not impact the evaluation performance, whether in-distribution or out-of-distribution. However, this could impact the performance of fine-tuned networks starting from a network of lower mass.
>
> - **Equivalence of marginals [W1]**. This comes from the fact that the role of the scaling operator is entirely independent of the chosen permutation, the mass of the network being independent of the order of the weights in their layer. Then, the integral and the sum can be exchanged without further assumptions, and the order of the terms of the sum does not impact the result. This result is empirically supported by experiments on MLPs in Appendix D.5.
>
> - **Estimating the Bayesian posterior with DE [W2]**. We thank the reviewer for their detailed comment. We would like to start by pointing out that the fact that Deep Ensembles (DE) is not rigorously Bayesian is still disputed [**6** vs. **5**] and especially in high-dimensional settings [**1**]. Indeed, [**1**] is a crucial recent paper that links deep ensembles and Bayesian methods by reformulating the learning process as a convexified optimization in the space of probability measures. In this framework, DE learning comes out as a particular case of minimizing an unregularised (nonconvex) functional. However, the authors highlight that in practice, DE performs at least comparably to regularized minimizers in high-dimensional settings; this is in agreement with our (and community's) longstanding apprehension about its abilities and justifies the proposed practical construction of the ground-truth posterior. The details are provided in Section 5 of [**1**]. Moreover, even HMC, the *gold standard* in posterior estimation, is known for its inability to jump between modes [**7**] - even though [**5**] has shown that it can have good properties in practice -  which, we think, is a problem when trying to estimate extremely multi-modal distributions.
>
> On the contrary, we show - in Section 5.1 (see **[Q7]** in the next message) -  that ensembles do not suffer a collapse in the function space (and hence not in the parameter space either) in very high-dimension, reducing the impact of the regularizations (repulsiveness), supposed to make them fully Bayesian. The conclusion of [**1**] also supports this claim. As expressed by Reviewer **QYDw**, we argue that this aspect does not disqualify our paper; rather, we provide a discussion detailing this controversy in Appendix D.2. If the reviewer feels the contrary, we would be very grateful for them to elaborate on their point of view.
>
> - **Incorporation of priors in our work [W2].** We fully agree with the reviewer concerning the significance of the priors, which has been established in previous works (e.g., [**2**]), and their relevance in the Bayesian framework. However, we are not sure to fully understand the reviewer’s point. Would it be possible to have more details? For instance, we agree that we could include a remark and a discussion on the impact of the chosen prior on our theoretical results. While standard priors would not impact permutation symmetries (the regularization is often identical network-wise and, therefore, layer-wise), we agree that we could detail their impact on scaling symmetries. Would the reviewer also be interested in the empirical impact of the prior on the symmetries?
>
> - **[W3]** We agree with the reviewer that small-scale experiments can be of interest to empirically validate theoretical results. As for now, we have already implemented some experiments on toy-sized models. Additionally to the first figure, whose corresponding experiment is detailed in Appendix A, the property referred to in **[W1]** is empirically backed by results on 1-hidden layer MLPs in Appendix D.5. (To be continued in the next message)

---

> > ### Comment · Reviewer_hdLm · 2023-11-22
> >
> > Thank you, again, for addressing my comments so swiftly and in great detail - so far, I have no further questions or comments.
> > I feel that the proposed changes provide an upgrade to the manuscript and will raise my score accordingly.

---

> > > ### Author Response · Authors · 2023-11-22
> > >
> > > Dear Reviewer hdLm,
> > >
> > > We are grateful for your commitment, precise feedback, and for the fruitful discussion that helped us improve the paper.
> > > We will continue to improve our work with your suggestions in mind.
> > >
> > > Sincerely,
> > >
> > > The authors

---

> ### Author Response · Authors · 2023-11-18
>
> **Continuing [W3]** However, we argue that conducting real-world experiments is essential to avoid the emergence of a substantial gap between a community dedicated to small-scale experiments and those engaged in handling increasingly complex statistical objects, such as foundational models. This is the objective of our paper. We believe both approaches are similarly legitimate, but works should be focused on bridging the gap despite the corresponding inherent difficulties. Yet, we will try to incorporate a toy example to demonstrate further the relevance of our methods as well as improve the visualization side of our work in the next few days.
>
> And concerning the questions:
>
> - [**Q1**] $\theta$ corresponds to the weights of the second (output) layer. Indeed, in the core of the paper, we develop the formalism for simple one-hidden layer perceptrons with ReLU activations. In this case, $\omega$ corresponds to the weights of the first layer and $\theta$ to the weights of the second (we are sorry to reuse $\omega$ in this context). We transform the weights of the two layers using the operators on the rows and the columns from Definition 3.2. The details, the proof, and the extension of this property to layers with biases are available in Appendix C.1.1.
>
> - [**Q7**] Yes, exactly. Concerning the mutual information, we compute its discrete/empirical version (which is also the generalized Jensen-Shannon divergence). The formula, for a given sample $(\mathbf{x}, y)$ of the joint distribution is computed as the entropy of the mean over the $M$ predictors of weights $\omega_m$ of their predicted categorical distribution minus the mean over the predictors of the entropies of each predicted categorical distribution (as in, e.g., [**3**] Eq. 7 and 8, [**4**] Eq. 21):
>
> $$\mbox{MI} =\mathcal{H}\left(\frac{1}{M}\sum\limits_{m=1}^{M}P_{\omega_m}(\hat{y}|\mathbf{x},\mathcal{D}) \right) - \frac{1}{M}\sum\limits_{m=1}^{M} \mathcal{H}(P_{\omega_m}(\hat{y}|\mathbf{x}, \mathcal{D}))$$
>
> In the case of Section 5.1, we use this formula with M=2 and compute it for 500,000 pairs of trained ResNet-18. We add a section on the mutual information in Appendix D.8. Please note that the fact that we use MI with multiple categorical distributions complicates its interpretation within the information theory framework and explains our difficulty in answering [**Q6**], but we will get back to you.
>
> - **Functional collapse [Q7].** We have rewritten and reorganized this part of the paper. To clarify this point, let us suppose that we take a couple of models that have converged to symmetric weights. By definition, these models are the same in the function space. Taking any input data from a dataset, the mean of their predictions will be exactly their predictions. Therefore, the mean of the entropy of their predictions will be the same as the entropy of the mean of their predictions. It follows that they have a null mutual information. We can extend this reasoning to approximately-symmetric networks, which would have very low MI.
>
> Now, we also know that the usual diversity between members in an ensemble is satisfactory (otherwise, nobody would use ensembles as they would require training a lot of networks and discard a part of them), our experiments show that this diversity has, in fact, very low variance: pairs of networks exhibit roughly the same quantity of functional diversity. This means that despite the huge amount of functionally equivalent local minima, it is very unlikely that two networks will fall into these corresponding minima. This emphasizes the richness of the posterior since many (most?) of its modes have pairwise functional diversity and are, therefore, not symmetric.
>
> We hope that we helped, and we thank you again for the relevance of your review. As previously, we remain available to clarify any other questions or continue the exciting discussion on some of your concerns that would remain, for instance, on the role of the prior.
>
> Sincerely,
>
> The authors
>
> ### References
>
> [**1**] Wild, Veit David, et al. "A Rigorous Link between Deep Ensembles and (Variational) Bayesian Methods." In NeurIPS 2023.
>
> [**2**] Fortuin, Vincent. "Priors in bayesian deep learning: A review." _International Statistical Review_. 2022.
>
> [**3**] Smith, Lewis, and Yarin Gal. "Understanding measures of uncertainty for adversarial example detection." In UAI 2018.
>
> [**4**] Hüllermeier, Eyke, and Willem Waegeman. "Aleatoric and epistemic uncertainty in machine learning: An introduction to concepts and methods." Machine Learning. 2021.
>
> [**5**] Wilson, Andrew G., and Pavel Izmailov. "Bayesian deep learning and a probabilistic perspective of generalization." In NeurIPS.
>
> [**6**] D'Angelo, Francesco, and Vincent Fortuin. "Repulsive deep ensembles are bayesian." In NeurIPS. 2021.
>
> [**7**] Mangoubi, Oren, Natesh S. Pillai, and Aaron Smith. "Does Hamiltonian Monte Carlo mix faster than a random walk on multimodal densities?." arXiv preprint arXiv:1808.03230 (2018).

---

> > ### Comment · Reviewer_hdLm · 2023-11-21
> >
> > Again, thank you for addressing my concerns in so much detail. As is reflected in my comments below, I believe your work contains quite some interesting findings, but a more explicit discussion is necessary in some places to make the argument more convincing.
> > * **[W1]** Thank you for elaborating on this. I think the manuscript would benefit from the explanations you provided to my questions.
> > * **[W2]** I’m aware of there being quite some proponents of this argument regarding DE, and you make a point here. The explicit discussion in D.2 of your grounds to choose this type of ground truth is decidedly helpful.
> > * **[W2] priors.** Yes, some discussion on the impact of the chosen prior on your theoretical results would certainly be welcome to readers from a more rigorously Bayesian background, providing a more comprehensive perspective.
> > * **[W3]** I agree that the question of real-world vs toy examples is a somewhat philosophical one – thank you for taking up my suggestions of also providing small-scale examples, analyzing the posterior from multiple angles is definitely valuable.
> > * **[Q7]** Thank you for clarifying this (also in D.8 and 5.1). Do I understand correctly that your argument is the following: the MI analysis shows that, despite the fact that we would expect a substantial number of ensemble members to approach one of the symmetrical modes and thus collapse to the same functional mapping, this does not happen in practice, suggesting that symmetries are not enough to explain (a lack of) diversity?

---

> ### Author Response · Authors · 2023-11-22
>
> Dear Reviewer **Hdlm**,
>
> Again, we thank you for your feedback, which strengthened our paper and helped improve its clarity. As for the comments on your last answer:
>
> - **[W1]** We definitely agree with you. We have added section E.4, which briefly summarizes our discussion on the minimum-mass problem and its implications (or not) on the generalization properties of the DNNs.
>
> - **[W2] Priors.** We also agree with you on this point. We now start the paper (Section 3) by explicitly stating our hypothesis on the prior and pointing towards a new section dedicated to the priors. This section contains a brief background on MLE and MAP and details the impacts of the priors on our work. For instance, we make explicit that the min-mass problem is exclusively relevant for Gaussian priors and would be non-convex for Laplace distributions for instance. We also briefly present the role of the prior in the complexity of the relationship between the posterior and the loss landscape. Finally, we aim to continue improving our paper before and after the camera ready, and we will perform experiments showing the impact of the chosen variance (weight decay) on the distance to the solution of the min-mass problem. Unfortunately, this has not been possible until now due to the number of questions and despite our best efforts.
>
> Although briefly mentioned in **[W1]** and **[W2]**, we are getting back to you as promised concerning the prior, the loss landscape, and the posterior estimation using deep ensembles.
>
> - **Concerning estimating the posterior using independently trained checkpoints** **[W2]**. We are pleased that you found the former Section D.2 insightful. We have now upgraded this discussion to a full section, which you will find as Appendix B. We improve our small literature review on the controversy around deep ensembles being Bayesian and posterior estimation in general; we compare the computational needs required for estimating the Bayesian posterior deep-ensembles-style and with HMC and explore the potential limitations of full-batch HMC in practice. Notably, we show that HMC breaks the corollary of Proposition 1, which holds in the general setting to which [**1**]’s ResNets are submitted.
>
> - **Concerning the priors and the loss landscape**. We have developed a new section, E.2, discussing the link between our work and the loss landscape. We briefly highlight the key impact of the prior on the complexity of the relationship between the loss landscape and the posterior. In this section (E.2.2), we also test the hypothesis we mentioned in our response to **[W2]** in our previous message. We post-train OptuNets for 5 epochs and measure the scalar product between the gradient at each step and the unit vector going from the current network to the corresponding minimum-mass network. We show that this scalar product oscillates with a mean slightly above 0 yet seems negligible compared to the noise of the SGD. Again, this calls for a sensitivity analysis on the variance of the prior: experiments that we have added to our list but will not be able to deliver on time.
>
> - **[Q7]** Yes, your understanding is entirely correct. We have updated section 5.1 to reflect our discussion and improve the clarity of our interpretation. We may return to it later to further improve its clarity and add some details in the Appendix.
>
> We have started a toy example in Section E.5 that we intend to improve in the following hours and after the end of the discussion. We aim to highlight the multimodality of the posterior distribution and provide empirical confirmations to Proposition 1. It could also help us better understand the distances that we are using. If fruitful, we will develop the corresponding subsection to a full section in the Appendix.
>
> We thank you again for your feedback, and we remain available for any other advice you would have for us or concerns that could arise.
>
> Sincerely,
>
> The authors
>
> ### Reference
>
> [**1**] Izmailov, Pavel, et al. "What are Bayesian neural network posteriors really like?." In ICML. 2021.

---

### Official Review · Reviewer_24pF · 2023-10-31

**Soundness:** 2 fair
**Presentation:** 2 fair
**Contribution:** 1 poor
**Rating:** 3
**Confidence:** 4

**Summary:**

The paper sets out to study the Bayesian posterior of deep neural networks. It hypothesizes that the quality of uncertainty quantification made possible by a Bayesian treatment is reduced due to the presence of scaling and permutation symmetries in the weights of the network. It presents a formal definition of permutation and weight symmetry. It evaluates the quality of typical Bayesian methods for estimating the posterior distribution on computer vision tasks and presents experimental results on “functional collapse” and weight permutation events in the process of training.

**Strengths:**

The paper introduces a formal treatment of scaling symmetries, and introduces the notion of a `minimum scaled network`. The work demonstrates that the optimization problem to reach this minimum-scaled network is convex. They then demonstrate that on their specific small mnist model (OptuNet), a specific neural network optimization scheme with weight decay does not reach the optimum (in terms of L2 loss) achieved by further convex optimization of the L2 norm under scaling transformations.

**Weaknesses:**

The paper as a whole presents a meandering set of concepts and results, without drawing conclusions or presenting insights into further study of Bayesian method design. This is apparent from the conclusion section, which is vague and raises questions: what are the actual insights provided by considering symmetries? What questions and hypotheses would you consider in future work? What was the “real impact of scaling symmetries”?

The core experiment demonstrating that L2 regularisation does not remove weight scale symmetries is presented in Figure 2, but this involves a single, custom network architecture under a single specific optimization scheme. It would be insightful to study this under various benchmarks and models to demonstrate that this effect is consistent and that the difference between the minimum reached by SGD with L2 regularisation and the minimum reached by further convex optimization is significant and actually affects the quality of Bayesian posteriors.

The experiments presented do not provide evidence for the claim presented in the method section: "SYMMETRIES INCREASE THE COMPLEXITY OF BAYESIAN POSTERIORS" (see questions below). Nor do they study interventions informed by the theory to evaluate the benefit of the formal treatment of scale and permutation symmetries.

The reported accuracies in Table 1 are low. While this is understandable for the purposefully restricted OptuNet, a vanilla single-mode baseline resnet-18 model on cifar-100 can reach 75% accuracy without any Bayesian treatment, and with some tuning 79%. (https://paperswithcode.com/sota/image-classification-on-cifar-100 https://huggingface.co/edadaltocg/resnet18_cifar100). It seems the results and the presented checkpoint dataset represent an artificially underfitted model space. If the aim is to improve Bayesian deep learning for a real-world setting, it might be beneficial if the models studied are optimized to a reasonable performance.

**Questions:**

- What do you mean by ‘regroup some points around the modes’ in section 3.1? What causes this effect and how is it problematic?
- How is the “target distribution with 1000 checkpoints” for table 1’s MMD scores established? What method is used to determine this distribution?
- I am not sure what to take away from Table 1 and section 4.1-4.3. The study of uncertainty quantification performance of these methods has been done before, how does this experiment connect to the formal study of scale and permutation symmetries?
- Similarly, how does the study of ‘functional collapse’ in Figure 3 and section 5.1 suggest that  “complexity of the posterior is orders of magnitude higher than what we understand, taking symmetries into account.” I don’t follow how symmetries were taken into account in this study, and what to take away from this to improve the Bayesian treatment of neural networks. It appears to me that the symmetries in the neural networks studied here were left untreated, and no interventions were made to reduce the complexity of the Bayesian posterior through these symmetries.
- For section 5.2, what is the insight gained from the study of weight permutation? Is “a variation in the Πs” somehow problematic? How does this inform the design of Bayesian deep learning methods?

Typo/nits:
- “non-functionnally”  (bottom page 2)
- “influencial” (start section 3)

---

> ### Author Response · Authors · 2023-11-14
>
> Dear Reviewer **24pF**,
>
> We appreciate the detailed feedback and acknowledge your inclination toward a rejection based on the following points:
>
> 1. the lack of conclusive insights on the effect of weight-space symmetries on the posterior,
>
> 2. the generalizability of the experiments concerning the min-mass problem,
>
> 3. the limited empirical demonstration that the symmetries increase the complexity of Bayesian posteriors,
>
> 4. the below-standards performance of our ResNet-18 trained on CIFAR-100 in section 4.3.
>
> Here are our responses to each of the mentioned points:
>
> - **Concerning the lack of conclusive Insights [W1]:** We acknowledge your feedback on the clarity and would like to provide additional clarification. Our paper aims to explore all the different aspects of the interaction between weight-space symmetries and the posterior, all of which remained vastly unchartered. Our work introduces several novel contributions: firstly, we are among the pioneers in sharing the ground truth of DNN posteriors; secondly, we contribute to the field by establishing equation 8 (without debatable hypotheses), which provides a first explanation of the links between posterior and weight-space symmetries and explains the patterns that practitioners encounter when visualizing posterior estimations. Thirdly, we explore the connection between the quality of posterior estimation and the effectiveness of uncertainty quantification by comparing distributions of extremely high dimensions. Finally, we provide in Section 5.1 a result that is important for the practitioners and which updates (upwards) the confidence that we can grant to the ubiquitous _deep ensembles_.
> In contrast to a large part of the papers in the surrounding field, our analysis tackles real-world architectures. We would like the reviewer to keep in mind that implementing experiments on the posterior distributions of these models is an extremely complex task. For example, it is impossible to obtain direct estimations of the value of this posterior in a bin of weight space, given that we cannot choose where the model will converge. This is an important limiting factor for our experiments, which may be important to consider.
>
> - **On the generalizability of the experiments of the min-mass problem [W2],** we agree that the impact of empirical experiments on the minimization of the regularization term on the scaling symmetries suffers from their restriction to our custom architecture. This is why we plan to incorporate results on ResNet-18 before the end of the rebuttal and eventually add experiments to check the impact of the value of the weight decay. We will get back to you when we have finished. These experiments were not included in the original version of our work due to the complexity linked to taking residual connections into account when solving the log-log convex problem.
>
> - **Concerning the limited empirical demonstration that the symmetries increase the complexity of Bayesian posteriors [W3],** we understand your concern that the MMD does not definitively prove that removing symmetries improves posterior estimation given the challenges of high-dimensional distributions. Again, experiments on the posterior of modern networks are particularly complex and were previously mostly unchartered. However, we will incorporate more straightforward distances into our work to improve the clarity of this side of the paper.
>
> - **Regarding the performance of our ResNets [W4],** our initial accuracy was lower due to our focus on obtaining a pure posterior unaffected by data augmentations. After a code review, we noticed an issue with the batch normalization. We have updated the results and are achieving close to 80% accuracy with deep ensembles. Unfortunately, it is unclear whether achieving very high single-model performance (79%) without data augmentations is possible. We hope that this corrected performance will be satisfactory.
>
> More specifically, for your questions:
>
> - **Concerning the sentence on the high-level impact of scaling symmetries [Q1],** we thank the reviewer for highlighting that it is unclear. We meant that removing the scaling symmetries tends to regroup the points around the main modes. Equation (8) sheds light on this matter, showing that functionally equivalent networks can be dispersed due to these symmetries. Formally, removing the symmetries means removing the dispersion due to the continuous mixture over the possible scales. We do not argue that this is a problem, but rather information that the practitioners could be interested in to achieve a more in-depth comprehension of deep learning models.

---

> ### Author Response · Authors · 2023-11-14
>
> - **Regarding the target distribution for the MMD [Q2],** we will include a detailed explanation of our MMD calculation in the next version of the document. To provide a clearer understanding, the MMD involves mapping the checkpoints into a Hilbert Space, computing the empirical mean, and subsequently determining the distance between these empirical means. Notably, the empirical MMD exhibits convergence in probability at a rate of O((m + n)^(-1/2)) to its true value, where $m$ and $n$ denote the number of samples in each set. This convergence demonstrates the reliability and consistency of the MMD.
> As explained in Section 4.1, on which we are currently working, we use 1000 independently trained checkpoints as the target sample. Please note that the results of all methods are based on networks that were not used in the target samples to avoid biasing the MMDs.
>
> - **On the connection of symmetries and uncertainty studies [Q3]:** The link to prior uncertainty studies lies in the fact that all the uncertainty estimation techniques we assess require some form of posterior estimation. However, most researchers typically validate this estimation solely on the quality of the uncertainty estimation (mostly proper scores like the NLL and Brier score, calibration like ECE and ACE, and out-of-distribution detection). We believe it is legitimate to evaluate the quality of the posterior estimation and explore whether there is a correlation with the quality of uncertainty quantification. Moreover, Table 1 highlights the non-trivial benefits of multi-mode methods compared to single-mode, notably in estimating epistemic uncertainty and its downstream impact on the detection of out-of-distribution samples. Furthermore, the table presents quantitative evaluations of the different mainstream uncertainty quantification methods on a larger scale than usual (especially for SGHMC) - the experiments being scaled up to TinyImageNet - and with many more metrics than usual. Indeed, usual benchmarks are often restricted to Accuracy, ECE, and NLL. Additionally to our new posterior quality metrics, we report another proper score (Brier), performance in OoD detection using the Mutual Information, as well as quantitative metrics of the diversities of the predictions, both in and out of distribution. The update of the document now adds one more reliable metric for top-label calibration (ACE) that readers can consider in their evaluation of the different methods at play.
> - **Regarding symmetries and their link to 'Functional Collapse' [Q4]:** given the extremely high number of permutation symmetries in a modern neural network, $10^{6000}$ for a simple ResNet-18, as well as the infinite number of scaling symmetries (the parameters are continuous), one could expect to find functionally similar neural networks in a large pool of independently trained DNNs. Our experiments in Section 5.1 show that this is not the case. The mutual information measures the functional diversity between two estimators on a given dataset, and we show that, among 500 thousand couples (1000 choose 2) of independently trained ResNet-18, none exhibit low functional diversity (functional collapse). That is, all couples have quantitatively more or less the same diversity. This hints that regardless of the huge size of the space of (nearly-)equivalent functionals, the complexity of the posterior is such that we empirically never fall into this space.
>
> - **Insights from weight permutations [Q5]:** Weight permutations are an important aspect for understanding Bayesian neural network strategies. Some of them, like weight averaging, as mentioned in [**2**], could be impacted if weights undergo permutations. This is why some papers, such as [**1**], suggest removing symmetries to facilitate weight averaging. BNNs employing weight averaging, like those found in [**2**, **3**], and [**4**], appear to benefit from stabilizing and averaging weights when the neural network ceases to exhibit permutations.
>
> We appreciate your valuable feedback and are committed to refining our paper to address these concerns adequately. We remain available if you feel that your concerns have not been fully addressed or have other concerns not mentioned in your first review.
>
> Sincerely,
>
> The authors
>
>
> ### References
>
> [**1**] Ainsworth, Samuel K., Jonathan Hayase, and Siddhartha Srinivasa. "Git re-basin: Merging models modulo permutation symmetries." In ICLR. 2023 (Best paper)
>
> [**2**] Izmailov, Pavel, et al. "Averaging weights leads to wider optima and better generalization." In UAI. 2018.
>
> [**3**] Maddox, Wesley J., et al. "A simple baseline for Bayesian uncertainty in deep learning." In NeurIPS. 2019.
>
> [**4**] Franchi, Gianni, et al. "TRADI: Tracking deep neural network weight distributions." In ECCV. 2020.

---

> ### Author Response · Authors · 2023-11-22
>
> Dear Reviewer **24pf**,
>
> We thank you for your review. We acknowledge the challenges of responding during this period but are sorry that we were not able to interact. We would be grateful if you glanced at our discussions with the other reviewers, as their valuable inputs have greatly helped us enhance the paper. We have made the following modifications that seem to address a large part of your concerns (this list is not exhaustive):
>
> - We have clarified and rewrote the parts of the manuscript that you found were not clear, thanks to the discussion with Reviewer **hdLm**. Notably, we have reworked Section 5.1 on the functional collapse and presented the benefits of the study of 5.2 at its end. It should be clear now that Section 5.1 deals with weight-space symmetries.
>
> - We have completely refactored our introduction of deep ensembles as an estimation of the Bayesian posterior, adding four pages of in-depth analysis of the strengths and weaknesses of the different methods at hand with Reviewer **Mjnv**. In this analysis, we trained 1,000 more networks corresponding to the specific architecture of [**1**]. We obtain an accuracy 5% greater than in [**1**]. In this part, we use our formalism to show that some aspects of the behavior of the gold standard for distribution estimation can be questioned.
>
> - We have fixed the small technical imprecision that decreased the performance of our ResNets-18, which is now above average.
>
> - We have extended our work, providing loss landscape interpretations of the minimum mass problem. We intend to generalize its study to ResNets in the following weeks.
>
> - We have explained some surprising results in calibration and added a more reliable metric than ECE.
>
> - We have proposed and tested a hypothesis to understand why neural networks do not converge to the minimum mass despite the log-log convexity of the problem.
>
> - We have detailed the formalism and the computation of the MMDs used in our Tables to measure the discrepancy between distributions.
>
> - We have extended our dataset to more diverse tasks and architectures.
>
> Although time is running short, we remain available if you feel some of your questions remain unanswered.
>
> Sincerely,
>
> The authors
>
> ### Reference
>
> [**1**] Izmailov, Pavel, et al. "What are Bayesian neural network posteriors really like?."In ICML. 2021.

---

### Official Review · Reviewer_Mjnv · 2023-10-31

**Soundness:** 3 good
**Presentation:** 3 good
**Contribution:** 2 fair
**Rating:** 5
**Confidence:** 4

**Summary:**

This work explores Bayesian neural networks and how their geometry is affected by weight space symmetries such as permutations and re-scaling. The authors show that the posterior can be expressed as a mixture over such symmetries, highlighting the redundant structure in such distributions. The authors then explore the effect of the scaling symmetry in more detail, showing that even with weight decay, the symmetry persists. Finally, using the gained insights, a rich exploration of different approximation methods is performed, and several metrics are studied to assess the quality of the resulting posterior. Multi-modal approximations generally offer the closest approximation in terms of maximum-mean discrepancy and best test performance, while surprisingly, single-mode approximations offer the best uncertainty predictions.

**Strengths:**

1. Connecting Bayesian neural networks with the recently obtained insights into the symmetries of the loss landscape of standard neural networks is a timely contribution and highly relevant to the field. The work is very well-written and mostly easy to follow.
2. The experiments are impressive and on a very large-scale, which is often missing in the BNN literature. I especially appreciate the different datasets investigated in this work. I also find the separation into single and multi-mode approximations very interesting, highlighting once more that deep neural networks really possess the multi-modal structure that they’re typically associated with.
3. I also appreciate the more in-depth investigation of the scaling symmetry, something that usually receives less attention compared to the more prominent permutation symmetry. I find it surprising that this symmetry persists to a good degree, even if weight decay is employed.

**Weaknesses:**

1. I think the first part of the paper regarding the symmetries is a bit extensive and insights from it are somewhat limited. To the best of my knowledge, it has been known for a while that the BNN posterior effectively consists of a mixture over such symmetries, so I don’t see much novelty in equation (8). While it sets up the subsequent discussion very nicely, I wouldn’t advertise this as one of the core contributions of this work. The in-depth study of the role of the scaling symmetry on the other hand is novel to my knowledge and fosters a better understanding.
2. An important aspect of the experimental setup, which is however almost not discussed at all, is the choice of “gold standard” for the posterior, to which all other approximations are compared against. This choice is of course crucial for the empirical evaluation! If I understand correctly, the authors chose a deep ensemble consisting of 1000 members. I’m a bit unsure regarding this choice, as it is quite controversial to what degree deep ensembles really implement an approximation to the Bayesian predictive distribution. There are several works arguing for yes [1, 2], while others oppose this view (to some degree) [3, 4]. [4] even argues that repulsive terms are needed to ensure that deep ensembles perform proper Bayesian inference asymptotically. I don’t think that this questions the validity of the experimental setup too strongly, but a proper discussion of this choice and differing view points **have to be discussed** such that readers can make up their own mind regarding this discussion.
3. While [2] is cited, the work really lacks a comparison to it. [2] also performs an empirical exploration of Bayesian posteriors and how other approximate methods perform, where the gold standard is full batch HMC. They only evaluate on one dataset (CIFAR10) but I still think this should be compared carefully, and how this work distinguishes itself from [2] should be properly stated. [2] also published their HMC checkpoints, I would really be interested in seeing whether the produced deep ensemble in this work produces a similar posterior. This could also clarify the discussion raised in point 2.
4. The work is not very careful at introducing a lot of concepts formally. As maximum-mean-discrepancy comparisons are one of the core contributions of this work, I think it would be worth it to formally introduce it in the main text. You also report a metric called NS, which is only described in the caption of Table 1, and I could not find a description in the main text nor a pointer to the Appendix, detailing how the “removal of symmetries” is actually performed. I find this quite an interesting point and I would like to see more discussion regarding how this removal enables a better comparison between different posteriors. Glancing at the numbers, there seems to be no consistent pattern (i.e. MMD sometimes goes up, and sometimes down after removing the symmetries) which is surprising as I would have expected to always see a decrease. Could you elaborate more on this? The multi-modal extension of some of these algorithms is also not clear. Are you simply averaging over all the samples coming from the different modes?
5. The uncertainty estimation results are surprising as the single-mode posteriors consistently perform better. The employed measure ECE has seen a lot of criticism recently [5] so I’m wondering whether a similar problem is at the origin here. Comparing multi-modal and single-modal is maybe also a bit unfair, as the testing accuracies vastly differ.


[1] Wilson and Izmailov, Bayesian Deep Learning and a Probabilistic Perspective of Generalization

[2] Izmailov et al., What Are Bayesian Neural Network Posteriors Really Like?

[3] D’Angelo and Fortuin, Repulsive Deep Ensembles are Bayesian

[4] He et al., Bayesian Deep Ensembles via the Neural Tangent Kernel

[5] Ashukha et al., Pitfalls of In-Domain Uncertainty Estimation and Ensembling in Deep Learning

**Questions:**

1. Why is your MNIST performance so low? As far as I know even a simple MLP with 2 hidden layers can easily reach $\approx 97\%$ performance.
2. Are you employing data augmentation in these experiments? And relatedly, are you tempering the posteriors, i.e. performing a grid search over temperatures? I’m asking since the cold posterior effect [6] could be a potential confounder here for test performance, as different approximations might be influenced differently.


[6] Wenzel et al., How Good is the Bayes Posterior in Deep Neural Networks Really?

---

> ### Author Response · Authors · 2023-11-14
>
> Dear Reviewer **Mjnv**,
>
> We thank you for your valuable feedback. We acknowledge your inclination towards a _weak rejection_ based on the following points:
>
> 1. the lack of novelty in the theory
>
> 2. the missing discussion on the ground-truth posterior:
>
> 3. the missing comparison to HMC in [**2**]
>
> 4. the introduction of MMD and Results
>
> 5. the single-mode ECE that seems anomalous
>
> Here is our answer to these points:
>
> - **On the lack of novelty on the theoretical part [W1],** we want to emphasize that while many papers in the literature focus on symmetries and loss landscapes [**6**, **7**, **8**], the key focus of our paper is on uncertainty. Although symmetries are briefly mentioned in some works [**5**], the posterior's intricacies are often overlooked due to its experimental complexity compared to the loss. However, we argue that quantifying uncertainty, particularly through the posterior, is crucial.
> More specifically, if you feel that we miss an important reference - that justifies the lack of novelty - on the theoretical side of our paper, especially the posterior as a mixture, we would be extremely interested and grateful.
> - **Concerning the ground truth posterior [W2],** we appreciate your observation and fully agree. We conducted experiments (as detailed in Section 5.1) illustrating that Deep Ensembles already exhibit some inherent repulsiveness. Yet, we will expand on this discussion in the paper, addressing its importance and the controversial nature of using an ensemble of DNNs as a ground truth posterior. In this new discussion, we notably intend to recall Wild et al. exciting and recent work [**3**].
>
> - **Regarding the comparison to HMC [W3],** we acknowledge the relevance of the suggestion. We have found the checkpoints released by the authors and have started training the ResNet models corresponding to the HMC proposed by [**2**]. It will take some time to ensure that our training procedures are as close as possible to [**2**], which is not trivial as we implement our models in PyTorch and not in JAX. However, we intend to finish this work before the end of the rebuttal period.
>
> - **Concerning the formal introduction of the MMD and its results [W4],** removing symmetries improves the histogram estimation but does not necessarily enhance the MMD. We plan to calculate simpler distances between histograms considering that the MMD we used (following Schrab and Gretton et al. [**4**]) is based on multiple kernels, is very complex,  and could bring the data into a Hilbert Space where symmetries are mitigated, potentially improving results. However, we expected MMDs to enable better distance estimation for multi-mode distributions.
>
> - **Concerning the single-mode ECE [W5]:** We agree that the ECE results seem anomalous. As mentioned in response to [**Q1**] (next point), our OptuNets - being shallow - tend to exhibit underconfidence. As suggested, we add the Adaptive Calibration Error (ACE) in Table 1. It confirms that the top-label calibration worsens when taking an ensemble of OptuNet, albeit to a lesser extent than the ECE. To prove the underconfidence of the networks, we provide calibration plots of the BNN and multi-BNN in Appendix B.4. We can provide more plots corresponding to other methods and architectures if desired. We also report in Appendix B.4 the relevant comment that comparing calibration performance is mostly pertinent at similar accuracies. We will acknowledge the quality of the feedback for the camera-ready.

---

> ### Author Response · Authors · 2023-11-14
>
> Now, for your questions:
>
> - **Concerning the MNIST Performance and Experiment Protocol [Q1],** we agree that we could expect 97% accuracy with a 2-layer MLP. However, OptuNet has only 392 parameters, a significantly lower number than a comparable MLP: considering a hidden dimension of 10, we would expect at least 7600 parameters, more than 19 times as much as our network.
>
> - **Regarding the data augmentation [Q2],** we deliberately keep the CIFAR experiments simple, employing only horizontal flips to avoid poisoning the posterior. Indeed, some data augmentations have been shown to sometimes harm ensembles and multi-mode methods (e.g., [**1**]). Finally, we refrain from tempering the posteriors to focus on understanding the relationship between weight-space symmetries and the posterior.
>
> Once again, we appreciate your insightful comments and are committed to addressing them in the revision.
>
> Sincerely,
>
> The authors
>
>
> ### References:
>
> [**1**] Wen, Yeming, et al. "Combining ensembles and data augmentation can harm your calibration." In ICLR. 2021.
>
> [**2**] Izmailov, Pavel, et al. "What are Bayesian neural network posteriors really like?."In ICML. 2021.
>
> [**3**] Wild, Veit David, et al. "A Rigorous Link between Deep Ensembles and (Variational) Bayesian Methods." In NeurIPS. 2023.
>
> [**4**] Schrab, Antonin, et al. "MMD aggregated two-sample test." JMLR 2023.
>
> [**5**] Wiese, Jonas Gregor, et al. "Towards Efficient MCMC Sampling in Bayesian Neural Networks by Exploiting Symmetry." arXiv preprint arXiv:2304.02902 (2023).
>
> [**6**] Fort, Stanislav, Huiyi Hu, and Balaji Lakshminarayanan. "Deep ensembles: A loss landscape perspective." arXiv preprint arXiv:1912.02757 (2019).
>
> [**7**] Liu, Chaoyue, Libin Zhu, and Mikhail Belkin. "Loss landscapes and optimization in over-parameterized non-linear systems and neural networks." Applied and Computational Harmonic Analysis (2022).
>
> [**8**] Fort, Stanislav, and Adam Scherlis. "The goldilocks zone: Towards better understanding of neural network loss landscapes." In AAAI. 2019.

---

> > ### Comment · Reviewer_Mjnv · 2023-11-20
> > **Reply**
> >
> > **Posterior and Symmetries:** There is for instance [1] (equation 3) that has characterised the posterior of a neural network as a mixture over all permutations. They do not consider scale-symmetries but that is a relatively straight-forward extension in my opinion. I do acknowledge that it is interesting that not all scales get equal probability but I'm unsure what insights one can draw from this.
> >
> > **DE as ground-truth:** I think this choice is very crucial (as all consequent insights heavily depend on it) so the validation with the HMC checkpoints, which offer theoretical guarantees to converge to the true posterior, is essential.  I hope that the authors can run that experiment before the end of the rebuttal period. How do the two methods compare in terms of computational needs (HMC vs DE)? Why not run HMC (maybe not full-batch but relatively large batch) in the first place as it is universally viewed as the gold standard?
> >
> > **Comparison to Izmailov et al.:**  I would have also liked a comparison just in terms of the two works. What new insights and results are obtained through this work, in comparison to Izmailov et al.? You seem to evaluate on more datasets with a different ground-truth posterior but what novel insights can be deduced?
> >
> > **Removing symmetries:** It is still unclear to me how you remove the symmetries when comparing two posterior approximations. Is this somehow done in the calculation of MMD? I would really appreciate a bit more detail here as this could be one of the stronger contributions of this work. Having a metric that allows to compare two posteriors while accounting for symmetries in a tractable way would be very useful.
> >
> >
> > [1] Kurle et al., On Symmetries in Variational Bayesian Neural Nets, 2021

---

> ### Author Response · Authors · 2023-11-22
>
> Dear Reviewer Mjnv,
>
> We are very grateful for your reactivity and feedback. We address each point in turn.
>
> - **On the reference to the posterior as a mixture of distributions [W1].** We thank you for your reference interpreting the posterior distribution as a discrete mixture of distributions. We acknowledge that the permutation part of our Proposition 1 is very intuitive (from a statistical physics point of view, it boils down to the non-identifiability of the inner neurons). However, we are concerned by the lack of formalism - the notations are not defined - as well as the absence of any sketch of proof concerning Eq. (3) of [**1**]. Despite careful research, we have not found any supplementary material that would include any sketch of proof or definition of the notations: e.g., the core element of Eq. (3), $p_n$, is not defined at all, and it is trivial that all distributions are mixtures of an arbitrary number of arbitrary distributions (although we understand what the authors meant).
>
> It is important to clarify that our intention is not to criticize the excellent work published in a NeurIPS workshop; rather, we emphasize the necessity of considering our contributions that are supported by mathematical and empirical proofs. We will cite [**1**] and indicate that we extend and formalize their result for full transparency.
>
> On our side, we have paid special attention to providing fastidious and formalized reasoning steps on all the propositions of our paper, but also empirical evaluations. As such, we argue that despite the intuitiveness of a part of Proposition 1, the formalism of symmetries remains a valuable contribution. We remain highly interested in other properly formalized references that would undermine the novelty of our theoretical approach.
>
> This said, we welcome the reviewer's advice and shorten the introduction to our formalism from the core of the paper, inlining some equations and moving a formal definition in Appendix C.
>
> - **On theoretical guarantees in high-dimensional settings & comparison with HMC and estimation of the posterior with independently-trained checkpoints [W2].** We have been genuinely impressed by the paper from Izmailov et al. to be able to scale algorithms of this quality to real-world architectures and datasets - although ResNet-20 remains rather small, with 275,000 parameters, around 40 times less than the ResNet-18 we used for TinyImageNet, which requires images 4 times as big - and have the utmost respect for their inspiring work. However, we are not certain that HMC, with the batch size we could have used, would have been relevant in practice. Please note that we have implemented SGHMC, which we test in Table 1 but which does not provide satisfactory results. We acknowledge that improving its results with hyperparameter tuning may be possible, although theoretical concerns have also been raised [**7**]. We have added a new section for this discussion: Appendix B. Indeed, we should consider the extremely high dimensionality of the Bayesian posterior for models such as ours and [**2**]’s, which is very far from the classical use-case of MCMC methods. We have no doubts concerning the quality of the algorithm and its implementation by [**2**]. However, we show that - in contrast to deep ensembles - HMC breaks the albeit basic and intuitive Proposition 1. Although this can be interpreted as an advantage (it is useless to sample from symmetric positions - yet DE does not, as shown in Section 5.1), this reveals that quantifying the quality of our estimation of the posterior with independent checkpoints using HMC as ground truth may not be as straightforward as supported by the Reviewer. We discuss this point in the new Section B of the Appendix.
>
> - **Computational cost:** We have a limited computational budget. The reviewer may refer to the ethics statement, where we quickly estimated the total computational footprint of our work. More specifically, concerning the experiments on ResNet-20 on CIFAR-10, training 1000 checkpoints took less than 200 NVIDIA V100 hours on single GPUs. Although the full computational budget of [**2**] is not clearly stated, the technical difficulty and the high number of TPUv3 required (512) are mentioned _at least_ 4 times in their paper. Using a rule of thumb and hypothesizing perfect scaling, full-batch HMC with ResNet-18 on TinyImageNet would require 512x40 (number of parameters) x 4 (size of the images) x 10/6 (number of images) = 137,000 TPUv3, which is out of reach. We confidently estimate the full cost of their training on CIFAR-10 to be more than two orders of magnitude greater than ours, and the VRAM requirements are simply out of reach. Indeed, [**2**] also needed to perform ablations and hyperparameter tuning, which were simply unnecessary in our case since we use common-knowledge hyperparameters [**4**]. We also detail this part in our new Appendix B.

---

> ### Author Response · Authors · 2023-11-22
>
> Following the Reviewer's advice, we provide precise explanations in support of our choice of independently trained checkpoint in the core of the paper (in the introduction of Section 4) as well as a pointer to Appendix B. Moreover, we include a new table of metrics: Table 2 - not yet fully filled - that compares the performance of deep ensembles and our baselines with HMC as proposed by [**3**]. We show that deep ensembles outperforms HMC on all metrics, the quantitative out-of-distribution functional diversity excepted (OODMI). We also compare the distance metrics based on a ground truth estimated by HMC and DE. It seems that the distances based on DE express a larger difference between the mono-modal and multi-modal methods but this remains to be confirmed.
>
> Finally, we know that science is not about arguments of authority. Still, you may be interested in this [excellent blog](https://cims.nyu.edu/~andrewgw/deepensembles/) by Andrew G. Wilson, the last author of [**2**], and the following quotation - confirmed in [**3**] - and to which our findings fit: “Moreover, even if we could use full batch MCMC, we never get anywhere near the asymptotic regime, and typically just use a handful of samples in practice. If, given computational constraints, deep ensembles are typically more Bayesian than many of the alternatives, then you cannot reasonably classify them as a non-Bayesian method.”
>
> In our assessment of the debate, we conclude that the selection between HMC and Ensemble methods is largely subjective, as both approaches are susceptible to criticism. While HMC faces challenges related to unrealistic memory complexity, particularly with certain datasets, Ensemble methods may offer a more straightforward but maybe time-consuming alternative.
>
> **On the contributions compared to [2] [W3].** To address your question fully, we provide a list of the contributions of [**2**] and our work.
>
> Concerning [**2**], their main focus and contributions are listed below. We are not the authors of [**2**], and some points may have been missed despite our diligence.
>
> ## Contributions of [**2**]
>
> **1/ HMC Training:** [**2**] concentrate on training HMC chains and the theoretical and technical hardships linked to scaling full-batch MCMC.
>
> **2/ Checkpoint Sharing:** [**2**] share 720 HMC-samples of a modified ResNet-20 trained on CIFAR-10, 570 modified ResNet-20 on CIFAR-100 checkpoints and 1200 CNN-LSTM on a small-scale task of NLP for collaborative purposes.
>
> **3/ Technique Comparison:** [**2**] provide a comparative analysis of Stochastic Gradient Hamiltonian Monte Carlo (SGHMC), HMC, and deep ensembles for Bayesian Neural Network (BNN) estimation and uncertainty quantification.
>
> **4/ Posterior Tempering Discussion:** [**2**] explore and discuss the concept of cold posterior and its impact in practice.
>
> **5/ Effect of Priors:** [**2**] investigate into the impact of priors in BNNs.
>
> **6/ Posterior Visualization and Link to Loss Landscape:** [**2**] propose posterior visualization and describe its connection to the loss landscape.
>
> **7/ Advancement of MCMCs and posterior knowledge:** [**2**] show that HMC can perform very well for very high dimensional DNN posterior distributions despite the known drawback that an “HMC chain is extremely unlikely to jump between isolated modes” [**2**] (page 5).
>
> ## Our contributions
>
> Our focus and main contributions are the following:
>
> **1/ Ensemble Training:** We emphasize training ensembles of independent DNNs.
>
> **2/ Checkpoint Sharing:** We share 100,000 MLPs, 10,000 LSTMs, 10,000 OptuNets trained on MNIST, 1000 modified ResNet-20 on CIFAR-10, 10,000 ResNet-18 on CIFAR-100 and 2,000 ResNet-18 trained on ImageNet-200. We argue in **[W2] **that these checkpoints are legitimate for the task of posterior estimation. However, they may also serve many other objectives, such as pruning or the comprehension of the latent space. We suggest that they are somewhat more relevant for these tasks since they correspond to common training instances.
>
> **3/ Symmetry Impact Study:** We provide a full formalism for the impacts of weight-space symmetries on the Bayesian posterior of DNNs. This study is not only theoretical as it sheds new light on estimating the posterior by HMC, for instance (see our first answer). Unless mistaken, symmetries are not mentioned explicitly in [**2**].
>
> **4/** **The minimum mass problem:** We show that this problem is strictly convex after changing variables and provide a code to solve it for modern networks. We perform experiments on this matter and show that, surprisingly, regularized networks do not converge to the optimum. Furthermore, we extend our study to provide a loss landscape analysis of this problem and a hypothesis to explain this finding.
>
> **5/ New posterior quality assessment approach:** we propose a new method to evaluate the quality of the estimation of the posterior of DNNs and evaluate this method on several architectures and tasks.

---

> ### Author Response · Authors · 2023-11-22
>
> **6/ Benchmark of uncertainty quantification methods**: we perform an extensive benchmark of multiple methods for uncertainty quantification, including relevant metrics that are not always evaluated in other papers. We provide a specific focus on the effect of multi-modal estimation.
>
> **7/** **Functional collapse:** We propose and study the notion of _functional collapse_ of ensembles. Specifically, we show that there is, surprisingly, no functional collapse when ensembling checkpoints for large enough architectures. This is a reassuring result for practitioners (including the authors) but also has deeper implications on the complexity of the posterior and the reliability of DE for estimating this posterior.
>
> **8/ Ensemble vs. HMC discussion:** We explore and discuss the comparative advantages and disadvantages of ensembles of independent DNNs versus HMC. We leverage our mathematical formalism on symmetries and discover empirical limits of HMC to which DE seems immune. Finally, we compare our MMDs using HMC and ensemble as target posterior estimation.
>
> **9/ Posterior visualization**: we also propose to explore the marginal posteriors of ResNet-20 and MLPs. For the first case, this highlights that DE’s posterior estimation may be more multi-modal than HMC’s.
>
> Overall, while we agree with the reviewer that certain points are shared, although performed differently, to different scales, with varying performance of baselines (ours are 5% stronger on CIFAR-10), we argue that our paper develops its own originality and offers a novel point of view on an important topic.
>
> Following the Reviewer’s suggestion, and although the available space in the related works is very limited, we include a brief summary to detail our main difference with [**2**]. If the Reviewer feels that this is not enough, we can add a summary of this answer to the Appendix along with a pointer in the core paper.
>
> **On the symmetry breaking algorithms [W4].** Our main contribution on this side is the non-trivial algorithm that transforms the weights of neural networks into a log-log strictly convex problem, which enabled both Figure 2 and Figure 11 (on page 38), the latter describing the loss landscape implications of the _min-mass_ problem. For the experiments of Tables 1 & 2, we use variations of the normalization [**5**] and sorting [**6**] algorithms that we rediscovered and reimplemented. However, please note that despite the differences, we cite the authors at the origin of these ideas. If the reviewer feels that it would be an interesting addition to the paper, we can provide the pseudo-code. However, as written earlier, we believe in Open Science: we will release all of our codes after the anonymity period and put a link to our code directly at the end of the abstract. Finally, we could discuss the optimality of sorting and normalizing the weights. While it guarantees an important reduction of the weight space, these are not the best algorithms to regroup checkpoints. This topic is of particular interest to the community of model merging, but to the best of our knowledge, their work mainly focuses on regrouping pairs of checkpoints, which could be suboptimal for large groups such as ours.
>
> **On the introduction and the formalism of MMDs** **[W4]**. As written in **[W1]**, we have completely updated our introduction of MMDs in Section 4.1. We provide more details on their formalism in Appendix C.5, which also describes precisely the meaning of “MMD” and “NS” in our Tables. Briefly, NS computed just like the MMD after normalizing [**5**] and then sorting [**6**] the weights.
>
> We would be very grateful if you could inform us whether you found our arguments convincing, and we remain available for any remaining concerns and questions. Once again, we deeply thank you for your feedback on our work and are pleased to enrich our paper with your relevant remarks.
>
> Sincerely,
>
> The authors
>
> ### References
>
> [**1**] Kurle, Richard, et al. "On symmetries in variational Bayesian neural nets." In NeurIPs Workshops (2021).
>
> [**2**] Izmailov, Pavel, et al. "What are Bayesian neural network posteriors really like?."In ICML. 2021.
>
> [**3**] Wilson, Andrew G., and Pavel Izmailov. "Bayesian deep learning and a probabilistic perspective of generalization." _Advances in neural information processing systems_ 33 (2020): 4697-4708.
>
> [**4**] He, Kaiming, et al. "Deep residual learning for image recognition." In CVPR. 2016.
>
> [**5**] Neyshabur, Behnam, Russ R. Salakhutdinov, and Nati Srebro. "Path-sgd: Path-normalized optimization in deep neural networks."In NeurIPS. 2015.
>
> [**6**] Pourzanjani, Arya A., Richard M. Jiang, and Linda R. Petzold. "Improving the identifiability of neural networks for Bayesian inference." In NeurIPSW. 2017.
>
> [**7**] Zou, Difan, and Quanquan Gu. "On the convergence of Hamiltonian Monte Carlo with stochastic gradients." In ICML. 2021.

---

### Official Review · Reviewer_QYDw · 2023-11-05

**Soundness:** 4 excellent
**Presentation:** 3 good
**Contribution:** 3 good
**Rating:** 6
**Confidence:** 3

**Summary:**

The paper presents a study of weight-space symmetries in Bayesian neural networks (BNNs). The authors propose mathematical definitions of permutation and scaling symmetries and define the BNN posterior as a mixture of these. They further present a new way to define a unique network among all scaling-symmetrical candidates.

The paper further contains an empirical study of BNN posteriors. Here, the authors evaluate a selection of posterior approximation methods for their approximation quality on classification and OOD detection tasks, study the functional diversity among ensemble members, as well as investigate how often the BNN parameters permute during training.

**Strengths:**

1. I find the mathematical treatment of the scaling and permutation symmetries, as well as their contribution to the posterior, quite exciting. It may not be overly novel, but the exposition is interesting and inspiring.
2. The empirical analysis is exciting. The study of the different posterior approximations provides a good overview of the effects of the approximations, and it feels like there's even more to be gained from Table 1 than what the authors discuss. Furthermore, section 5 appears quite novel and contains some fascinating insights.
3. The authors promise to release a dataset of estimated posteriors for a large number of models, which could be of great value to the community.
4. The paper overall appears original and of high technical quality and should be of great interest to the ICLR community.

**Weaknesses:**

1. The paper seems to lack a specific focus. While the theoretical and experimental parts of the paper are both exciting, they seem largely disconnected. The developed mathematical formalism does not appear to be used for anything, and it is even unclear what these definitions buy us in terms of understanding BNN posteriors. The experiments in section 4 do not appear to consider symmetries at all, and whereas those of section 5 do, the link to the definitions in section 3 is unclear. To me, the paper seems to be compiled from two distinct papers, each of which would be interesting to read.
2. The clarity of the paper could be improved. The figures (1 and 2 in particular) are not explained in sufficient detail to understand what they show, and the text itself is sometimes unclear.

**Questions:**

1. What is the link between the formalism of section 3 and the experiments in sections 4 and 5?
2. I don't fully understand figure 2. What is on the second axis in the left plot, what does the middle plot show, and what is the difference between "opt." and "convergence" in the right plot?
3. In section 4.1, you write "For better efficacity, MMDs are computed on RKHS that allow for comparing all of their moments." What does this mean?
4. If I understand the paper correctly, the estimates of the BNN posteriors come from the parameters of a 1000-member ensemble trained using maximum likelihood. Can we be sure this is a faithful representation of the posterior? For instance, do we know anything about the higher-order moments of those modes?
5. Related to the previous question, if the posterior is indeed approximated by 1000 ML solutions, does the MMD metric even make sense for any of the single-mode methods? One could argue that single-mode methods aren't meant to describe the posterior adequately, but the hope is that their predictive distributions aren't too affected by the single-mode limitation.

---

> ### Author Response · Authors · 2023-11-14
>
> Dear Reviewer QYDw,
>
> Thank you for your detailed review and feedback. We acknowledge the concerns that led you to recommend rejection. If our understanding is correct, they primarily revolve around:
>
> 1. the lack of a clear focus, and
>
> 2. the clarity of the figures and the protocol for the computation of the MMD
>
>  We address each point in turn:
>
> - **Concerning the lack of a clear focus [W1, Q1]:** The primary focus of our paper is to elucidate the impact of weight-space symmetries on the posterior of modern DNNs. We believe that enhancing the understanding of DNNs involves both theoretical and experimental aspects, both of which are encompassed in our work. As such, we believe that our paper is not a compilation of two different papers. Our qualitative results in Figures 1 and 10 (Appendix) illustrate how symmetries can distort the posterior estimation. While these results are not fully conclusively supported by Table 1, the challenges of representing distributions in extremely high-dimensional spaces alter the meaning of the simplest distance metrics. We hypothesize that MMD may address these symmetry issues, potentially obscuring the phenomena we aim to highlight. However, we acknowledge your concerns and will try to add another more explainable metric working directly on the marginals (see also **[Q4]**).
>
> - **Concerning the clarity of figures and MMD [W2, Q2, Q3],** recognizing the need for conciseness, we have chosen what to include in the main paper versus the appendix. We have moved some detailed explanations of Figure 1 to the main paper and provided additional clarification for Figure 2 in Section 3.6. In Figure 2, we provide insights on the difference of mass between 1) the network at the end of the training (that we call at convergence) and 2) the same network after minimization of its regularization term (at the optimum). We are curious about your opinion on whether these changes improve the paper's clarity on these points. Regarding the MMD, we will rephrase this section to emphasize that MMD serves as a suitable discrepancy metric for characterizing distributions when appropriate kernels are chosen and clarify the sentence you highlighted in **[Q3]**.
>
> Concerning your other questions:
>
> - **Regarding the nature of Ground Truth Posterior [Q4],** we acknowledge the inherent uncertainty in representing the posterior with 1000 DNNs. While we understand this observation, we should have addressed this debate more explicitly in the paper to provide context for new readers. We will add such a discussion in the next update of our submission. However, this nuance should not invalidate our results. Indeed, MMDs compare samples drawn from different distributions and have the advantage of converging very quickly. However, we intend to add at least another distance on the marginals, which will be more interpretable.
>
> - **Concerning the utility of Single-Mode Methods [Q5]:** Single-mode strategies are standard in uncertainty quantification, with only a few papers exploring multi-mode approaches. It is crucial to demonstrate the potential issues with single-mode methods, as this contributes to the broader understanding of uncertainty quantification. While we agree that they do not necessarily seek to optimize their estimation of the posterior, we argue that it remains important to test their ability to estimate the epistemic uncertainty properly. We also agree that comparing multi-modal distributions to single-mode distributions can be unfair; however, please keep in mind that all of the single-mode methods have been chosen on the basis that they claim to sample from approximations of the posterior.
>
> Based on these comments, we would like to clarify our understanding of your decision to recommend rejection. Would you have any additional concerns, or could you confirm that your chosen mark accurately reflects this evaluation? Moreover - if our answers do not convince you - we would be very grateful to benefit from your feedback.
>
> Sincerely,
>
> The authors

---

> > ### Comment · Reviewer_QYDw · 2023-11-22
> >
> > Dear authors,
> >
> > Thank you very much for your replies to my questions and concerns, and many apologies for not responding before now (as you remark, life can be busy).
> >
> > The rebuttal discussions have been interesting to follow, and I really appreciate how you have added new hypotheses, thoughts, and intuitions here. These are aspects I missed in the original submission, and I hope you'll continue to add such insights to the paper. I am extremely impressed by the amount of work you have put into updating the paper, which has improved the paper's clarity and value dramatically. I especially like the new appendix B, which adds another aspect to your study of weight-space symmetries.
> >
> > My concern about the disconnectedness of the paper still remains, though. I still find little connection between the theoretical and empirical parts of the paper in the sense that they can be read independently of each other. This also seems to be a concern of reviewers Mjnv and 24pF, and it is not to say that either part isn't interesting - on the contrary - but I think your paper would be much stronger if, say, your mathematical formalism directly led to experiments that would otherwise not have been obvious, and which would provide us with clear, new understandings of BNNs.
> >
> > Some other concerns, which have also been raised by the other reviewers, still remain. For instance, reviewers Mjnv and hdLm also commented on the choice of representing the posterior using DEs. I very much understand your reasoning for using DEs for this, especially in the context of studying symmetries, but it is important to acknowledge the potential issues of using DEs as a posterior approximation, even if other choices, such as HMC, have issues themselves. It doesn't make this work less interesting, but the reader must be aware of the potential implications of your approximation choice, and I would prefer if you rephrase "ground truth" and the like to, say, "posterior approximation".
> >
> > All that being said, I think your work is interesting, and the improvements you have made to the paper so far are impressive. I have therefore raised my score to recommend acceptance, and I hope you will continue to add additional experiments, explanations, and insights to your paper.

---

> > > ### Author Response · Authors · 2023-11-22
> > >
> > > Dear Reviewer **QYDw**,
> > >
> > > We are pleased that our discussions with the reviewers were insightful and that you value our work during this rebuttal and the modifications that we have implemented.
> > > We fully agree with you that the limitations of the posterior estimation by DE were not yet transparent enough in the core of our previous version of the manuscript. We now make this limitation more explicit at the end of the introduction of Section 4, just before the pointer to Appendix B. As recommended, we have removed all occurrences of “ground truth,” which we agree were misleading.
> > >
> > > We will continue to improve our work, and we thank you again.
> > >
> > > Sincerely,
> > >
> > > The authors

---

### Official Review · Reviewer_cm11 · 2023-11-10

**Soundness:** 3 good
**Presentation:** 3 good
**Contribution:** 3 good
**Rating:** 8
**Confidence:** 3

**Summary:**

this paper conducts a large-scale exploration of the posterior distribution of deep bayesian neural networks (bnn), focusing on real-world vision tasks and architectures. it highlights the critical role of weight-space symmetries, particularly permutation and scaling symmetries, in understanding the bayesian posterior. the paper introduces new mathematical formalisms to elucidate these symmetries' impacts on the posterior and uncertainty estimation. it evaluates various methods for estimating the posterior distribution, using maximum mean discrepancy (mmd) to assess their performance in capturing uncertainty. a significant contribution is the release of a large-scale dataset comprising thousands of models across various computer vision tasks, facilitating further research in the field of uncertainty in deep learning. the study investigates the proliferation of modes due to posterior symmetries and the tendency of ensembles to converge towards non-functionally equivalent modes. it also discusses the influence of symmetries during the training process. the findings suggest the complexity of bayesian posterior can be attributed to the non-identifiability of modern neural networks, viewing the posterior as a mixture of permuted distributions​.

**Strengths:**

1. originality in symmetry analysis: the paper's exploration of weight-space symmetries, particularly permutation and scaling symmetries, in deep bayesian neural networks is highly original. it builds upon existing studies of dnn loss landscapes, like those by li et al. (2018) and fort & jastrzebski (2019), and extends these concepts to the bayesian context. the distinction between permutation and scaling symmetries and their unique impacts on the posterior is a novel contribution (section 1)​​.

1. comprehensive methodological approach: the authors leverage a mathematical formalism to highlight symmetries' impacts on the posterior and uncertainty estimation in deep neural networks, adding theoretical depth (section 1). it facilitates a more nuanced understanding of the bayesian posterior, bridging a gap in current literature​​.

1. empirical validation with a large-scale dataset: the release of a comprehensive dataset including the weights of thousands of models is a substantial contribution. it not only validates the paper's findings but also enables the community to conduct further research and empirical validation across various computer vision tasks (section 4). this dataset is a practical resource that can catalyze future advancements in uncertainty quantification in deep learning​​.

1. insights into posterior symmetries and model performance: the paper delves into the complexity of bayesian posteriors, attributed to the non-identifiability of modern neural networks. this perspective, viewing the posterior as a mixture of permuted distributions, is a significant insight. it aligns with recent works that discuss the intricacies of dnn architectures and their implications on model performance (e.g., entezari et al. (2022))​​.

1. advancing uncertainty quantification: the study's focus on uncertainty quantification, specifically how symmetries affect uncertainty estimation, is a major strength. the use of maximum mean discrepancy (mmd) to evaluate the quality of various posterior estimation methods on real-world applications is a rigorous approach, contributing to a better understanding of uncertainty in deep learning (section 4)​​.

p.s. really nice paper! was a joy to read.

**Weaknesses:**

1. limited scope in empirical validation: while the dataset released is extensive, the paper's empirical validation primarily focuses on vision tasks and specific architectures like resnet-18. this raises questions about the generalizability of the findings across different types of neural network architectures and tasks. expanding the empirical validation to include a broader range of architectures would strengthen the findings.

1. need for further exploration of symmetries in training: the paper discusses the influence of symmetries during the training process but does not delve deeply into this aspect. further exploration of how these symmetries manifest and can be leveraged or mitigated during the training process could provide valuable practical insights for model development.

1. lack of broader contextualization: while the paper does an excellent job of situating its contributions within the immediate field of bayesian neural networks, it could benefit from a broader contextualization. this includes discussing its findings in relation to other approaches in deep learning and uncertainty quantification, and how these approaches diverge from the paper’s findings.

**Questions:**

1. regarding originality in symmetry analysis: the paper provides an original analysis of weight-space symmetries in bayesian neural networks. can you elaborate on how these findings might extend or differ from the insights provided by existing studies on dnn loss landscapes, such as those by li et al. (2018) and fort & jastrzebski (2019)? specifically, how do these symmetries manifest differently in bayesian networks compared to traditional dnns?

2. on comprehensive methodological approach: could you provide practical examples or case studies demonstrating how your results can be applied to actual neural network design or optimization problems?

3. regarding empirical validation with a large-scale dataset: the release of a dataset including thousands of model weights is a significant contribution. however, the focus seems to be on vision tasks primarily. how might the findings and applicability of your research differ when applied to other types of neural network architectures or non-vision tasks?

4. on insights into posterior symmetries and model performance: your paper suggests the complexity of bayesian posteriors can be attributed to the non-identifiability of modern neural networks. how might this insight impact the current practices in model evaluation, particularly in fields where high certainty and reliability are crucial?

5. regarding advancing uncertainty quantification: could you discuss the limitations or challenges you faced using maximum mean discrepancy (mmd) to evaluate posterior estimation methods? are there scenarios where mmd might not be an effective measure, and if so, what alternatives would you recommend?

6. on limited scope in empirical validation: given the paper's focus on specific architectures and vision tasks, are there plans to extend this research to include a wider variety of neural network types and tasks? how do you foresee the findings changing with different architectures or in different application domains?

8. on need for further exploration of symmetries in training: the paper touches upon the influence of symmetries during the training process. could you detail any specific strategies or methodologies that could be employed during training to either leverage or mitigate these symmetries?

9. regarding lack of broader contextualization: how do the findings of your paper align or contrast with other approaches in deep learning and uncertainty quantification? can you provide insights into how your research contributes to or diverges from the broader landscape of deep learning research?

---

> ### Author Response · Authors · 2023-11-14
> **Response to Reviewer cm11**
>
> Dear reviewer **cm11**,
>
> We thank you for your feedback, relevant questions, and heartwarming comments. We are happy to see that we were able to convey a part of the enjoyment of this research.
>
> If our understanding is correct, you lean for acceptance. Your main concerns about our paper are the following:
>
> 1. the limited empirical scope of the findings,
>
> 2. the limited study of the influence of symmetries during the training process,
>
> 3. the lack of broader contextualization
>
>
>
> Here are our answers for these 3 points:
>
> - **Regarding the limited empirical scope of the findings and dataset [W1, Q3, Q6]**, we fully agree with the reviewer on this point. To improve on this side, we intend to enrich our dataset of checkpoints with two other types of models, starting with simple MLPs on regression tasks, as well as LSTMs on a time-series forecasting dataset. We also aim to add a table in the supplementary materials with the results on the MLPs and these LSTMs before the end of the rebuttal.
>
> More specifically, for [**Q3**], a limitation of our work - at least for the scaling symmetries part - resides in using non-negative homogenous activation functions, such as the ReLU. This may become a problem in vision tasks where the GeLU is starting to take precedence, in ViTs, of course, but also in modern ConvNets such as ResNeXt [**1**] (even though it does not seem to yield better performance).
>
> One interesting aspect we could consider is the posterior of neural networks after fine-tuning on downstream tasks, either in NLP or vision. This could be more affordable considering the computational cost of training current moderate to large-sized models.
>
> - **Concerning the influence of symmetries during the training process [W2]**, we once again agree with the reviewer. The main problem in quantifying this influence comes from the difficulty of even detecting such symmetries due to the size of the corresponding space. For this problem, we are considering getting inspiration from the algorithms developed for weight space merging, such as [**2, 4**]. These algorithms could help pair models efficiently and detect near-symmetries.
>
> Specifically, for [**Q7**], symmetries could impact methods based on the averaging of weights, such as SWA and SWAG, but also [**3**], for instance. As such, it is probably beneficial for these methods to avoid collecting similar models in the function space. Luckily, the experiments of Section 5.1 tend to indicate that the number of permutations remains limited in the standard setting. We could check that this is also the case for SWA’s training procedure. On the other hand, reducing weight-space symmetries during training could prove essential to enable large-scale model merging as it is combinatorially difficult to find the best symmetries for merging large groups of models.
>
> - **On the lack of broader contextualization [W3, Q1, Q8]**, and as suggested by the reviewer, we will add a section in the Appendix to detail the relationships between our results and the theory developed for the study of the loss landscape.

---

> ### Author Response · Authors · 2023-11-14
> **Second part of the response to Reviewer cm11**
>
> Now, for the remaining questions:
>
> - **Regarding the application to optimization problems and network design [Q2, Q3]**, we believe that our dataset and the methods developed in this paper could help better understand SGD. Although we sought to estimate the distribution of the networks trained with an “early-stopping strategy” (in the sense of a predefined number of training epochs), we saved temporary checkpoints to recover a part of their path. This could help study the variation of the “posterior” with the number of steps and improve our empirical understanding of the non-convex convergence of modern neural networks. We also think that the large-scale dataset we are about to release could help progress on pruning tasks or comprehend the latent space of deep models.
>
> - **On the complexity of the posterior that can be attributed to weight-space symmetries and the impact on practices in model evaluation [Q4],** we would like to stress that the experiment of Section 5.2 shows that only a part of this complexity can be attributed to weight-space symmetries. To our understanding, it is relevant to remember the existence and effect of these symmetries when building systems that are expected to rely on the good quality of the posterior estimation, for instance, when we desire to rely on estimations of the epistemic uncertainties. Another case of importance for symmetries and posterior is model merging [**2**], as written above (**[W2]**).
>
> - The first **challenge when using MMD [Q5]** is the computational cost. In general, computing the distances or discrepancies between the posterior distribution of ResNet-18s is computationally demanding as it requires a large quantity of RAM. In our experiments, 128GB seemed only just sufficient. Moreover, the JAX implementation requested thousands of TB to compute the MMD, and we had to transcribe it to PyTorch. Despite the efforts in this direction, the MMD is scalable, but only to a certain point, and we had to restrict the comparisons between the distribution of the layers, leading to a form of block-diagonal comparison. The MMD is also a complicated object; this is why we are going to try to propose - before the end of the rebuttal - one or two simpler alternatives.
>
> We thank you again for the very high quality of your questions. We may get back to you later in the rebuttal period to improve our answers.
>
> Sincerely,
>
> The authors
>
>
> ### References:
>
> [**1**] Liu, Zhuang, et al. "A convnet for the 2020s." In CVPR. 2022.
>
> [**2**] Ainsworth, Samuel K., Jonathan Hayase, and Siddhartha Srinivasa. "Git re-basin: Merging models modulo permutation symmetries." In ICLR. 2023.
>
> [**3**] Franchi, Gianni, et al. "TRADI: Tracking deep neural network weight distributions." In ECCV. 2020.
>
> [**4**] Ramé, Alexandre, et al. "Recycling diverse models for out-of-distribution generalization." In ICML. 2023.

---

> ### Comment · Reviewer_cm11 · 2023-11-21
> **rebuttal reply**
>
> dear authors,
>
> thank you very much for your response!
>
> as detailed in my review, i find the paper fascinating and the contribution comprises both the results and the weight releases.
>
> i maintain my rating of 8.
>
> in good spirits,
>
> reviewer cm11

---

> > ### Author Response · Authors · 2023-11-22
> >
> > Dear Reviewer **cm11**,
> >
> > We thank you again for your review and your excellent questions. We will keep them in mind for our future work.
> >
> > Sincerely,
> >
> > The authors

---

### Author Response · Authors · 2023-11-14
**General response to the Reviewers**

Dear Area Chair and dear Reviewers,

We extend our gratitude to all the reviewers for their thoughtful and insightful feedback. While we are happy that our paper's exploratory and innovative nature is being recognized, we acknowledge that reviewing a less mainstream paper such as ours may raise some concerns. Hence, we wish to reiterate the four key contributions of our work:

1. A posterior dataset: We have established a novel dataset of DNN weights, a valuable resource for the broader study and comprehension of these networks. To our knowledge, this marks the first initiative of its kind. It already includes 20,000 OptuNets trained on MNIST, 10,000 ResNet-18 trained on CIFAR-100, and 1000 ResNet-18 trained on TinyImageNet. To improve the generalizability of this dataset, we will also include LSTMs trained on the exchange rate dataset for time series forecasting and small regression MLPs trained on the energy-efficiency dataset.
2. The link between symmetries and posterior: While the connection between symmetries and the posterior is not a fully new concept, our paper introduces novel theoretical insights without unrealistic hypotheses (Equation (8) and Proposition 2).
3. Uncertainty and posterior estimation: We evaluate the relationship between posterior estimation quality and uncertainty quantification on modern neural networks and demonstrate that it is intricate and non-trivial.
4. Qualitative insights: Although we have not fully quantitatively proven the significance of studying symmetries for posterior assessment - due to the difficulty of experimenting with the posterior distribution of modern networks - we present qualitative evidence, particularly in visualization.

Given the extensive effort invested in this paper, we would like to ensure that we fully understood some remarks that we found surprising.

We value the constructive comments received and have individually addressed them. In our revised manuscript, we have already made substantial changes, including correcting the ResNet-18 metrics (**24pf**), adding the Adaptive Calibration Error and calibration error plots (**Mjnv** and **hdLm**), clarifying the texts corresponding to Figures 1 and 2 (**QYDw**), adding a table of notations (**hdLm**), as well as further amendments highlighted in red text. Concerning clarity, we updated Figures 1 and 3, as Reviewer **hdLm** suggested.

Additionally, we plan to implement the following changes by the end of the rebuttal:

1. Train networks on CIFAR10 to better compare to the work of Izmailov et al. [**1**], as suggested by Reviewer **Mjnv**; **done**
2. Generalize our results on the min-mass problem to our ResNet-18s, as suggested by Reviewer **24pf**;
3. Add results on preconditioned stochastic Langevin dynamics on MNIST with OptuNet, as suggested by Reviewer **Mjnv**; **done**
4. Add a discussion on the nature of the estimation of the posterior with deep ensembles, as suggested by Reviewer **Mjnv** and highlighted in our answers to Reviewers **QYDw** and **24pf**; **done**
5. Improve the generalizability of our dataset with new architectures and tasks, as suggested by Reviewer **cm11,** and add a section on the dataset in the paper;  **done**
6. Add a section on loss landscape and deep ensembles, linked to the discussion with Reviewer **cm11**; **done**
7. Additionally, we acknowledge the criticisms cast on the MMD as a measure of distance between the distribution sampled by training independent neural networks. Therefore, we will propose another more interpretable discrepancy based on the marginals and compare the results with the MMDs. **done**

A complete version incorporating these adjustments will be promptly provided. We welcome any further questions or concerns and look forward to addressing them swiftly.

Finally, we are committed to Open Science. This is why we will also make all our codes available in addition to the key code samples already included in the supplementary materials.

Sincerely,

The authors


### References

[**1**] Izmailov, Pavel, et al. "What are Bayesian neural network posteriors really like?."In ICML. 2021.

*[Edit]: specified the delivered experiments*

---

### Author Response · Authors · 2023-11-18

Dear Area Chair and dear Reviewers,

We greatly acknowledge Reviewer **hdLm**'s effort to react swiftly to our response and to further focus on the unclear parts. We are aware that your time is very limited for this voluntary effort, and any feedback and engagement in discussion would be tremendously appreciated.

In the meantime, we have updated the document, improving the clarity and notably adding a discussion on the estimation of the posterior using deep ensembles in Appendix D.2. In this discussion, we summarize briefly the controversy around deep ensembles belonging or not to the Bayesian framework. We then detail some strengths and weaknesses of the different methods that have been proposed for posterior estimation (MCMC, ensembles, Laplace, VIs, and a bit of Langevin) and highlight the recent work of Wild et al. [**1**] (**QYDw, 24pf, hdLm, Mjnv**). We have also added pSGLD on MNIST (**hdLm**), a section detailing mutual information (**hdLm)**, a section detailing our checkpoint dataset (Appendix D.1), and improved the clarity of Section 5.1 (**24pf**, **hdLm**). Concerning [**2**] (**Mjnv**), we have downloaded the checkpoints trained on CIFAR-10. Izmailov et al. use the swish activation function, which is not non-negative homogeneous and, therefore, has no exact scaling symmetries. We will still extend our dataset with 1,000 more checkpoints using Izmailov et al.’s architecture and provide a comparison of the two estimations of the posterior.

Sincerely,

The authors

### References:

[**1**] Wild, Veit David, et al. "A Rigorous Link between Deep Ensembles and (Variational) Bayesian Methods." In NeurIPS 2023.

[**2**] Izmailov, Pavel, et al. "What are Bayesian neural network posteriors really like?."In ICML. 2021.

---

### Author Response · Authors · 2023-11-22
**General response to the Area Chair and the Reviewers**

Dear Area Chair and dear Reviewers,

We thank Reviewers **Mjnv**, **hdlm**, and **cm11** for their commitment to our discussions and helpful feedback. We feel that taking your concerns into account helped improve our work, and we will recognize your contribution in the Acknowledgements section of our work. We have striven to answer the reviewers’ concerns and questions. We are obviously disappointed by the lack of engagement of Reviewers **24pf** and **QYDw**, whose concerns are now nearly all addressed thanks to our discussion with the other reviewers (but we can understand given the current circumstances in the community, in which everyone is swamped with duties).

We have updated a new version of the document. We have upgraded the discussion on the posterior estimation by independently trained networks to a full section, which is now Appendix B (**hdLm, Mjnv, 24pf, QYDw**). Additionally to the improvement of the small literature review on this topic (**hdLm**), we have added a section on the comparison of the computational complexities of HMC and DE (**Mjnv**), as well as experiments showing some limits of HMC - despite its impressive results. Indeed, we show that contrary to DE, HMC’s posterior estimation goes in contradiction with Proposition 1: HMC - even in the best conditions - is biased on certain permutations, most likely due to its sequential walk and the huge size of the parameter space. We have also provided qualitative visualizations of the marginals that hint that DE’s posterior estimation is more multi-modal than HMC’s. We present additional arguments and references on this topic in the discussion with Reviewer **Mjnv**.

Furthermore, we have added a discussion on the Bayesian prior and its impacts in Appendix E.2 (**hdLm**) and a discussion on the link between our results on the min-mass problem on the loss landscape (**hdLm, cm11**) in Appendix E.3. We take advantage of this new section to perform experiments that enable us to provide a hypothesis as to why neural networks do not converge to the optimum of the minimum-mass problem (**hdLm**). We also rewrote the introduction of the MMDs and briefly introduced their formalism in C.5 (**Mjnv**). All these discussions are duly referenced in the core paper. We remain focused on delivering our final experiments.

Sincerely,

The authors

---

### Author Response · Authors · 2023-11-22
**Authors' Final General Answer**

Dear Area Chair and dear reviewers,

We again thank Reviewers **hdLm, Mjnv, cm11**, and **QYDw** for their commitment and constructive remarks regarding our work. We feel that addressing their concerns and questions has significantly improved the paper's quality. We will refrain from rewriting all the (numerous) changes we made during the rebuttal since they are listed in the previous general responses.

Since the last answer, we have finished filling Table 2, which provides the performance of our baselines on CIFAR-10, compares the MMDs estimated using the posterior estimations based on HMC and DE, and developed a small interpretation of the results. Furthermore, we propose a small comparison between distances of posteriors, based on MMDs vs. a Wasserstein distance on the marginals in the new section C.6.

We remain available, for instance, for any eventual concern or question of Reviewer **Mjnv** or the answer of Reviewer **24pf**.

We will continue to work and improve the draft after the discussion deadline. Notably, we will build on the toy example that we added to the draft to research insights that we could use from smaller-scale experiments. We will also rearrange the Appendix and improve its writing.

We are grateful for all the remarks and questions that we found of commanding quality.

Sincerely,

The authors

---

### Meta-Review · Area_Chair_4wmA · 2023-12-06

**Metareview:**

This paper studies weight-space symmetries in BNN posteriors, particularly due to permutations and weight scaling, and provide a set of empirical observations as well as a large set of pre-trained checkpoints for further study. After an active discussion between authors and reviewers, the paper is very borderline, with three reviewers leaning accept and two reject. While the reviewers praised the originality, comprehensive empirical results, study of uncertainty estimation, technical quality, they were critical of the focus, practical usefulness, and connectedness of theory and experiments. However, many (if not all) of these points seem to have been addressed in the very extensive rebuttal provided by the authors. Given the active improvements to the paper and the fact that the most critical reviewer has unfortunately not acknowledged the rebuttal, which (as far as I can tell) addresses many of their criticisms, I think it is warranted to accept this paper based on the understanding, that the authors will continue their efforts to address all reviewer comments in their camera-ready version.

**Justification For Why Not Higher Score:**

the scores are rather borderline

**Justification For Why Not Lower Score:**

I believe that the paper has drastically improved over the rebuttal period, and most reviewers agree

---

### Decision · Program_Chairs · 2024-01-16

Accept (poster)